# Approach for Combining Fault and Area Sources in Seismic Hazard Assessment: Application in South-Eastern Spain

Alicia Rivas-Medina[1,2], Belen Benito[1], Jorge Miguel Gaspar-Escribano[1]

[1] Departamento de Ingeniería Topográfica y Cartografía, Universidad Politécnica de Madrid, Madrid, Spain.

[2] Departamento de Ingeniería Civil, Universidad de Concepción, Concepción, Chile.

*Correspondence to*: A. Rivas-Medina (alicrivas@udec.cl)

**Abstract.** This paper presents a methodological approach to seismic hazard assessment based on a hybrid source model composed by faults as independent entities and zones containing residual seismicity. The seismic potential of both types of sources is derived from different data: for the zones, the recurrence model is estimated from the seismic catalog. For fault sources, it is inferred from slip rates derived from paleoseismicty and GNSS (Global Navigation Satellite System) measurements.

Distributing the seismic potential associated with each source is a key question when considering hybrid zone and fault models, and this is normally resolved using one of two possible alternatives: 1) considering a characteristic earthquake model for the fault and assigning the remaining magnitudes to the zone, or 2) Establishing a cut-off magnitude, Mc, above which the seisms are assigned to the fault and below which they are considered to have occurred in the zone. The present paper presents an approach to distributing seismic potential between zones and faults without restricting the magnitudes for each type of source, precluding the need to establish cut-off Mc values beforehand. This is the essential difference between our approach and other approaches that have been applied previously.

The proposed approach is applied in southern Spain, a region of low-to-moderate seismicity where faults move slowly. The results obtained are contrasted with the results of a seismic hazard method based exclusively on the zone model. Using the hybrid approach, acceleration values show a concentration of expected accelerations around fault traces, which is not appreciated in the classic approach using only zones.

## 1 Introduction

Active faults are the main earthquake sources in the crust. However, their incorporation in seismic hazard assessment is not straightforward since there is not enough data available to adequately model them. This leads to a limited use of faults as independent sources in seismic hazard analyses and to an extended use of seismic zones that cover a significant portion of the crust, assuming uniform seismic characteristics within each source.

This situation has begun to change in recent years, as more studies on active tectonics, paleoseismicity and fault deformation rates derived from GNSS measurements among others, become available. These recently available studies constrain fault

parameters such as rupture plane geometry, predominant sense of slip, slip rates, etc. (e.g., Dixon et al., 2003; Langbein and Bock, 2004; Papanikolaou et al., 2005; Walpersdorf et al., 2014 and Metzger et al., 2011).

Taking "fault type" rather than zones into consideration in seismic hazard studies requires addressing two factors: the 3D geometry of the source and the data required to characterize its seismic potential. In most practical cases, the seismic potential of faults is characterized based on the slip rate using characteristic earthquake models proposed by Wesnousky (1986) (for instance: Field et al., 2014; Akinci and Pace, 2017) instead of Gutenberg–Richter recurrence models (Parsons and Geist, 2009). Other approaches, such as extracting the seismic parameters of every single fault from the earthquake catalogue are not always viable, especially in areas with slow-moving faults. Additionally, the period considered in the catalogue may be too short compared with the recurrence time of the fault to provide an unbiased estimation of fault seismic parameters.

In principle, modeling all existing active faults as independent entities could be conceived as the most accurate source model for seismic hazard assessment. However, this vision is still rather idealistic. A more realistic view would include only a limited number of active faults (those with the highest seismic activity) as independent sources. Accordingly, small faults that generate low-magnitude events or slow faults that produce rare events cannot be properly characterized. To prevent a possible deficit in the seismic source model for a given region, the use of faults as seismic sources may be completed with zones that account for the seismic potential associated with these small/slow faults or simply with unknown faults that cannot be characterized independently. Hence, we propose considering a hybrid source model composed of faults and zones: the first modeled as independent sources and the second including residual seismicity.

Adequately establishing the distribution of seismic potential using a model that combines zones and faults poses a challenge, since these are derived from different data sources. For zones, the recurrence model is calculated based on the seismic catalog, whereas for faults, the recurrence model is derived from fault geometries and slip rate estimates based on GNSS-measured deformation rates. The problem is that some of the events contained in the catalogue may be associated with the faults and may have already been included when calculating the seismic potential of the faults based on the slip rate estimates. If all events are assigned to the zone, the events associated with the faults would be counted twice, leading to an overestimation of the total seismic potential (for both faults and zones).

Some authors assign initial β-values to seismic sources (e.g., Bungum, 2007) or propose a simple way of distributing the seismic potential based on a uniform magnitude value, Mc; assigning events with a magnitude lower than Mc to the zone and events with magnitude higher than Mc to the faults (Frankel, 1995; Woessner et al., 2015). The question is: how is this Mc value determined? Why can the fault not generate events with a magnitude below the Mc value? In a study region such as southern Spain, with slow faults and maximum magnitudes around 6.5-7.0, it is difficult to choose an Mc in a non-arbitrary manner.

The approach presented in this paper addresses the challenging question of how to estimate the anticipated ground motion exceedance rate, using a short period of earthquake observations and limited geological data (with significant uncertainties). This challenge is common to all probabilistic seismic hazard models (Kijko et al., 2016). The purpose of this study is to

approach this challenge proposing a model that contains different types of seismic sources (faults and zones) and adequately distributes the seismic potential, preventing double counting and taking completeness periods into account.

An application of the approach presented is carried out in SE Spain, the area with the highest seismic hazard in Spain. Most of the previous work that partly or wholly addresses this area includes zones only (García-Mayordomo et al., 2007; Benito et al., 2010; Mezcua et al., 2011; IGN-UPM working group, 2013; Salgado-Galvez et al., 2015) or is based on zoneless methods (Peláez and López Casado, 2002; Crespo et al., 2014). A first attempt to combine faults and zones was carried out by García-Mayordomo (2005), who developed a zone model for the area taking into account the use of the characteristic earthquake model for faults.

## 2 Source (Zones and Faults) Hybrid Approach to Hazard Estimation

The hybrid model proposed is composed of fault-type sources and zone-type sources. In addition, the term "region" is defined as the geometric container for both source types. Thus, the region presents the same geometry as the zone and its seismic potential (seismicity rate and seismic moment rate,) is the sum of the potentials of the two types of sources (faults and zone). The zone is used to represent the seismic potential of events that cannot be associated with specific faults. Although there is a geometrical equivalence between region and zone, their seismic potential is very different, as the seismic potential of the region equals the seismic potential of the zone plus the seismic potential of the faults contained within the region (Fig. 1).

The problem is then how to distribute the seismic potential of the region between the zone and the faults without counting some faults twice. The following considerations were taken into account:

- The seismicity rate of the region is derived from the seismic catalogue after excluding the events that lie outside their respective completeness periods CP(m). This period is defined for a given magnitude M as the period during which catalogue of events of magnitude M and higher is complete. This is comparable to assuming that all events of a given magnitude, M, that have actually occurred are effectively contained in the catalogue within the period CP(m=M) (and not outside of this period).
- The completeness periods, CP(m), for different magnitudes up to a maximum completeness magnitude value, $M_{MaxC}$, are lower than the observation period (OP) of the catalogue.
- Magnitude values above $M_{MaxC}$ present recurrence periods higher than the catalogue OP. These values usually constitute a sample that does not include a high enough number of records to clearly establish the recurrence period, as this makes it increasingly difficult to constrain rates for rarer events.

By representing the number of recorded events for different magnitude intervals as a function of time it is possible to identify the reference years, RY(m), for different magnitude intervals using the slope method (Hakimhashemi and Grünthal, 2012), also known as the temporal course of earthquake frequency (TCEF) (Nasir et al., 2013). This method consists of plotting the cumulative number of earthquakes of a given magnitude range over time and estimating the year, presenting a significant

gradual change in slope (Fig. 2). Consequently, CP(m) and $M_{MaxC}$ values can be calculated for m<= $M_{MaxC}$. It is then possible to estimate seismicity rates (Eq. 1) and the seismic moment in each magnitude interval (Eq. 2) as follows:

$$\dot{n}(m) = \frac{n(m)}{CP(m)} \tag{1}$$

where $\dot{n}(m)$ is the annual rate of events with magnitude (m) and n(m) is the number of recorded events with magnitude (m) in the catalogue in the completeness period CP(m).

$$\dot{M}_o(m) = \dot{n}(m) \cdot Mo(m) \tag{2}$$

where $Mo(m)$ $is$ the seismic moment released by events of magnitude m, obtained using the equation proposed by Hanks and Kanamori (1979) $(Log(Mo) = 1.5 \cdot Mw + 16.1)$.

Finally, the cumulative rates in the interval [$M_{Min}$, $M_{MaxC}$] can be estimated, where $M_{Min}$ is the minimum magnitude value used to compute seismic hazard, as shown in section 2.1. This is illustrated with an example in Fig. 2, with [$M_{Min}$, $M_{MaxC}$] =

[4.0, 5.9].

Although faults are capable of generating earthquakes with magnitude m > $M_{MaxC}$, the distribution of seismic potential is carried out in the completeness period [$M_{Min}$, $M_{MaxC}$]. In this way, we avoid using magnitudes with long recurrence periods that have not been recorded in the catalogue within the completeness periods. The computation of the seismic potential of the fault in the interval [$M_{MaxC}$, $M_{Max(fault)}$], where $M_{Max(fault)}$ is the maximum magnitude value of events that can be generated in a

fault, is constrained with other geological criteria (see below).

The seismic potential is represented by the total rate of earthquakes ($\dot{N}min$) and the cumulative rate of seismic moment ($\dot{M}o$), for the magnitude range [$M_{Min}$, $M_{MaxC}$], in the completeness period CP(m). Details on how to determine $\dot{N}min$ and $\dot{M}o$ for the entire region, the corresponding zone and faults are explained in the following section.

**2.1 Seismic Potential of the Region**

The $\dot{N}min$ and $\dot{M}o$ values representing the seismic potential of the region are derived from the seismic catalogue of the magnitude interval [$M_{Min}$, $M_{MaxC}$] for the completeness periods CP(m).

$$\dot{N}min_{region}\Big|_{M_{Min}}^{M_{MaxC}} = \sum_{M_{Min}}^{M_{MaxC}} \dot{n}(m) \tag{3}$$

$$\dot{M}o_{region}\Big|_{M_{Min}}^{M_{MaxC}} = \sum_{M_{Min}}^{M_{MaxC}} \dot{n}(m) \cdot Mo(m) \tag{4}$$

with $\dot{n}(m)$ the annual rate of events with magnitude (m) recorded in CP(m) and Mo(m) the seismic moment for magnitude

m. The notation $X\Big|_{Mi}^{Mj}$ represents the magnitude interval (Mi, Mj) in which variable X is computed.

## 2.2 Seismic Potential of Faults

The cumulative moment rate of the faults is estimated assuming that the fault planes are accumulating energy evenly and using the equation proposed by Brune (1968):

$$\dot{M}o_{fault} = \upsilon \cdot \mu \cdot A \tag{5}$$

Where $\upsilon$ is the slip rate, $\mu$ is the shear modulus and A is the area of the fault plane.

The slip rate $\upsilon$ and the area of each fault plane can be derived from specific studies based on paleoseismic analyses and GNSS measurements. There are also some databases available to search for these data, including the EDSF for Europe (Basili et al., 2013), the DISS for Italy (DISS Working Group, 2010) and the QAFI for Spain and Portugal (Garcia-Mayordomo et al., 2012). For the shear modulus may be estimated from values close to $\mu = 3.2 \times 10^{10}$ Pa (Walters et al., 2009; Martínez-Díaz et al., 2012)

This moment rate represents the average annual seismic moment accumulated in each fault that will be released by earthquakes of different magnitudes $m=0$ up to the maximum magnitude of the fault, $M_{Max(fault)}$. The value $M_{Max(fault)}$ can be evaluated from a geometrical aspect of the fault planes using empirical relationships proposed in the literature, as Wells and Coppersmith (1994), Stirling et al. (2002) or Leonard (2010) among others. Thus, $\dot{M}o_{fault}$ can be expressed as (Eq. 6):

$$\dot{M}o_{fault} = \int_{Mm=0}^{MMax(fault)} \dot{n}(m) \cdot Mo(m) \, dm \tag{6}$$

where $\dot{n}(m)$ can be estimated applying a recurrence model, as for instance, the modified GR model shown in Eq. 7.

$$\dot{n}(m) = \dot{N}min_{fault} \cdot \beta \left( \frac{e^{-\beta m}}{e^{-\beta(M_{m=0})} - e^{-\beta(M_{Max(fault)})}} \right) \tag{7}$$

and the seismic moment $Mo(m)$, can be estimated from the Hanks and Kanamori (1979) relation expressed in exponential terms, $Mo(m) = e^{dm+c}$ with $c = 16.1 \cdot \ln(10)$ and $d = 1.5 \cdot \ln(10)$, where m is the moment magnitude Mw (Anderson and Luco, 1983).

Substituting the previous relations in Eq. (6), solving the integral and reordering the equation for $\dot{N}min_{fault}$, we get Eq. 9.

$$\dot{N}min_{fault} = \frac{\dot{M}o_{fault} \cdot (d-\beta) \cdot \left( e^{-\beta(M_{m=0})} - e^{-\beta(M_{Max(fault)})} \right)}{\beta \cdot [e^{-\beta M_{Max}} Mo(M_{Max(fault)}) - e^{-\beta M_{m=0}} Mo(M_{m=0})]} \tag{8}$$

where $M_{m=0}$ is the minimum magnitude that may be generated at a fault rupture (here taken as m=0), $M_{Max(fault)}$ the maximum magnitude, and $\dot{M}o_{fault}$ the seismic moment rate accumulated in the fault (Eq. 5).

The total seismic moment rate for each fault ($\dot{M}o_{fault}$) and seismicity rate ($\dot{N}min_{fault}$) can be formulated as the sum of the seismic moment rate released at different magnitude intervals, thus, it follows:

$$\dot{N}min_{fault} = \dot{N}min\Big|_{M_{M=0}}^{M_{Min}} + \dot{N}min\Big|_{M_{Min}}^{M_{MaxC}} + \dot{N}min\Big|_{M_{MaxC}}^{M_{Max(fault)}} \tag{9}$$

$$\dot{M}o_{fault} = \sum_{M_{M=0}}^{M_{Min}} \dot{n}(m) \cdot Mo(m) + \sum_{M_{Min}}^{M_{MaxC}} \dot{n}(m) \cdot Mo(m) + \sum_{M_{MaxC}}^{M_{Max(fault)}} \dot{n}(m) \cdot Mo(m) \tag{10}$$

By implementing a recurrence model, it is possible to derive the seismicity rate and the moment rate in the interval [$M_{Min}$, $M_{MaxC}$] (see example in Fig. 3 with [$M_{Min}$, $M_{MaxC}$] = [4.0, 6.9]).

In this approach it is considered that all faults included in the same region will present the same β-value and different seismicity rates ($\dot{N}min_{fault}$), as this parameter depends on the seismic moment rate of each fault ($\dot{M}o_{fault}$).

## 2.3 Seismic Potential of the Zone

The parameters representing the zone are initially unknown. They can be calculated for the interval [$M_{Min}$, $M_{MaxC}$] given that:

$$Seismic\ Potential_{zone} = Seismic\ Potential_{region} - Seismic\ Potential_{faults} \tag{11}$$

Or specifically:

$$\dot{N}min_{zone}\big|_{M_{Min}}^{M_{MaxC}} = \dot{N}min_{region}\big|_{M_{Min}}^{M_{MaxC}} - \sum\dot{N}min_{fault}\big|_{M_{Min}}^{M_{MaxC}} \tag{12}$$

$$\dot{M}o_{zone}\big|_{M_{Min}}^{M_{MaxC}} = \dot{M}o_{region}\big|_{M_{Min}}^{M_{MaxC}} - \sum\dot{M}o_{fault}\big|_{M_{Min}}^{M_{MaxC}} \tag{13}$$

In principle, there are 2 equations with two unknowns related to the zone: $\dot{N}min_{zone}\big|_{M_{Min}}^{M_{MaxC}}$ and $\dot{M}o_{zone}\big|_{M_{Min}}^{M_{MaxC}}$.

Regarding the faults, $\dot{N}min_{fault}\big|_{M_{Min}}^{M_{MaxC}}$ and $\dot{M}o_{fault}\big|_{M_{Min}}^{M_{MaxC}}$ are derived using an initial (not definitive) β-value. Regarding the region, $\dot{N}min_{fregion}\big|_{M_{Min}}^{M_{MaxC}}$ and $\dot{M}o_{region}\big|_{M_{Min}}^{M_{MaxC}}$ are known, as they were extracted from the catalogue (Eqs. 1 and 2). A new additional equation is obtained relating $\dot{N}min_{zone}\big|_{M_{Min}}^{M_{MaxC}}$ and $\dot{M}o_{zone}\big|_{M_{Min}}^{M_{MaxC}}$ using Eq. (8) for the interval [$M_{Min}$, $M_{MaxC}$] in the zone, resulting in:

$$\dot{N}min_{zone}\big|_{M_{Min}}^{M_{MaxC}} = \frac{\dot{M}o_{zone}\big|_{M_{Min}}^{M_{MaxC}} \cdot (d - \beta_{zone}) \cdot \left(e^{-\beta_{zone}(M_{Min}) - e^{-\beta_{zone}(M_{MaxC})}}\right)}{\beta_{zone} \cdot \left[e^{-\beta_{zone}M_{MaxC}}Mo(M_{MaxC}) - e^{-\beta_{zone}M_{Min}}Mo(M_{Min})\right]} \tag{14}$$

Notice that Eq. 8 and Eq.14 are similar: they differ in the type of source and computation interval. Eq. 8 is for faults and it is computed in the magnitude interval [$M_{m=o}$, $M_{Max}$]. Eq. 14 is for zones, and the magnitude interval is restricted to [$M_{Min}$, $M_{MaxC}$]. Also note that the β-value of the zone in this equation can be equal to the β-value of the region as both sources present similar seismic natures.

With this third equation, it is possible to solve the system and obtain a new β-value for the faults (second iteration) that balances the three equations. The result is the distribution of seismic potential between the zone and the faults in the interval [$M_{Min}$, $M_{MaxC}$].

Considering that the faults may generate events with magnitudes larger than $M_{MaxC}$, the corresponding distribution of seismic potential in the interval ($M_{MaxC}$, $M_{Max(fault)}$] is calculated by extrapolating the recurrence model with the last β-value adjusted (Fig. 4).

Regarding the $M_{Max}$ value expected for the zone ($M_{Max(zone)}$), this can be considered equal to $M_{MaxC}$ or extended to a higher magnitude value if it is assumed that bigger events can occur in other unidentified sources (such as blind faults).

## 2.4 Analysis of Uncertainty

The proposed approach strongly relies on computing seismicity, earthquake rates and moment rates, within the magnitude interval [$M_{Min}$, $M_{MaxC}$] of the seismic catalogue that contains the complete record of events that have occurred in the entire region.

In order to capture the variability of seismic moment rates calculated from the earthquake catalogue, a sensitivity analysis of three key factors is conducted. These factors are: 1) the number of records used to compute moment rates, 2) the magnitude range covered by the complete catalogue, and 3) the proportion of earthquakes of different magnitude (b-value).

Synthetic catalogues derived from GR-modified recurrence models are generated for this purpose. Earthquake rates are computed using different numbers of events, magnitude intervals and b-values that could be representative of areas of low-

to-moderate seismic activity.

The procedure consists of five steps:

- Generating 2000 synthetic catalogues for different combinations of earthquake rates, magnitude intervals and b-values.

- Calculating earthquake rates for different magnitude values for each synthetic catalogue (Eq. 6).

- Calculating moment rates for different magnitude values for each synthetic catalogue.

- Calculating the sum of moment rates for different magnitude values in order to obtain the cumulative moment rate for each synthetic catalogue (Eq. 7).

- Computing the mean and the standard deviation of the distribution of calculated seismic moment rates.

Table 1 shows the coefficient of variation (COV = standard deviation /mean) associated with each combination: number of

events, magnitude interval and b-value. As can be seen, the greater the number of records in the sample and the lower magnitude range, the lower the uncertainty associated with the rate of seismic moment calculated. The b-value presents a different trend, recording the greatest variability for b-values between 1.0 and 1.5. This table is useful to estimate the uncertainty of the seismic moment rate calculated from the seismic catalogue as a function of the number of earthquakes, magnitude interval and b-value.

It is also important to consider the uncertainty associated with the slip rate and the area of the fault, as these are propagated into the distribution of seismic moment rates of the fault in proportion to the deviation of the area or slip rate value. The uncertainty of the slip rate value is more relevant for low slip rate values than for large slip rate values (a similar trend can be deduced for low and high area values). For instance, a deviation of ±1 mm/year in a slip rate of ±2 mm/year represents an uncertainty of 50%, leading to a COV value of 0.5 at the moment rate of the fault. However, the same deviation (± 1

mm/year) for a fault with a slip rate of ±10 mm/year represents an uncertainty of 10%, leading to a COV coefficient moment rate of only 0.1 for the fault.

## 3 Application of the Approach in Southeast Spain

The approach described above is applied in south-eastern Spain, the most seismically active area in the country. The tectonic deformation and seismicity is related with the north-western boundary between the Eurasian and African plates (e.g. Kiratzi and Papazachos, 1995), with an approximate shortening rate of about 4 mm/yr (Argus et al., 1989) in roughly NNW-SSE direction. Crustal deformation is accumulated over a broad area in which seismicity is diffuse (Benito and Gaspar-Escribano, 2007).

Assigning earthquakes to specific faults is not an easy task, partly due to errors in earthquake location and to the existence of blind, unknown faults: whereas earthquakes can be clearly associated with a rupture, such as the 2011 M 5.2 Lorca event generated in the Alhama de Murcia fault system (Cabañas et al., 2011), other events have occurred in areas where there are no mapped active faults, as for instance the 2007 Mw 4.7 Pedro Muñoz and 2015 Mw 4.7 Ossa de Montiel earthquakes, both located in Central Spain (QAFI database, García-Mayordomo et al., 2012).

## 3.1 Source Input Data

The seismogenic source model considered for SE Spain is composed of 12 regions that contain a total of 95 faults (Annex) Active fault data are taken from the QAFI database (v2.0) (García-Mayordomo et al., 2012), which includes information about fault segmentation, geometry and slip rate (see Fig. 5). The maximum expected magnitude in each fault is derived from the rupture length using Stirling et al. (2002) equations derived from the instrumental dataset. These equations are chosen because they are also the ones used in the QAFI database in order to ensure consistency with said database. Moment rates accumulated in the faults are estimated using the fault plane area and the slip rate value according to the formula proposed by Brune (1968). A value of $\mu = 3.2 \times 10^{10}$ Pa is assumed for the shear modulus (Walters et al., 2009; Martínez-Díaz et al., 2012).

The zones model proposed by García-Mayordomo et al. (2010) is used to obtain the geometries of the 12 regions (and thus of the zones) that account for the seismicity that cannot be ascribed to faults (see Fig. 5). All the regions considered in this model contain faults sources, with the exception of regions 28, 29, 33 and 40. In these cases, the seismic potential of the corresponding region is assigned to the zones.

The seismic moment released in the region is estimated from the seismic catalogue of Spain homogenized to Mw (IGN-UPM Working Group, 2013; Cabañas et al., 2015). This catalogue contains 1,496 earthquakes, with magnitudes ranging from 4.0 to 6.6. The uncertainty assessment of the catalogue used in this study is explained in Gaspar-Escribano et al. (2015). According to the completeness analysis, a $M_{MaxC}$ of 5.9 is estimated for SE Spain (although not every region reaches this

maximum magnitude value). The recurrence periods for magnitudes higher than 6 present are too long to allow us to establish completeness periods for these magnitude ranges. (see Fig. 2).

Table 2 shows the seismic potential for each region, calculated in the magnitude intervals [$M_{Min}$, $M_{MaxC}$] and [$M_{Min}$, $M_{Max(region)}$]. It is observed that the seismic potential in the first interval up to $M_{MaxC}$, constitutes at least a 60% of the seismic potential in the second interval, up to $M_{Max(region)}$.

Subsequently, a recurrence model (GR-mod) is assigned to all regions, obtaining the corresponding b-values and COV coefficients (see Table 3). Note that zone 30 lacks a COV estimate because the sample of records (only 7) is very limited, and the [$M_{Min}$, $M_{MaxC}$] interval is very narrow, resulting in an increased uncertainty in the hazard estimates for this region. A GR-mod recurrence model is also assigned to the faults. Finally, the distribution of seismic moments among all seismic sources is carried out (Table 4). As may be observed, the seismic moment rate associated with the zone has a strong influence on the estimated seismic hazard of the region. This is due to the limited number of known active faults than can be modelled as independent sources, a common situation in areas with low and moderate seismic activity. However, it is worth noting that the seismic potential of regions 35, 36 and 38 is dominated by the seismic potential of faults.

The seismic hazard calculation is carried out using the software CRISIS2012 (Ordaz et al., 2013), considering the strong motion equation of Campbell and Bozorgnia (2014), which makes it possible to include the fault geometry and the faulting style. The ground motion parameters predicted include peak ground acceleration (PGA) and 15 spectral accelerations within the period range (0.05 - 10 seconds), all obtained in hard soil ($Vs_{30}$=760 m/s) conditions.

## 3.2 Results

Seismic hazard results obtained with the proposed Hybrid Model (HM) and with the Classical Method based in zone (CM) are shown in Fig. 6a and 6b, respectively. Only the geometry of the zone model differs in the two analyses: the ground motion prediction equation (GMPE) and the other calculation parameters are the same in both approaches. The definition of seismic zones applied in the classic method is explained with detail in IGN-UPM Working Group (2013).

PGA estimates for the return period of 475 years using the zone approach (CM) reach maximum values in Granada, Almeria and the Murcia region, around 0.20 g. Minimum PGA values are obtained in Jaen, with values as low as 0.06 g.

Fig. 6a shows the seismic hazard map resulting from applying our approach (HM). It can be seen that the largest accelerations are estimated around the Carboneras Fault and the fault set of Granada, (0.38 g), followed by the Alhama de Murcia and La Viña faults systems (0.30g) and, to a lesser extent, by the Venta de Zafarraya, Carrascoy, Bajo Segura, Baza, Mijas and Cartama fault systems.

The seismic hazard map obtained using the HM displays more spatial variability than the one obtained with the CM, showing maximum values along fault sources that decrease sharply away from the faults. This trend reflects a "source proximity effect," implying higher acceleration values for the surface projection of the fault rupture plane that rapidly decrease away from the fault (by one half at a distance of about 15 km).

The differences between the expected maximum acceleration obtained with the two methods, CM and HM, for return periods of 475 and 4975 years appear in Fig. 7a and 7b, respectively. The trend presented in both maps is very similar for the two return periods. A different case is found in region 30 (Case Lietor fault), a very complex region with scarce seismic activity and large faults with low slip rates (see Annex). Here, the HM gives higher seismic hazard than the CM only for long return periods. For this region, the magnitude range [$M_{Min}$, $M_{MaxC}$] is very small and it is necessary to extrapolate the model to a larger scale, given the high uncertainty shown in Table 3. However, the results reflect that, for longer periods, these slow faults play a relevant role in the seismic hazard of the region (see Fig. 8), where the HM hazard curve reflects a substantial increase in hazard for long return periods.

To clarify how faults are conditioning the final seismic hazard in our model, the seismic hazard curves showing partial contribution of different sources in Murcia, Almeria and Granada are shown in Fig. 9 for PGA and SA (1.0s). For each city, black lines show the total seismic hazard curve and colored lines the seismic hazard curve associated with different sources (zone and faults) for each city.

In Murcia, seismic hazard for short return periods is associated with multiple sources (zone and faults), but for return periods exceeding 475 years (an exceedance probability of 0.1 or lower in 50 years) the seismic hazard is dominated by the Carrascoy Fault. This effect is very similar for PGA and SA (1.0 s).

In Almeria, only two sources, zone 38 and the Carboneras Fault, contribute significantly to seismic hazard. In PGA both sources combine equally to give the seismic hazard for the city, but, for SA (0.1 s), the Carboneras Fault predominates, especially for return periods of more than 475 years.

In Granada, there are many sources contributing to seismic hazard for the city. This is because there are many known faults in its vicinity. Seismic hazard is controlled by zone 35 for PGA and SA (1.0 s) and shorter return periods. This trend changes for return periods greater than 975 years.

Fig. 10 shows the uniform UHS hazard spectra obtained for four cities in the study area. These graphs can be used to compare the maximum accelerations predicted with the CM and HM in different spectral ordinates, evidencing that the trend observed in PGA persists throughout the entire spectrum.

**4 Discussion**

We present a hybrid method (HM) for determining a seismic source model that combines zones and faults as independent sources. The HM is based on the distribution of seismic potential among different sources and does not impose any restriction with respect to the type of recurrence model assigned to seismic sources. Moreover, the HM does not require defining a fixed cut-off magnitude Mc that separates the magnitude range in which faults and zones produce earthquakes, as in the works of Frankel et al. (1996).

This marks a difference with other approaches that model the seismic potential of faults using two alternatives: 1) single-magnitude rupture models, such as Wesnousky's characteristic earthquake model (1986), as seen in Field et al. (2014);

Akinci and Pace (2017) and 2) models that set a fixed cut-off magnitude and assume that the biggest magnitudes take place in the faults and the smaller ones occur in the zone. In contrast, the formulation of the HM considered above uses a GR-type recurrence model for faults, in line with the proposals made by some other authors (Woessner et al., 2015).

One strong point of the HM is that it assures a distribution of seismic potential between faults and zones that prevents double counting of seismicity. This is achieved by computing the seismic potential of faults and zone using the events contained in the completeness period of the catalogue for different magnitude ranges. Identifying the magnitude interval used to distribute seismic potential between zone and faults is of fundamental importance. Specifically, determining MmaxC is crucial in order to adequately limit this distribution: a low MmaxC value leads to a notable extrapolation of the recurrence model for faults

with large rupture planes; a high MmaxC value may not assure the complete record of all events of that magnitude. In applying the method to SE Spain the MmaxC value identified is 5.9, which may seem too low. However, this value is consistent with the low–moderate level of seismicity in the study area. In fact, according to the IGN seismic catalog (www.ign.es), the seismic catalogue does not contain any shallow event with magnitude equal to or higher than 6.0 in the instrumental period.

In addition, for the purposes of hazard calculations, the distribution of seismic potential for the entire magnitude interval (between the Mmin and the Mmax values expected for each source) requires and is dependent on the selected recurrence models to represent fault and zone activities. In this regard, different types of recurrence models may be used: modified Gutenberg-Richter, truncated Gutenberg-Richter (Gutenberg and Richter, 1944), or the models proposed by Main and Burton (1981) or Chinnery and North (1975), among others. The use of one model versus another depends on each

application and on the available data.

The results of applying the HM are compared with the results of the CM in terms of expected accelerations. A single GMPE is used for both calculations. We have not used any other GMPE (or combination of GMPEs through a logic tree) to simplify the calculations and allow a direct comparison of hazard results.

The results obtained with the HM show an increment of expected accelerations near fault traces (in a factor of 2) in relation

to the results of the CM approach. This is consistent with observations of very high ground motions in the epicentral areas of recent earthquakes, such as the 2009 L'Aquila and 2011 Lorca events (Akinci et al, 2010; Cabañas et al., 2014). This increment is achieved at the expense of decreasing expected accelerations in areas located farther away from faults. This is a consequence of the redistribution of seismic potential in the region, which is not increased, but redistributed in several sources (zone and faults).

**5 Conclusions**

An approach for combining zones and faults in a seismic source model is formulated in this paper.

It is based on the distribution of seismic potential among different sources under certain conditions for preventing double-counting of seismicity. Two points of the methodology are critical and must be carefully assessed: the analysis of completeness and the choice of recurrence model used to represent the seismic activity of either source. They are determined by the data available (composition of the seismic catalog, fault slip rates and geometries, etc.) in the study region, and hence not easily automatable and extendible for other areas. Thus, the approach followed in applying it to SE Spain should be reevaluated when applied to a different area. For instance, it is to be expected that implementing this approach in a region with rapidly moving would produce significantly different results, requiring further adjustments. The higher fault slip rates would imply that the faults consume a larger proportion of the seismic potential available, compromising the convergence of the iterative method to obtain the zone β-value.

An initial assumption of the approach is that the seismic moment potential accumulated in active faults is released only seismically. This condition can be easily modified in the formulation presented above. Additional data informing about other ways of releasing seismic energy, as slow slip events or aseismic transients, would help constrain this point.

The seismic hazard map obtained with the HM presents a more heterogeneous aspect compared to the CM seismic hazard map, which assigns a uniform seismic potential to each region. In the HM hazard map, the accelerations expected along fault traces increase and decrease farther away from fault traces, thus keeping the seismic potential budget of the region in balance. This effect can be useful for applications in which the effects of being near a fault must be emphasized, such as urban seismic risk studies for cities located atop active fault planes.

As a final conclusion, we identify some points that require further development and are the focus of an interesting line of research. Specifically, these include: 1) determining catalogue completeness for different time-magnitude intervals in the study area; 2) selecting the recurrence model assigned to fault sources according to the data available, and 3) determining the proportion of seismic potential accumulated in faults that is released through earthquakes

**Acknowledgements**

We would like to thank Dr. M. Ordaz for his time and support during a research stay carried out by ARM at the Instituto de Ingeniería, UNAM. ARM benefited from a pre-doctoral grant funded by the Universidad Politécnica de Madrid.

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

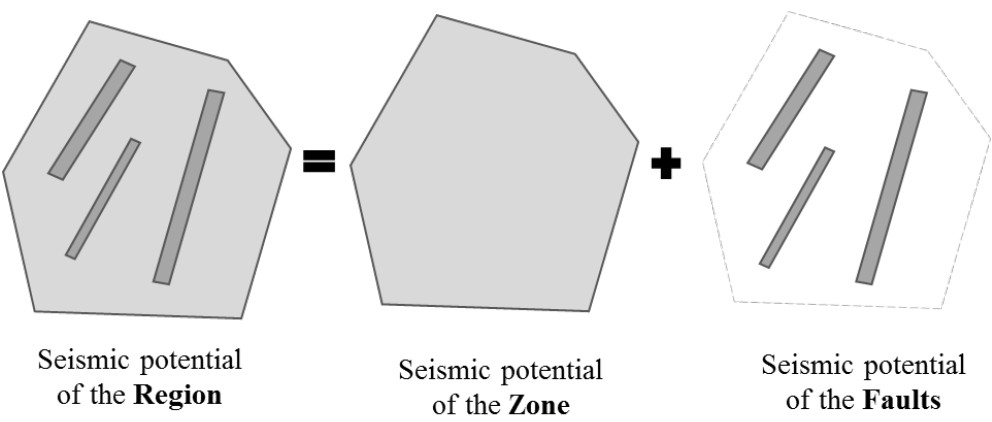

Seismic potential
of the **Region**

Seismic potential
of the **Zone**

Seismic potential
of the **Faults**

**Figure 1. Diagram showing the distribution of the seismic potential of a region, expressed as the sum of the seismic potential of the faults and the seismic potential of the zone.**

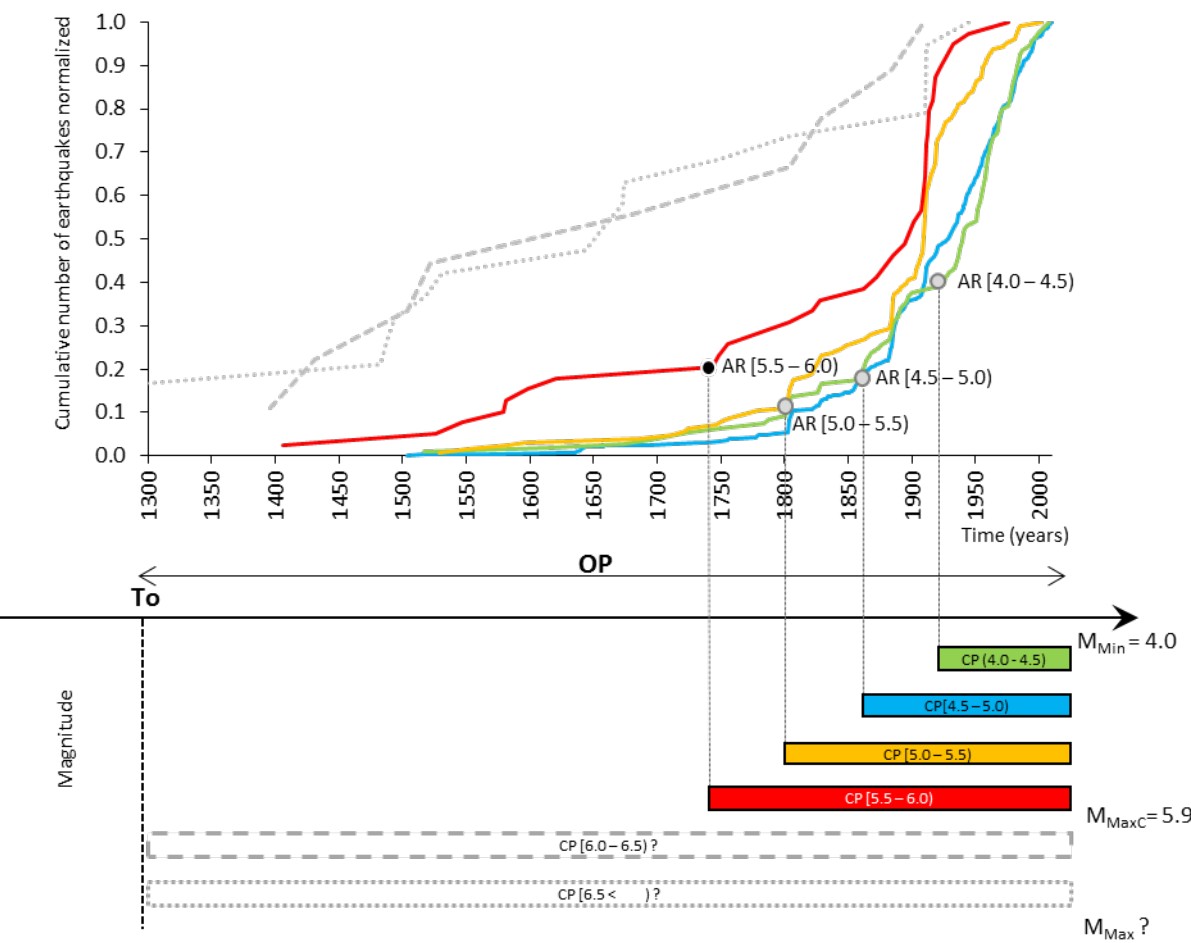

**Figure 2. Completeness analyses of the seismic catalogue. The figure at the top shows the (normalised) cumulative number of earthquakes per year for different magnitude intervals. Solid circles indicate the inflection point that marks the lower limit of the completeness period for the respective magnitude interval CP(m). The figure at the bottom shows the CP(m) corresponding to each magnitude interval. Note that the CP(m) is not well constrained for magnitudes above the MMaxC value (note dashed and dotted curves in both figures).**

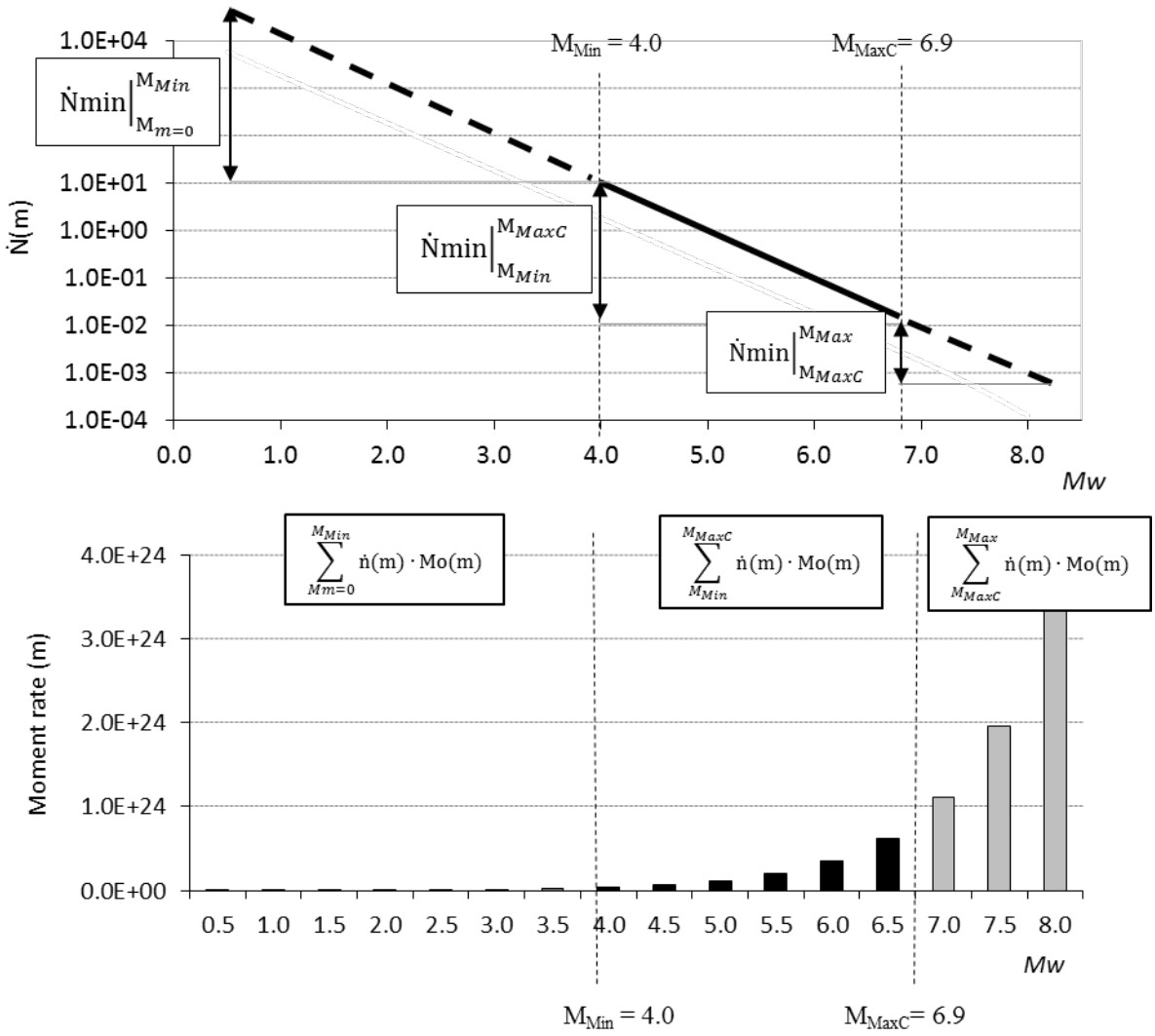

**Figure 3. Seismicity rate (cumulative number of events per year vs. magnitude) (top) and moment rate (cummulative seismic moment per year vs. magnitude) plots. The different magnitude intervals mentioned in the text are marked.**

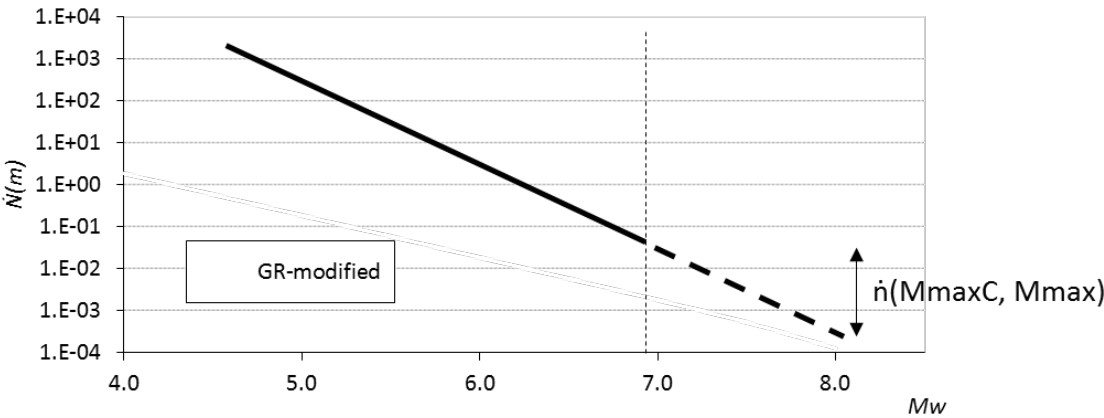

**Figure 4. Graph extrapolating the recurrence model of the fault up to the maximum expected magnitude value, as deduced from geological criteria.**

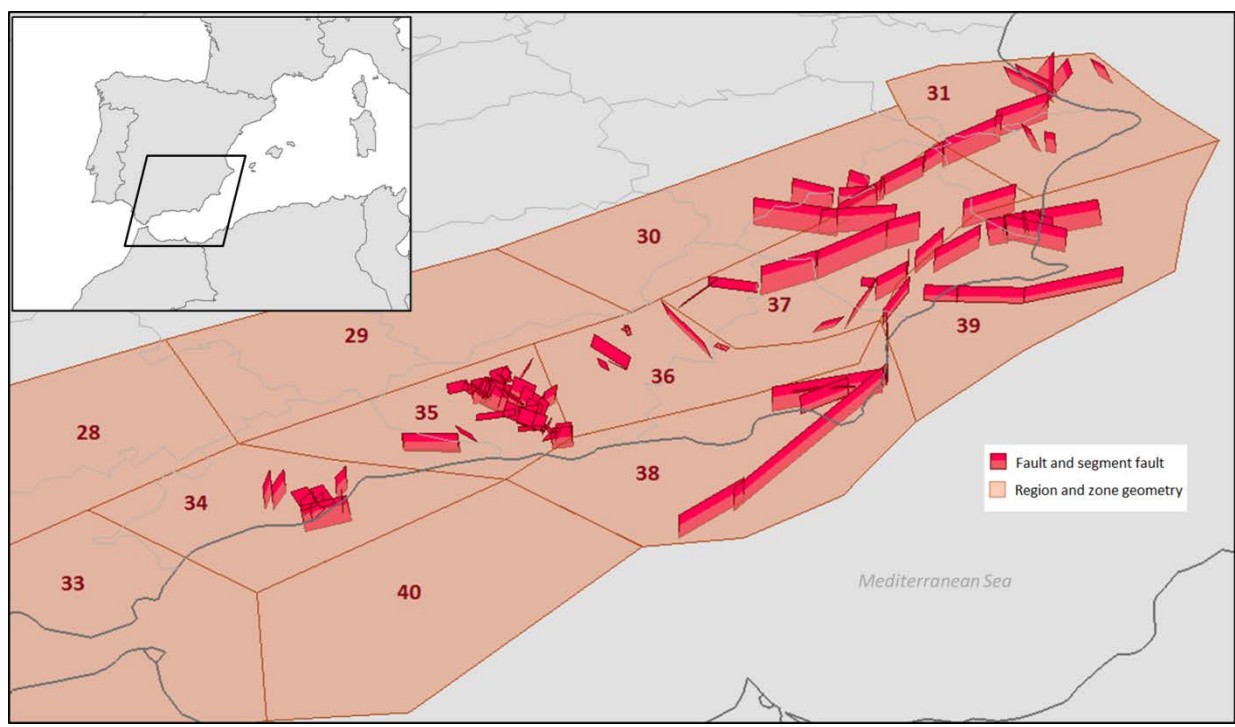

**Figure 5. 3D view of the seismic sources considered for hazard calculation, including faults (red) and zones (brown).**

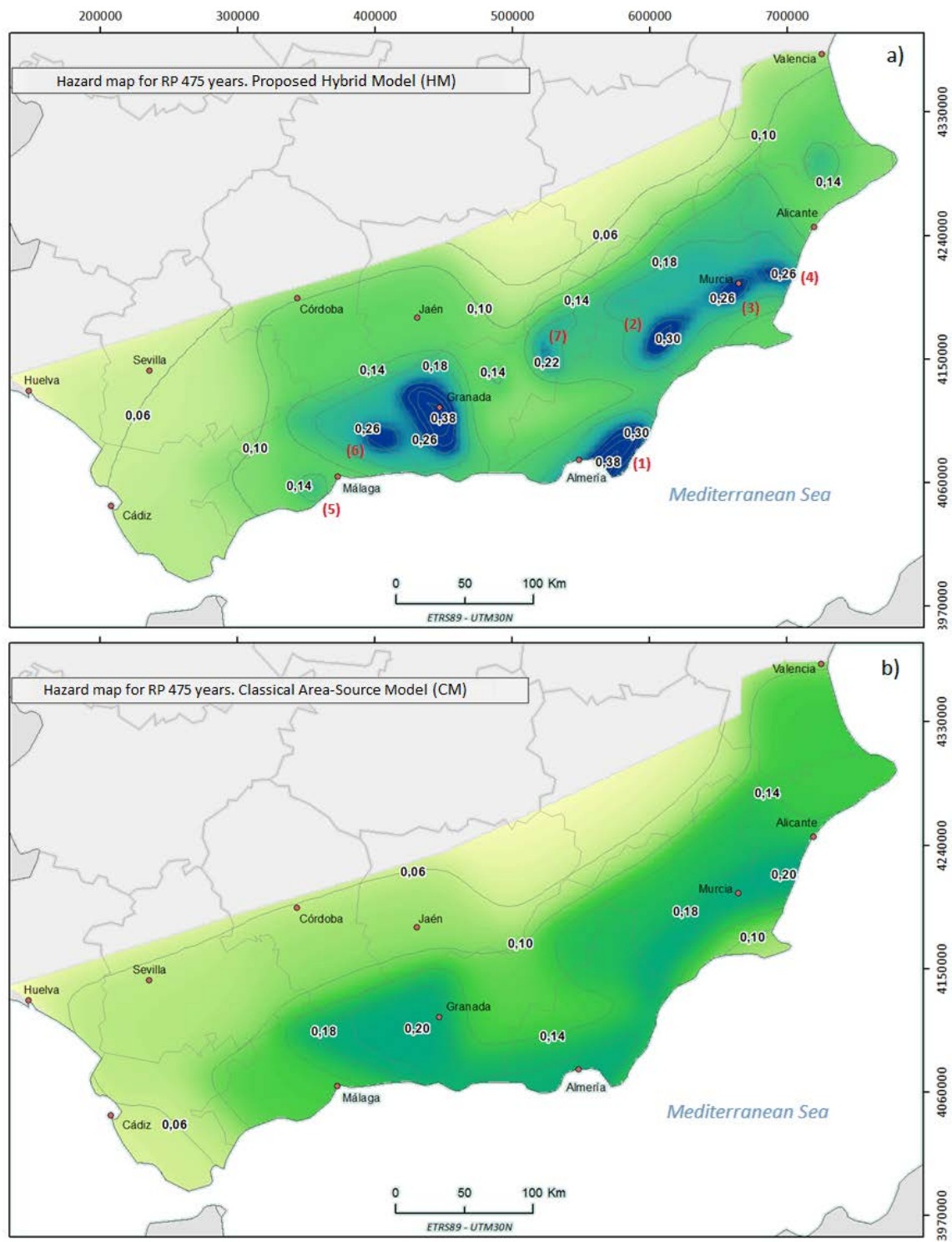

**Figure 6. PGA for return period of 475 years derived from (a) the proposed hybrid approach, and (b) classic zone methodology. Note the fault proximity effects in (a) for these faults: (1) Carboneras Fault, (2) Alhama de Mucia Fault, (3) Carrascoy Fault, (4) Bajo Segura Fault, (5) Mijas and Cartama faults, (6) Zafarraya Fault and (7) Baza Fault.**

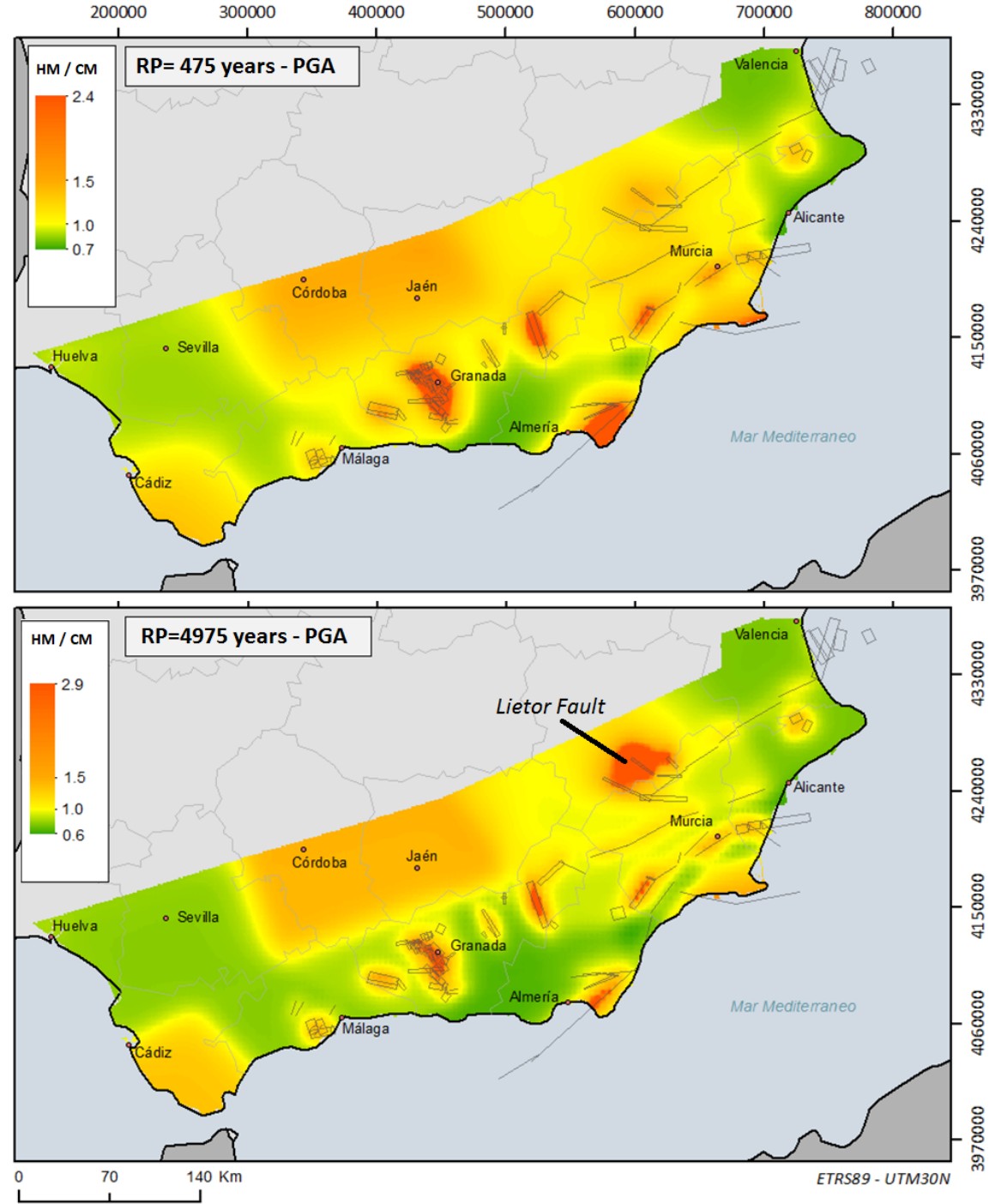

**Figure 7. Seismic Hazard Results Comparison of the two models (HM / CM) for return periods of (a) 475, and (b) 4975 years.**

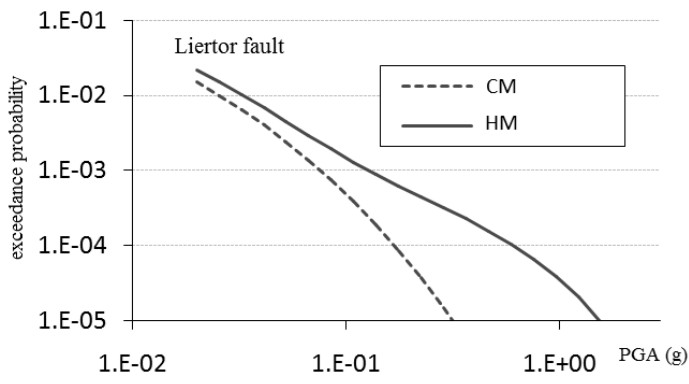

**Figure 8. Seismic hazard curve (with HM and CM) of a site close to the Lietor Fault.**

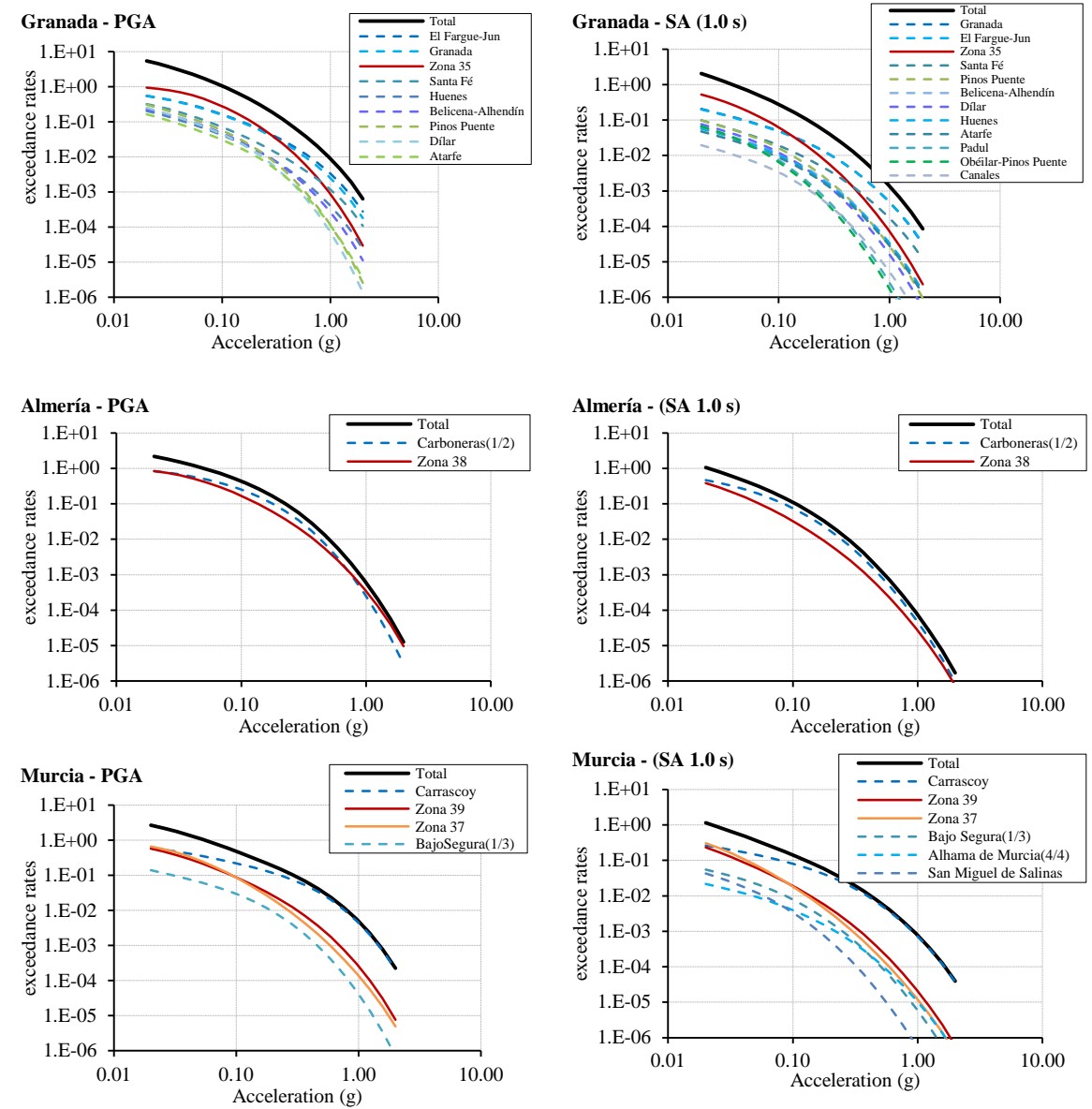

**Figure 9. Seismic hazard curve (with HM) of Murcia, Almería and Granada considering all the seismic sources involved. The black lines show the total seismic hazard curve and the colored lines show the seismic hazard curve associated with different sources (zone and faults).**

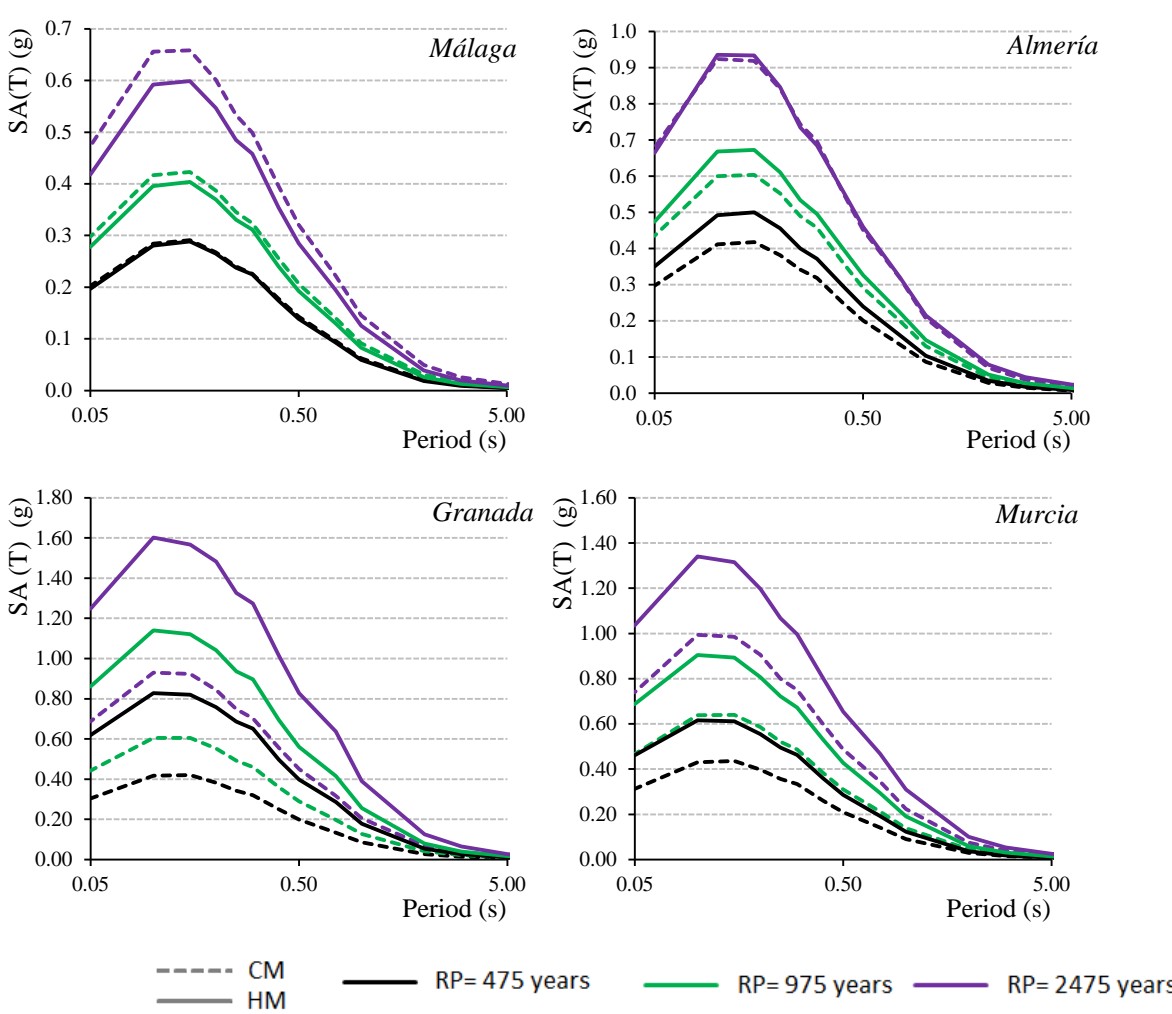

**Figure 10. Uniform Hazard Spectra (UHS) obtained in four cities with CM and HM for three return periods.**

**Table 1. COV coefficient associated with seismic moment rate obtained using synthetic catalogues.**

| records | M(4.0 - 5.0) b | | | | M(4.0 - 6.0) b | | | | M(4.0 - 7.0) b | | | |
|---------|------|------|------|------|------|------|------|------|------|------|------|------|
|         | 0.5  | 1.0  | 1.5  | 2.0  | 0.5  | 1.0  | 1.5  | 2.0  | 0.5  | 1.0  | 1.5  | 2.0  |
| 200     | 0.00 | 0.09 | 0.10 | 0.09 | 0.19 | 0.30 | 0.39 | 0.31 | 0.36 | 1.00 | 1.17 | 0.43 |
| 180     | 0.09 | 0.10 | 0.10 | 0.09 | 0.20 | 0.29 | 0.33 | 0.28 | 0.35 | 0.87 | 1.21 | 0.42 |
| 140     | 0.10 | 0.11 | 0.11 | 0.10 | 0.21 | 0.31 | 0.33 | 0.25 | 0.39 | 0.92 | 1.17 | 0.31 |
| 120     | 0.11 | 0.13 | 0.13 | 0.13 | 0.25 | 0.40 | 0.47 | 0.34 | 0.49 | 1.12 | 0.87 | 0.36 |
| 100     | 0.11 | 0.13 | 0.13 | 0.12 | 0.25 | 0.39 | 0.50 | 0.36 | 0.48 | 1.16 | 1.15 | 0.43 |
| 60      | 0.16 | 0.15 | 0.17 | 0.15 | 0.32 | 0.47 | 0.57 | 0.40 | 0.58 | 1.37 | 1.12 | 0.44 |
| 40      | 0.18 | 0.20 | 0.21 | 0.21 | 0.39 | 0.64 | 0.79 | 0.59 | 0.71 | 1.73 | 1.40 | 0.62 |
| 20      | 0.27 | 0.31 | 0.33 | 0.32 | 0.61 | 0.91 | 1.18 | 0.79 | 1.08 | 2.60 | 2.21 | 0.91 |

**Table 2. Seismic rate and seismic moment rate recorded in the different regions for two magnitude intervals (MMin - MMax) and (MMin - MMaxC) obtained from the seismic catalogue. The table includes the ratio of seismic moment rate of the two intervals, indicating what percentage of the seismic movement rate liberated from Mmin to Mmax is contemplated in the magnitude intervals over which hazard is distributed (MMin – MMaxC). Not that no faults have been catalogued within regions 28, 29, 33 and 40, which is why no values have been assigned (---).**

| Region | $M_{Min}$ | $M_{Max}$ | $M_{MaxC}$ | $M_{Min}$ - $M_{Max}$ | | $M_{Min}$ - $M_{MaxC}$ | | % $\dot{M}o$ recorded in complete periods |
| --- | --- | --- | --- | --- | --- | --- | --- | --- |
| | | | | $\dot{N}(4.0)$ | $\dot{M}o$ (Nm/year) | $\dot{N}(4.0)$ | $\dot{M}o$ (Nm/year) | |
| 28 | 4.0 | 5.4 | --- | 0.211 | 1.90E+22 | --- | --- | --- |
| 29 | 4.0 | 6.2 | --- | 0.176 | 7.90E+22 | --- | --- | --- |
| 30 | 4.0 | 4.6 | 4.6 | 0.053 | 1.86E+21 | 0.053 | 1.86E+21 | 100% |
| 31 | 4.0 | 6.5 | 5.7 | 0.241 | 7.88E+22 | 0.239 | 4.70E+22 | 60% |
| 33 | 4.0 | 5.4 | --- | 0.082 | 1.41E+22 | --- | --- | --- |
| 34 | 4.0 | 6.3 | 5.5 | 0.219 | 3.28E+22 | 0.218 | 2.56E+22 | 78% |
| 35 | 4.0 | 6.5 | 5.5 | 0.574 | 8.73E+22 | 0.570 | 7.09E+22 | 81% |
| 36 | 4.0 | 6.2 | 5.4 | 0.142 | 2.21E+22 | 0.141 | 1.41E+22 | 64% |
| 37 | 4.0 | 6.0 | 5.7 | 0.442 | 1.01E+23 | 0.440 | 9.07E+22 | 90% |
| 38 | 4.0 | 6.5 | 5.4 | 0.527 | 6.75E+22 | 0.525 | 5.70E+22 | 84% |
| 39 | 4.0 | 6.6 | 5.4 | 0.313 | 6.00E+22 | 0.308 | 4.30E+22 | 72% |
| 40 | 4.0 | 6.0 | --- | 0.135 | 4.06E+22 | --- | --- | --- |

**Table 3. Parameters extracted from the seismic catalogue for each region used to estimate the COV coefficient for Table 1, regions 28, 29, 33 and 40 have been excluded because they do contain no faults.**

| Region | $M_{Min}$ | $M_{MaxC}$ | β-value region | Record | COV |
|--------|-----------|------------|----------------|--------|-----|
| 30 | 4.0 | 4.6 | 1.800 | 7 | --- |
| 31 | 4.0 | 5.7 | 1.980 | 66 | 0.4 |
| 34 | 4.0 | 5.5 | 2.345 | 35 | 0.6 |
| 35 | 4.0 | 5.5 | 2.242 | 117 | 0.3 |
| 36 | 4.0 | 5.4 | 2.400 | 25 | 0.7 |
| 37 | 4.0 | 5.7 | 1.917 | 83 | 0.3 |
| 38 | 4.0 | 5.4 | 2.240 | 85 | 0.3 |
| 39 | 4.0 | 5.4 | 1.750 | 61 | 0.3 |

**Table 4. Seismic potential distribution of faults and zones. The last column includes the percentage of regional seismic potential assigned to each source within the region.**

| Region | Source | β-valor | Ṅmin | Ṁo (Nm/yr) | |
|--------|--------|---------|------|-----------|------|
| 30 | Σ fault | 1.700 | 0.0078 | 2.76E+20 | 15% |
| | zone | 1.800 | 0.0451 | 1.58E+21 | 85% |
| | Total | | 0.0529 | 1.86E+21 | |
| 31 | Σ fault | 1.950 | 0.0372 | 7.37E+21 | 16% |
| | zone | 1.980 | 0.2017 | 3.97E+22 | 84% |
| | Total | | 0.2389 | 4.70E+22 | |
| 34 | Σ fault | 2.250 | 0.0244 | 2.92E+21 | 11% |
| | zone | 2.345 | 0.1932 | 2.27E+22 | 89% |
| | Total | | 0.2176 | 2.56E+22 | |
| 35 | Σ fault | 2.186 | 0.3474 | 4.33E+22 | 61% |
| | zone | 2.242 | 0.2227 | 2.77E+22 | 39% |
| | Total | | 0.5701 | 7.09E+22 | |
| 36 | Σ fault | 2.330 | 0.0804 | 8.02E+21 | 57% |
| | zone | 2.400 | 0.0603 | 6.08E+21 | 43% |
| | Total | | 0.1407 | 1.41E+22 | |
| 37 | Σ fault | 1.900 | 0.1247 | 2.57E+22 | 28% |
| | zone | 1.917 | 0.3152 | 6.50E+22 | 72% |
| | Total | | 0.4399 | 9.07E+22 | |
| 38 | Σ fault | 2.180 | 0.3269 | 3.55E+22 | 62% |
| | zone | 2.240 | 0.2131 | 2.15E+22 | 38% |
| | Total | | 0.5400 | 5.70E+22 | |
| 39 | Σ fault | 1.820 | 0.1005 | 1.35E+22 | 31% |
| | zone | 1.750 | 0.2077 | 2.96E+22 | 69% |
| | Total | | 0.3082 | 4.30E+22 | |

