# Peer review of "Approach for Combining Fault and Area Sources in Seismic Hazard Assessment: Application in South-Eastern Spain"

_Natural Hazards and Earth System Sciences, 2018_

## Referee Comment (RC1) · Anonymous Referee #1 · 3 Apr 2018

General: The paper is generally well written and presents a new alternative for combining faults and zones in a PSHA evaluation. The language is generally acceptable, but here and there the Spanish "accent" is coming through, and sometimes wrong wording is used. A thorough "language washing" by a native English-speaker is needed before publication.

On Mmax: This review is concerned with using the catalogue only up to Mmax recorded in the catalogue. What about blind faults that are not mapped and may trigger larger magnitudes away from the mapped faults. The authors should be more clear about this situation. This issue is exemplified in Fig. 6 where the Granada and Almeria hazard

is so widely different from the zonation based hazard map. Fig. 6 should also be expanded with 1 Hz hazard difference in which thedifference between methiods could be even higher.

On Mmax: After reading I am still somewhat uncertain how MmaxC and Mmax relate. The authors should make some additional effort in clarifying the different Mmax used for faults and areas. May be use more clear annotations Mmax(fault)= MmaxF; Mmax(zone)=MmaxZ and Mmin correspondingly.

On Mmax: May be I have overlooked, but how is Mmax established quantitatively from the catalogue?

QAFI database not defined/referenced.

Fig. 6 and 8: What is the Mmax used in the reference model? Due to the big differences it is important to be very clear about the reference computation. The implications in an application is significant.

―――――――――――――――――

---

## Referee Comment (RC2) · Anonymous Referee #2 · 8 Apr 2018

\#

\# General comments

\#

This paper is interesting and covers an important topic of interest to NHESS readers.

However, there are several significant weaknesses in the paper, which I estimate will require \*major revisions\* to address.

Although the major revisions may involve considerable work from the authors, I do think it should be possible to revise the paper to meet the standards of NHESS publication.

[Figure]

Before going into detail below, I note some 'overall' problematic aspects of the current draft:

- Poor referencing. The paper contains no references in the Introduction, despite discussing other work. Other areas also need more references (e.g. often non-obvious 'facts' are stated, without information on where they come from). I note a number of examples below, but not all – in general, the authors need to be more rigorous about justifying statements with references.

- There is insufficient explanation of the earthquake catalogue analysis methods (i.e. the discussion on page 3 in the paragraph around line 20 needs to be expanded).

- A number of statements in the paper are unclear or lacking in rigor (details below).

- I think there is a (fixable) logical flaw in the method. The authors are assuming (equations 8 and 9)

1: region_seismicity = zone_seismicity + fault_seismicity;

and furthermore, at various points in the analysis they assume (equations 10 and 4):

2: all of these seismicity sources can individually be represented with GR curves;

The combination of (1) and (2) would be fine *if* all the seismicity sources have the same 'b-value', but the authors suggest different 'b-values' on fault vs region/zone. Mathematically, I don't think (1) holds under this assumption (i.e. you can't sum 2 GR curves with different b values and expect to get another GR curve). I have suggested a possible fix below, which looks like it may be reasonable based on the data in the paper – but I can't be sure. Anyway, the authors should address this somehow.

On the other hand, I expect these issues can be addressed. Furthermore I note there are good points to the paper: it treats an important topic, provides a good application of the methods (which I found much easier to read than the methods description itself), and includes a reasonable attempt to quantify the uncertainties (Section 2.4).

#

**Specific Comments**

#

- The Introduction does not contain any citations of other work. However, there is repeated mention of 'other studies' {e.g. p1 Line 25; p1 Line 27; p2 Line 3; p2, Line 22; p2, Line28 ... and many other instances, I will not list them all}. Throughout this section, the authors need to provide references to justify their descriptions of other work (even when they refer to general issues that are not specific to a particular study).

- P 3, Line 15: ** Magnitudes values above MMaxC present recurrence periods higher than the catalog OP or constitute a sample an insufficient number of records to apply statistics. ** – This statement needs to be rephrased to correct the grammar. More importantly I think it is factually incorrect.

For illustration, suppose we have a catalogue of 100 years duration, and this catalogue contains zero events with Mw>=X. Then I claim that we **can apply statistics** to place bounds on the rate of events with Mw>=X, and furthermore, that **the return period of such events MIGHT be less than the observational period**. Both of these points contradict the author's statement. For example, with zero events in 100 years, an exact Poisson test of the true rate of events with Mw>=x (e.g. poisson.test(x=0,T=100) in the R software) gives a 95% confidence interval on the true rate of [0, 0.037] events/year. So the return period is probably greater than around 1/0.037=27 years. This makes sense – if the return period were very small then it is likely an event would occur in the data. Furthermore, it is obviously possible that the return period is < 100 years (we don't get a '100 year event' in every hundred year observational period).

I think the authors probably want to say that it is increasingly difficult to constrain rates of rarer events – that's correct - but the authors need to weaken the statement accordingly.

- P 3, lines 17-18: The authors should explain the 'Stepp (1972)' approach to estimating the reference years for different return periods. Although Figure 2 is related to this, I cannot understand the method from the figure.

- P3, Lines 19-20: **Then, it is possible to estimate ...** – the authors should explain how they do this estimation. There could be a range of methods (depending on the extent to which one makes parametric assumptions, like that the data-generating process has a GR distribution).

- P 3, Line 24: ** This way we avoid miscalculation problems ...**. I think this statement is too simplistic, and should be rewritten. In reality, errors in the estimated rates may be quite significant even for events with true return period significantly less than the catalogue duration. There are statistical methods for quantifying such uncertainties in different contexts.

For example, for individual rates one might assume a Poisson process and estimate confidence intervals as I suggested in an earlier comment. Or for a full magnitude-frequency distribution, one might estimate uncertainty in the rates for any magnitude by assuming a GR distribution, and using maximum likelihood with profile likelihood confidence intervals - or using Bayesian techniques, etc. If the authors google-search 'Gutenberg Richter maximum likelihood estimation' they will find many related references.

- P 4, Section 2.2: It's not clear to me whether the authors assume there is only 1 fault per region (this is how the math appears, and consistent with the use of the word 'fault' rather than 'faults' in most places), or if there are multiple faults and GR curves which are summed over (that would seem sensible, but no summation is mentioned), or if there are multiple faults which are all treated with a single GR model (but there is no summation mentioned in equation 3, and the word 'fault' is mostly used, rather than 'faults'). This needs to be made much clearer to the reader, with 'summations over faults' included in the equations as appropriate. Table 4 makes me think it is 'multiple

faults with single GR model', but it is far from clear.

- P 4, line 12: It is unclear how 'v' is estimated (I suppose from other geophysical studies? Or paleo work?). Please provide references. Also, how do you choose the shear modulus? References.

- P 4, line 16: Note that 'm=0' is not the lowest value of m. Maybe just say 'from very low up to the maximum ...'.

- P 4, line 26: Related to the above comment, you might make the lower summation index in the first right-hand-side term be "negative infinity" rather than zero. Also, I think you don't model earthquakes with m < M_min later in the paper? If so, I suggest mentioning that here. Indeed you might re-write this part to avoid mention of events below M_min, perhaps it is just confusing?

- P 4, line 22: I'm not sure if you have defined the $\overline{d}$ variable. Also I don't understand the notation "Mo|" in the numerator (what is the bar? what is Mo? Should it be for faults only?)

- P 5, line 9: Here, I think you should remind the reader that the 'region' parameters are assumed known, based on Equations 1 and 2.

- P 5, line 14: **Note that the b-value of the zone appears in this equation can be equaled to the b-value of the region as both sources present similar seismic nature.** Mathematically I think this is problematic. Question: Is a distribution defined by summing 2 GR distributions with different 'b-values' still a GR distribution? I don't think so. However, that seems to be an implicit assumption of your method (i.e. because the regional seismicity is the sum of 2 different GR distributions, fault and zone, each having different b values). This suggests an internal contradiction in the methods, which should be fixed.

Suggestion for a fix: Consider modifying your method so that b-fault is equal to b-region and b-zone. Then I think the issue would be avoided? Furthermore, from Table 4, that

would look to be an OK approximation, given the statistical uncertainties in Table 1 (???).

- P5, Section 2.4. I like this analysis overall. However, from Table 1 it seems like you are using a single b value in the simulations (?), in contrast to the above-mentioned 'distinct fault/region b values'. The testing here needs to be consistent with the methodology. So if you change the analysis as I suggested above then this section is probably fine – but otherwise, you should treat these 'distinct fault/region b values'

- P7, line 4: Need a reference for the shortening rate.

- P 7, line 14: Reference for the QAFI database?

- P 7, line 16: Not clear to me how you estimate M-max if there are multiple faults per region. Do you assume they all rupture at once?

- P 7, line 29: '.. lacks a COV estimate because the sample of records is not significant ..' – I think you need to re-word this. What do you mean by 'not significant'? It's unclear – the word 'significant' is suggestive of 'statistical significance', but that does not appear to be what you mean. I think you just mean 'there is very little data'. However, I don't see why you can't calculate a COV coefficient (albeit a very uncertain one).

- P 8, line 13: 'This result is characteristic of this method ...' – sounds like there must be other studies that show this, please add references.

- P8, line 20: "These results agree with real observations" – can you cite the study?

#

**Technical corrections**

#

- P1, Line 6: 'estimated' should be 'estimates'

- P2, Line 18: suggest changing 'a part of' to 'some of'

- P2, Line 18: 'must be linked to faults' – is this always true? What about if there are not catalogue events on the faults? Suggest rewording.

- P 2, line 26: 'fixing a Mc value results certainly complicated' - This doesn't make sense, suggest rewording. I'm not 100% sure what you want to say – do you mean 'it is difficult to choose Mc non-arbitrarily'?

- P2, Line 28: ** The approach presented in this paper, as all probabilistic seismic hazard models, face the challenging question of estimating the expected ground motions with the basis of a short period of observations of earthquake occurrences and limited 30 geological data (with significant uncertainty) to construct recurrence models.** This needs to be re-worded to correct the grammar. One suggestion: ** The approach presented in this paper faces the challenging question of how to estimate the expected ground motion exceedance rate, using a short period of earthquake observations and limited geological data (with significant uncertainties). This challenge is faced by all probabilistic seismic hazard models. **

- P 3, L5: **The zone is defined as the source which seismic potential is residual, excluding the seismic potential of faults **. There are grammatical issues here, as stated it does not make sense. I think you mean 'The zone is the same as the region with fault seismicity removed'. However, you may choose to make different edits (considering also repetition in the subsequent sentence).

- P3, Line 10: This sentence needs rewording (grammar). What about: "The seismicity rate of the region is derived from the seismic catalog using events that are contained in the period for which ...."

- P 3, Line 11: You use 'CP' here, but on line 13 you use 'PC'. Please double check for consistency throughout the paper.

- P 3, Line 13: **The complete periods PC(m) (for different magnitude up to a maximum magnitude of completeness value, MMaxC.) are included in the observation period

(OP) of the catalog. ** Is it really correct to say they are 'included'. I suspect you mean to say they are 'inferred using' or are 'less than'?

- P 4, line 1: I suggest you replace the word 'cumulative' with 'total'.

- P 4, line 20: 'establish' instead of 'stablish'

- P 4, line 20 and below. You refer to $\dot{Nmin}$ – should it have a subscript 'fault', considering later notation (Equation 8)?

- P5, line 9: 'An new' should be 'A new'

- P5, line 19: I think this should mention 'faults', e.g. 'by extrapolation of the faults recurrence model'

- P6, line 11: should say 'moment rates for different magnitude values, ...'

- P6, line 12: similar problem as above

- P 7, line 26: 'It is appreciated that ..' – this sounds strange – suggest you change to 'Note that ..'. Also, it's not clear to me why you say this at all, consider making the point clearer.

---

## Referee Comment (RC3) · Anonymous Referee #3 · 10 Apr 2018

The paper it is a valuable and original contribution in the subject of fault source modelling in PSHA. The proposed approach has been used in a SH calculation in Spain with satisfactory results. It would be interesting to check the viability of the model in other parts of the world with faster moving faults.

Nevertheless, there are few issues that, if tackled appropriately, will greatly improve the paper, and I think they should be addressed obligatory before publication in NHESS (please see list below).

Additionally, I attached the manuscript with many comments, suggestions, and corrections, all of them highlighted in yellow and commented using Adobe reader tools.

[Figure]

- Improve the English. Please, for a new submission make sure that the entire article is revised by an English native speaker. The style could also be greatly improved. Avoid the excessive use of "extra comments" in brackets (e.g., page 1, line 9; p.2 lines 17, 30, 31; p.3 line 4, 12;... and so many more...). I attached the manuscript pdf highlighting that, typos, few mistakes and so.

- Citations. In the introduction there is not a single cite about other previous approaches about incorporating faults as sources in PSHA neither globally nor locally. The authors must cite other previous work in the subject (e.g., Youngs and Coppersmith, 1985; Wesnousky, 1986; Anderson and Luco, 1983; Bungum, 2007; ..... and much more recently the different UCERF versions (e.g., Field et al., 2009, 2014), the SHARE project (Woessner et al., 2015),... among others. If there are, cite also other previous work done in your study area.

- Seismic Moment Equation. It is necessary to cite and write the equation used for calculating the seismic moment from magnitude (Mo as function of (Mw), Hanks y Kanamori, 1979, IASPEI, 2005) for a completely comprehension of equation (2) and, later, equation (4). Additionally, you should state before that with the variable "m" you always refer to magnitude in the moment magnitude (Mw) scale.

- Equation 4. I assume this equation is an original contribution from this work? Please, you need to show how do you get to equation 4 from Anderson (1979) integral. Need to explain what is parameter d (in relation to Mo f (Mw).

- Equation 10. Further explanations should be given on how you get to equation 10.

- Figure captions. They are very short. More explanations in the captions are needed to fully understand the figures. Particularly figure 2 (see attached pdf).

- Figure 5. This figure can be greatly improved from what it is now (a dumped screen). A inset locating geographically the studied area is necessary.

- Fault sources. The paper will improve greatly if you further explain how did you select

the faults and what are there characteristics in terms of slip rate, Mmax, kinematics, etc. . .

- Ground Motion Prediction Equation. The GMPE you chose for your calculations was produced to address near fault-source effects. I understand that the point of the paper is to use your methodology for sharing the seismic potential between zones and faults, but because you have chosen this particular GMPE for your calculations you should explore the impact of using it in your hazard results compare to the use of a general GMPE one. I mean, this is important because in your results it is clearly shown that the hazard increases a lot near the faults, as you state p.10 l.3: "The results show an increment of expected accelerations near fault traces (in a factor of 2)." And later: "This increment is achieved at the expense of decreasing expected accelerations in areas located farther away from faults." This statement should be properly discussed in the Discussion section.

- Results. This section should be rewritten. It contains many statements that are better placed in a "Discussion" section. See attached pdf.

- Discussion and Conclusion should be separated sections. The paper will improve greatly if you discuss your results in terms of earthquake rates contribution from faults vs zones, instead of accelerations. This way it would be showed the real (pure) impact of your approach in the hazard, without consideration of the GMPE. Subsequently, you can explore the impact of using a different GMPE in the calculations.

- References list. There are a couple of references listed but missing in the manuscript.

- Table 2. Mmax relates to the Max event from the fault. When there is more than one fault in the region, which value do you state on this table? In the range Mmin-Mmax you show the accumulated moment rate from all the faults in the region? How can be regions with MaxC blank? . . . More information is needed to understand this table properly.
- Table 3. More information is needed in the caption to understand it properly. Information on some regions is missed. The MmaxC values are strikingly low... What happened to the larger values (big historical events)?

Please also note the supplement to this comment:
https://www.nat-hazards-earth-syst-sci-discuss.net/nhess-2018-28/nhess-2018-28-RC3-supplement.pdf
* * *
[Figure]

**Supplement:**

[Figure]

**Approach for combining faults and area sources in seismic hazard assessment: Application in southeastern Spain**

Alicia Rivas-Medina[1,2], Belen Benito[1], Jorge Miguel Gaspar-Escribano[1]

[1] Departamento de Ingeniería Topográfica y Cartografía , Universidad Politécnica de Madrid, Madrid, Spain.

[2] Departamento de Cs. Geodésicas y Geomática, Universidad de Concepción - Campus Los Ángeles, Los Ángeles, Chile.

*Correspondence to*: A. Rivas-Medina (alicrivas@udec.cl)

**Abstract.** This paper presents a methodological approach for seismic hazard assessment that considers a hybrid source model composed by faults as independent entities and zones (containing the residual seismicity). The seismic potential of both types of sources is derived from different data: for the zones, the recurrence model is estimated from the seismic catalog. For fault sources, it is inferred from kinematic parameters derived from paleoseismicty and GNSS measurements.

Distributing the seismic potential associated to each source is a key question when considering hybrid models of zone and faults, which some authors solve by assigning to the fault only the earthquakes that exceed a fixed magnitude value Mc. In the present approach, instead of restricting the magnitudes of each type of source, the distribution of seismic potential is carried out only for magnitudes below the maximum magnitude value completely recorded in the catalog (Mmaxc). This is derived from a completeness analysis and can be lower than the Mmax generated by the faults, taking into account that their the recurrence period can be higher than the observation period of the catalog.

The proposed approach is applied in southern Spain, a region of low-to-moderate seismic where faults move slowly. The results obtained are contrasted with the results of a seismic hazard model using the traditional zone model exclusively. Results show a concentration of expected accelerations around fault traces using the hybrid approach, which is not appreciated in the classic approach using zones exclusively.

**1 Introduction**

Active faults are the main earthquake sources in the crust. However, their incorporation in seismic hazard assessment is not straightforward due to the lack of data required to model them appropriately. This leads to a limited use of faults as independent sources in seismic hazard analyses and to an extended use of seismic zones, which cover a significant portion of the crust, assuming uniform seismic characteristics within each source.

This situation is changing in the last years, as more studies on active tectonics, paleoseismicity and fault deformation rates derived from GNSS measurements (among others) constraining fault parameters (such as rupture plane geometry, predominant sense of slip, slip rates, recurrence periods, etc.) are available.

[Figure]

The consideration of sources as "fault type" instead of zones in seismic hazard studies requires addressing two factors: the 3D geometry of the source and the data needed to characterize its seismic potential. In most practical cases, the characterization of the seismic potential of faults is based on their kinematics. Other approaches, such as extracting the seismic parameters of each single fault from the earthquake catalog is not always viable, especially in areas with slowly

5    moving faults. Additionally, the period considered in the catalogue may be too short in comparison with the recurrence time of the fault to provide unbiased estimated of fault seismic parameters.

In principle, the modeling of all existing active faults as independent entities could be conceived as the most accurate source model for seismic hazard assessment. However, this vision is still quite idealistic. A more realistic view would include only a limited number of active faults (those with the highest seismic activity) as independent sources. Accordingly, small faults

10    generating low magnitude events or slow faults producing rare events cannot be properly characterized. To prevent a possible deficit in the seismic source model of a region, the use of faults as seismic sources may be completed with zones that account for the seismic potential linked to these small/slow faults or just unknown faults that cannot be characterized independently. Hence, we propose to consider hybrid source models, composed by faults and zones: the first ones modelled as independent sources and the second ones encompassing the residual seismicity.

15    The problem of using a model combining zones and faults is establishing the distribution of seismic potential among them, taking into account that they are derived from different data sources. For zones, the recurrence model is calculated from the seismic catalog, and for faults, it is derived from fault geometries and slip rate estimates (based on GNSS-measured deformation rates). The problem is that a part of the events contained in the catalog must be linked to faults and they are already included in the seismic potential of faults derived from slip rate estimates. If all events are assigned to the zone, the

20    events linked to faults would be double-counted, leading to an overestimate of the total seismic potential (related to both faults and zones).

Some authors propose a simple way of distributing the seismic potential based on a uniform magnitude value Mc, and assigning the events with magnitude lower than Mc to the zone and the events with magnitude higher than Mc to the faults. The question is: how is this Mc value determined? Why the fault cannot generate events with magnitude below the Mc value

25    in possible blind faults. In a study region such as southern Spain, with slow faults and maximum magnitudes around 6.5-7.0, fixing a Mc value results certainly complicated. Hence, a new approach to distribute the seismic potential is proposed in this paper.

The approach presented in this paper, as all probabilistic seismic hazard models, face the challenging question of estimating the expected ground motions with the basis of a short period of observations of earthquake occurrences and limited

30    geological data (with significant uncertainty) to construct recurrence models. The purpose of this work is not to solve this challenge, but rather, to propose a model that contains different types of seismic sources (faults and zones) and distributes the seismic potential appropriately, avoiding double-counting and considering periods of completeness.

[Figure]

[Figure]

**2 Source hybrid approach (zones & faults) for hazard estimation**

The hybrid model proposed is composed by fault-type sources and zone-type sources. At the same time, the term *region* is defined as the geometric envelope of both source types. Thus, the region presents the same geometry as the zone and the seismic potential (seismicity rate and seismic moment rate) of both sources (faults and zone). The zone is defined as the

5    source which seismic potential is residual, excluding the seismic potential of faults. Although there is a geometrical equivalence between region and zone, their seismic potential are very different, as the seismic potential of the region equals the seismic potential of the zone plus the seismic potential of the faults enclosed in the region (Fig. 1).

The problem is then distributing the seismic potential of the region between the zone and the faults trying to avoid double-counting. The following considerations are made:

10    - The seismicity rate of the region is derived from the seismic catalog using the records which origin time and magnitude are contained in the period for which the catalog can be considered complete, CP(m) (complete period for magnitude m and higher).

- The complete periods PC(m) (for different magnitude up to a maximum magnitude of completeness value, $M_{MaxC.}$) are included in the observation period (OP) of the catalog.

15    - Magnitudes values above $M_{MaxC}$ present recurrence periods higher than the catalog OP or constitute a sample an insufficient number of records to apply statistics.

The representation of the number of recorded events for different magnitude intervals as a function of time allows the identification of the reference years RY(m) for different magnitude intervals using the Stepp (1972) approach. Consequently, the PC(m) and $M_{MaxC}$ values can be calculated for m<= $M_{MaxC}$. Then, it is possible to estimate seismicity rates and seismic

20    moment in each magnitude interval and cumulative rates in the interval [$M_{Min}$, $M_{MaxC}$], where $M_{Min}$ is the minimum magnitude value used to compute seismic hazard. This is illustrated with an example in Fig. 2, with [$M_{Min}$, $M_{MaxC}$] = [4.0, 6.9].

Although faults are capable of generating earthquakes with magnitude m > $M_{MaxC}$, the distribution of seismic potential is carried out in the complete period [$M_{Min}$, $M_{MaxC}$]. This way we avoid miscalculating problems related to recurrence periods

25    larger than OP and the eventual occurrence of events with magnitudes higher than the maximum expected magnitudes in the fault ($M_{Max}$) that are not contained in the seismic catalog. The computation of the seismic potential of the fault in the interval [$M_{MaxC}$, $M_{Max}$] (where $M_{Max}$ is the maximum magnitude value of events that can be generated in a fault) is constrained with other geological criteria (see below).

[Figure]

The seismic potential is represented by the cumulative rate of earthquakes ($\dot{N}min$) and the cumulative rate of seismic moment ($\dot{M}o$), for the magnitude range [$M_{Min}$, $M_{MaxC}$], in the complete period PC(m). Details on how to determine $\dot{N}min$ and $\dot{M}o$ for the entire region, the corresponding zone and faults are explained in the following.

**2.1 Seismic potential of the region**

5  The $\dot{N}min$ and $\dot{M}o$ values representing the seismic potential of the region are derived from the seismic catalog in the complete periods CP(m), for the magnitude interval [$M_{Min}$, $M_{MaxC}$].

$$\dot{N}min_{region}\Big|_{M_{Min}}^{M_{MaxC}} = \sum_{M_{Min}}^{M_{MaxC}} \dot{n}(m) \tag{1}$$

$$\dot{M}o_{region}\Big|_{M_{Min}}^{M_{MaxC}} = \sum_{M_{Min}}^{M_{MaxC}} \dot{n}(m) \cdot Mo(m) \tag{2}$$

with $\dot{n}(m)$ the annual rate of events with magnitude (m) recorded in PC(m) and Mo(m) the seismic moment for magnitude

10  m.

**2.2 Seismic potential of faults**

The cumulative moment rate of the fault is estimated from the slip rate ($\upsilon$), the rigidity modulus ($\mu$) and the area of the fault plane (A), using the expression of Brune (1968), assuming that the fault plane is accumulating energy evenly:

$$\dot{M}o_{fault} = \upsilon \cdot \mu \cdot A \tag{3}$$

15  This moment rate represents the average seismic moment accumulated in the fault per year that will be released by earthquakes of different magnitudes (from very low, close to $m=0$ up to the maximum magnitude of the fault, $M_{Max}$). The value of $M_{Max}$ can be evaluated from a geometrical element of the fault plane using empirical relationships proposed in the literature, as Wells and Coppersmith (1994), Stirling et al. (2002) or Leonard (2010) (among others).

Assigning a recurrence model to the fault and assuming an initial b-value (for example, b=1) that will be adjusted iteratively,

20  it is possible to stablish a relation between the seismicity rare and the moment rate by solving the integral proposed by Anderson (1979). Using a modified Gutenberg –Richter recurrence model, $\dot{N}min$ is calculated as:

$$\dot{N}min = \frac{\dot{M}o \cdot (d-\beta) \cdot \left(e^{-\beta(M_{Min})} - e^{-\beta(M_{Max})}\right)}{\beta \cdot \left[e^{-\beta M_{Max}} Mo(M_{Max}) - e^{-\beta M_{min}} Mo(M_{Min})\right]} \tag{4}$$

The total seismic moment rate of each fault ($\dot{M}o_{fault}$) and seismicity rate ($\dot{N}min$) can be formulated as the sum of the seismic moment rate released in different magnitude intervals, it follows:

25  $$\dot{N}min = \dot{N}min\Big|_{\sim 0}^{M_{Min}} + \dot{N}min\Big|_{M_{Min}}^{M_{MaxC}} + \dot{N}min\Big|_{M_{MaxC}}^{M_{Max}} \tag{5}$$

$$\dot{M}o_{fault} = \sum_{\sim 0}^{M_{Min}} \dot{n}(m) \cdot Mo(m) + \sum_{M_{Min}}^{M_{MaxC}} \dot{n}(m) \cdot Mo(m) + \sum_{M_{MaxC}}^{M_{Max}} \dot{n}(m) \cdot Mo(m) \tag{6}$$

Implementing a recurrence model is possible to derive the seismicity rate and the moment rate in the interval [$M_{Min}$, $M_{MaxC}$] (see example in Fig. 3 with [$M_{Min}$, $M_{MaxC}$] = [4.0, 6.9]).

[Figure]

[Figure]

**2.3 Seismic potential of the zone**

The parameters representing the zone are initially unknown. The can be calculated for the interval [$M_{Min}$, $M_{MaxC}$] considering that:

$$Seismic\ Potential_{zone} = Seismic\ Potential_{region} - Seismic\ Potential_{faults} \qquad (7)$$

Or specifically:

$$\dot{N}min_{zone}|_{M_{Min}}^{M_{MaxC}} = \dot{N}min_{region}|_{M_{Min}}^{M_{MaxC}} - \sum \dot{N}min_{fault}|_{M_{Min}}^{M_{MaxC}} \qquad (8)$$

$$\dot{M}o_{zone}|_{M_{Min}}^{M_{MaxC}} = \dot{M}o_{region}|_{M_{Min}}^{M_{MaxC}} - \sum \dot{M}o_{fault}|_{M_{Min}}^{M_{MaxC}} \qquad (9)$$

In principle, there are 2 equations with two unknowns related to the zone: $\dot{N}min_{zone}|_{M_{Min}}^{M_{MaxC}}$ and $\dot{M}o_{zone}|_{M_{Min}}^{M_{MaxC}}$.

Regarding the fault, $\dot{N}min_{fault}|_{M_{Min}}^{M_{MaxC}}$ and $\dot{M}o_{fault}|_{M_{Min}}^{M_{MaxC}}$ are derived using an initial (not definitive) b-value. An new additional equation is obtained relating $\dot{N}min_{zone}|_{M_{Min}}^{M_{MaxC}}$ and $\dot{M}o_{zone}|_{M_{Min}}^{M_{MaxC}}$ using Eq. (4) for the interval [$M_{Min}$, $M_{MaxC}$] in the zone, giving:

$$\dot{N}min_{zone}|_{M_{Min}}^{M_{MaxC}} = \frac{\dot{M}o_{zone}|_{M_{Min}}^{M_{MaxC}} \cdot (\bar{d}-\beta_{zone}) \cdot \left(e^{-\beta_{zone}(M_{Min})} - e^{-\beta_{zone}(M_{MaxC})}\right)}{\beta_{zone} \cdot \left[e^{-\beta_{zone}M_{MaxC}}Mo(M_{MaxC}) - e^{-\beta_{zone}M_{Min}}Mo(M_{Min})\right]} \qquad (10)$$

Note that the b-value of the zone appears in this equation can be equaled to the b-value of the region as both sources present similar seismic nature.

With this third equation, it is possible to solve the system and obtaining a new b value for the fault (second iteration) that balances the three equations. This gives the distribution of seismic potential between the zone and the faults in the interval [$M_{Min}$, $M_{MaxC}$].

Considering that the fault may generate events with magnitudes larger than $M_{MaxC}$, the corresponding distribution of seismic potential in the interval ($M_{MaxC}$, $M_{Max}$] is calculated by extrapolation of the recurrence model with the last b-value adjusted (Fig. 4).

Regarding the $M_{Max}$ value expected for the zone, this can be equalled to $M_{MaxC}$ or extended to a higher magnitude value if it is accepted that bigger events can occur in other unidentified sources (such as blind faults). unidentified sources (such as blind faults).

**2.4 Analysis of uncertainty**

The proposed approach strongly relies on the computation of the seismicity (earthquake rates and moment rate) within the magnitude interval [$M_{Min}$, $M_{MaxC}$] of the seismic catalog that contains the complete record of events occurred in the entire region.

[Figure]

In order to capture the variability of seismic moment rates calculated from the earthquake catalog a sensitivity analysis of three key factors is conducted. These factors are: 1) the number of records used to compute moment rates, 2) the magnitude range covered by the complete catalog and 3) the proportion of earthquakes of different magnitude (b-value).

Synthetic catalogs derived from GR-modified recurrence models are generated for this purpose. Earthquake rates are
5   computed using different amounts of events, magnitude intervals and b-values that could be representative from areas of low-to-moderate seismic activity.

The procedure comprises five steps:

- Generation of 2000 synthetic catalogs for different combinations of earthquake rates, magnitude intervals and b-values.

10   - Calculation of earthquake rates for different magnitude values, for each synthetic catalog (Eq. (4)).

- Calculation of moment rate different magnitude values, for each synthetic catalog.

- Sum of moment rate different magnitude values, to obtain the cumulative moment rate for each synthetic catalog (Eq. (5)).

- Computation of the mean and the standard deviation of the distribution of calculated seismic moment rates.

15   Table 1 shows the coefficient of variation (COV = sigma/mean) associated to each combination (number of events, magnitude interval, b-value). As can be seen, the greater the number of records in the sample and lower magnitude range, the lower the uncertainty associated with the rate of seismic moment calculated. The b-value presents a different trend, recording the biggest variability for b-values between 1.0 and 1.5. This table is useful to estimate the uncertainty on the seismic moment rate calculated from the seismic catalogue, as a function of the number of earthquakes, magnitude interval and b-
20   value.

Is important also to consider the uncertainty associated with the slip rate and the area of the fault, as they propagate into the distribution of seismic moment rate of the fault proportionally to the deviation of the area or slip rate value. The uncertainty of the slip rate value is more relevant for low slip rate values than for large slip rate values (a similar trend can be deduced for low and high area values). For instance, a deviation of ±1 mm/year in a slip rate of ±2 mm/year represents an uncertainty
25   of 50%, leading to a COV value of 0.5 in the moment rate of the fault. However, the same deviation (± 1 mm/year) for a fault with slip rate of ±10 mm/year, represents an uncertainty of 10%, leading to a COV coefficient moment rate of the fault of only 0.1.

[Figure]

**3 Application of the approach in southeast Spain**

The approach described above is applied in southeastern Spain, the most seismically active area of the country. The tectonic deformation and seismicity is related with the convergence between the African and Eurasian plates, with approximate shortening rate of 5 mm/year in NNW-SSE direction. Crustal deformation is accumulated over a broad area where the

5   seismicity is diffuse. The assignation of earthquakes to specific faults is not an easy task: whereas earthquakes can be clearly associated to a rupture (such as the 2011 M 5.2 Lorca event generated in the Alhama de Murcia fault system), other events occurred in areas with unknown active faults (as the 2007, Mw 5.1 Pedro Muñoz and 2015, Mw 4.7 Ossa de Montiel earthquakes, both located in Central Spain).

These seismotectonic characteristics makes southeastern Spain a suitable scenario for applying the hybrid approach

10   combining faults and zones proposed in this work. The recently available data on fault geometries and activity parameters in Spain and the seismic catalog that contain records dating back to 1048 are used to constrain the source model parameters.

**3.1 Sources input data**

The seismogenic source model considered for SE Spain is composed by 12 regions that contain a total of 95 faults. Active fault data are taken from the QAFI database (v2.0), which includes information about faults segmentation, geometry and slip

15   rate (see Fig. 5). The maximum expected earthquake magnitude generated in each fault is derived from the fault geometry, applying the empirical relationship of Stirling (2002). Moment rates accumulated in the faults are estimated using the fault plane area and the slip rate value by the formula of Brune (1968). The zones model of García-Mayordomo et al. (2010) is used to obtain the geometries of the 12 regions (and thus of the zones) that account for the seismicity that cannot be ascribed to faults (see Fig. 5). Not all the regions considered contain faults sources, i. e., regions 28, 29, 33 and 40. In these cases, the

20   seismic potential of the corresponding region is assigned to the zones.

The seismic moment released in the region is estimated from the seismic catalog recently homogenized (IGN-UPM Working Group, 2013; Cabañas et al., 2015). It contains 1496 earthquakes, with magnitudes ranging from 4.0 to 6.6. The uncertainty assessment of the catalog is explained in Gaspar-Escribano et al. (2015). A $M_{MaxC}$ is estimated in 5.9 for SE Spain (although not all the *regions* reach this maximum magnitude value).

25   Table 2 shows the seismic potential for each region, calculated in the magnitude intervals $[M_{Min}, M_{MaxC}]$ and $[M_{Min}, M_{Max}]$. It is appreciated that the seismic potential in the first interval up to $M_{MaxC}$, constitutes at least a 60% of the seismic potential in the second interval, up to $M_{Max.}$

Subsequently, a recurrence model (GR-mod) is assigned to all regions, obtaining the corresponding b-values and COV coefficients (see Table 3). Note that zone 30 lacks a COV estimate because the sample of records is not significant and the

30   $[M_{Min}, M_{MaxC}]$ interval is very narrow, resulting in an increased uncertainty in the hazard estimates in this region. A GR-mod recurrence model is also assigned to the faults. Finally, the distribution of seismic moment among all seismic sources is carried out (Table 4). As it can be observed, the seismic moment rate associated to the zone has a strong influence on the

estimated seismic hazard of the region. This is due to the reduced amount of known active faults than can be modelled as independent sources, a common situation in areas with low and moderate seismic activity. However, it is noteworthy that the seismic potential of regions 35, 36 and 38 is dominated by the seismic potential of faults.

The seismic hazard calculation is done with the software CRISIS2012 (Ordaz et al., 2013), considering the strong motion
5   equation of Campbell (2013), which allows including the fault geometry and the style of faulting. The ground motion parameters predicted include peak ground acceleration (PGA) and 15 spectral accelerations in the period range (0.05 - 10 seconds), all obtained in hard soil ($Vs_{30}$=760 m/s) conditions.

**3.2 Results**

Seismic hazard results obtained with the proposed Hybrid Model (HM) and with the Classical Method based in zone (CM)
10   are shown in Fig. 6a and 6b, respectively. Only the geometry of the zone model differs from both analyses, remaining the ground motion prediction equations (GMPEs) and the other calculation parameters in all the estimates.

The seismic hazard map resulting from the CM approach is rather homogeneous: there are no big differences in the spatial distribution of expected acceleration values. This result is characteristic of this method because the seismic sources are very big. PGA estimates for return period of 475 years using the zone approach (CM) reach maximum values in Granada,
15   Almeria and the Murcia region, around 0.20 g. The minimum PGA values are obtained in Jaen, with values as low as 0.06 g.

Fig. 6a shows the seismic hazard map resulting from the application our approach (HM). It can be seen that the largest accelerations are estimated around the Carboneras fault and the fault set of Granada, (0.38 g), followed by the Alhama de Murcia and La Viña faults systems (0.30g) and, to a minor extent, by the Venta de Zafarraya, Carrascoy, Bajo Segura, Baza, Mijas and Cartama fault systems.

20   The seismic hazard map obtained through the HM displays more spatial variability, showing maximum values along fault sources that decrease strongly away from faults. This trend reflects a "source proximity effect", implying higher acceleration values in the surface projection of the fault rupture plane that rapidly decrease away from the fault (by one half at a distance of about 15 km). These results agree with the real observations in recent earthquakes, including Lorca 2011.

The differences between the expected maximum acceleration obtained with the two methods, CM and HM, for return
25   periods of 475 and 4975 years appear in Fig. 7a and 7b, respectively. The tendency presented in both maps is very similar for the two return periods.

A different case is found in region 30 (Case Lietor fault), a very complex region with small seismic activity and large faults with low slip rates. Here the HM gives higher seismic hazard than the CM only for long return periods. For this region, the magnitude range [$M_{Min}$, $M_{MaxC}$] is very small and it is necessary to extrapolate the model very much, giving the high
30   uncertainty shown in table 3. However, the results reflect that, for longer periods, these slow faults play a relevant role in the seismic hazard of the region. The hazard curves associated with the two models (HM and CM) in this region are shown in Fig. 8. The HM hazard curve reflects a substantial increase in hazard for long return periods.

[Figure]

[Figure]

To clarify how faults are conditioning the final seismic hazard in our model, the seismic hazard curves showing partial contribution of different sources in Murcia, Almería and Granada are shown in Fig. 9 for PGA and SA (1.0s). For each city, black lines show the total seismic hazard curve and color lines the seismic hazard curve associated with different sources (zone and faults) of each city.

5    In Murcia, seismic hazard for short return periods corresponds to several sources involved (zone and faults), but for return periods exceeding 475 years (probability of exceedance of 0.1 and lower in 50 years) the seismic hazard is dominated by the Carrascoy fault. This effect is very similar for PGA and SA (1.0 s).

In Almeria, only two sources contribute to the seismic hazard significantly, zone 38 and the Carboneras fault. In PGA both sources combine equally to give the seismic hazard of the city, but for SA (0.1 s) the Carboneras fault predominates,
10   especially for return periods of more than 475 years.

In Granada, there are many sources contributing to the seismic hazard of the city. This is because there are many known faults in its vicinity. Seismic hazard is controlled by zone 35 for PGA and SA (1.0 s) and shorter return periods. This tendency changes return periods greater than 975 years.

**4 Discussion and conclusions**

15   The estimation of expected ground motions for a future time span is the purpose of a seismic hazard assessment. Such assessment is specially challenging in areas with low seismic activity and faults with long recurrence periods. In these areas, the seismic record contains many information gaps, including the estimation of earthquake rates, seismic recurrences and maximum magnitude values. Additionally, the identification and characterization of seismogenic, active faults are not evident.

20   This paper proposes an approach to prepare a hybrid source model for use in seismic hazard assessment, especially suitable for low seismicity areas. For a given region, the model includes the faults that can be adequately characterized for hazard applications as independent seismic sources. The remaining seismicity, related to other unidentified or unconstrained faults, is taken into account trough seismic zones where the seismicity is uniformly distributed. Both types of seismic sources are modelled with their corresponding recurrence models and maximum expected magnitudes, which are constrained with
25   different kind of data. Fault geometries and slip rates are used to assess the seismic potential of faults. Earthquake rates derived from the seismic catalogue and magnitudes complemented with geological data are considered to constraint the seismic potential of the zone. To prevent double counting, a strong point of this approach is that it assures a distribution of seismic potential between faults and zones within the complete period of the catalog for different magnitude ranges. This distribution of seismic potential depends on the selection of recurrence models to represent fault and zone activities. This is
30   one of the controlling points that have to be carefully addressed considering the specific information available for the study area.

[Figure]

The application of the hybrid model in SE Spain takes advantage of the available (still incomplete) fault data and the relatively long catalog (although lacking high magnitude events). The GR-mod model is used for faults and zones and provides reasonable results in the SE Spain. This approach is particularly suitable for areas of moderate seismicity such as SE Spain, where we focused the application. The results show an increment of expected accelerations near fault traces (in a

5    factor of 2). This is consistent with observations of very high ground motions in the epicentral areas of recent earthquakes, such as the 2009 L'Aquila and 2011 Lorca events. This increment is achieved at the expense of decreasing expected accelerations in areas located farther away from faults. Therefore, this approach gives a more important role to active faults than the classic seismic hazard assessment based on the zone model exclusively. Thus, it can be useful for applications in which near fault effects need to be highlighted, such as urban seismic risk studies for cities located atop active fault planes.

10   The performance of the HM needs to be proven in areas with higher seismic activity, where the budget of observed seismic potential available for distribution between faults and zones can be consumed by the very active faults. In this case it is necessary to analyze different recurrence models, to properly assess catalog completeness and to establish the MmaxC value. This important task will be the focus of future studies.

**Acknowledgements**

15   We thank Dr. M. Ordaz for his time and support during a research stay carried out by ARM at the Instituto de Ingeniería, UNAM. ARM benefited from a pre-doctoral grant funded by the Universidad Politécnica de Madrid.

**References**

Anderson, J. G. (1979). Estimating the seismicity from geological structure for seismic-risk studies. *Bulletin of the Seismological Society of America*, 69(1), 135-158. DOI: 10.1016/0148-9062(79)90309-7

20   Brune J.N. (1968). Seismic moment, seismicity, and rate of slip along major fault zones. *Journal of Geophysical Research*. 73: 777–784. DOI: 10.1029/jb073i002p00777

Cabanas L., Carreno E., Izquierdo A., Martínez J.M., Capote R., Benito B., Gaspar J., Rivas A., García J., Pérez R., Rodríguez M.A. & Murphy P. *Informe del sismo de Lorca del 11 de mayo de 2011. Instituto Geográfico Nacional.* Available: http://www.ign.es/ign/resources/sismologia/Lorca.pdf (in Spanish).

25   Cabañas, L., Rivas-Medina, A., Martínez-Solares, J. M., Gaspar-Escribano, J. M., Benito, B., Antón, R., & Ruiz-Barajas, S. (2015). Relationships between Mw and Other Earthquake Size Parameters in the Spanish IGN Seismic Catalog. *Pure and Applied Geophysics*, 172(9), 2397-2410. DOI: 10.1007/s00024-014-1025-2

Campbell, K. W., & Bozorgnia, Y. (2014). NGA-West2 ground motion model for the average horizontal components of PGA, PGV, and 5% damped linear acceleration response spectra. *Earthquake Spectra*, 30(3), 1087-1115. DOI:
30   10.1193/062913eqs175m

[Figure]

[Figure]

García-Mayordomo, J., Insúa Arévalo, J. M., Martínez Díaz, J. J., Jiménez Díaz, A., Martín Banda, R., Martín Alfageme, S., ...& Masana, E. (2012). The Quaternary Active Faults Database of Iberia (QAFI v.2.0). *Journal of iberiangeology,* 38(1), 285-302. DOI: 10.5209/rev_jige.2012.v38.n1.39219.

García-Mayordomo, J., Insua-Arévalo, J. M., Martínez-Díaz, J. J., Perea, H., Álvarez-Gómez, J. A., Martín-González, F., ...& Giner-Robles, J. (2010). Modelo integral de zonas sismogénicas de España, in *Contribución de la Geología al Análisis de la Peligrosidad Sísmica*, Insua-Arévalo y Martín-González (Eds), Sigüenza (Guadalajara, España), 193–196 (in Spanish).

Gaspar-Escribano, J. M., Rivas-Medina, A., Parra, H., Cabañas, L., Benito, B., Barajas, S. R., & Solares, J. M. (2015). Uncertainty assessment for the seismic hazard map of Spain. *Engineering Geology*, 199, 62-73. DOI: 10.1016/j.enggeo.2015.10.001.

IGN-UPM working group. (2013). Actualización de mapas de peligrosidad sísmica de España 2012. *Centro Nacional de Información Geográfica*, (267 pp.).

Leonard. M., (2010). Earthquake Fault Scaling: Self-Consistent Relating of Rupture Length, Width, Average Displacement, and Moment Release. *Seismological Society of America*, 100 (5A), 1971 – 1988. DOI: 10.1785/0120120249.

Ordaz M., Martinelli F., D'Amico V. and Meletti C. (2013).CRISIS2008: A Flexible Tool to Perform Probabilistic Seismic Hazard Assessment. *Seismological Research Letters*, 84-3: 495-504. DOI: 10.1785/0220120067.

Stepp J.C. (1972). *Analysis of completeness of the earthquake sample in the Puget Sound area and its effects on statistical estimates of earthquake hazard*. Proceedings of First Int. Conference on Microzonazion. 2: 897-910.

Stirling, M., Rhoades, D., & Berryman, K. (2002). Comparison of earthquake scaling relations derived from data of the instrumental and preinstrumental era. *Bulletin of the Seismological Society of America*, *92*(2), 812-830. DOI: 10.1785/0120000221

Wells, D. L., & Coppersmith, K. J. (1994). New empirical relationships among magnitude, rupture length, rupture width, rupture area, and surface displacement. *Bulletin of the seismological Society of America*, 84(4), 974-1002.

30

[Figure]

**Figure 1.** Diagram with the distribution of the seismic potential of a region different sources.

[Figure]

[Figure]

[Figure]

**Figure 2.** Graphs with the completeness analyses of the seismic catalog to identify the completeness period for magnitude PC(m) an MMaxC value.

[Figure]

[Figure]

[Figure]

**Figure 3 Graphs with the identification of seismicity rate and seismic moment rate of a fault, using a modified Gutenberg-Richter recurrence model.**

[Figure]

[Figure]

[Figure]

**Figure 4. Graph with the extrapolation of the recurrence model of the fault up to the maximum expected magnitude value, as deduced form geological criteria.**

[Figure]

[Figure]

**Figure 5. 3D** view of the seismic sources considered for hazard calculation, including faults and zones.

[Figure]

[Figure]

[Figure]

[Figure]

[Figure]

**Figure 6.** Hazard map for return period of 475 years derived with (a) the proposed hybrid approach and (b) the classical zone methodology. Note the fault proximity effects in (a) for these faults: (1) Carboneras fault, (2) Alhama de Mucia fault, (3) Carrascoy fault, (4) Bajo Segura Fault, (5) Mijas and Cartama faults, (6) Zafarraya fault and (7) Baza fault.

[Figure]

[Figure]

**Figure 7. Seismic Hazard Results Comparison of the two models (HM / CM) for return periods of (a) 475 and (b) 4975 years.**

[Figure]

[Figure]

[Figure]

**Figure 8. Seismic hazard curve (with HM and CM) is a site close to the Lietor fault.**

[Figure]

[Figure]

[Figure]

**Figure 9. Seismic hazard curve (with HM) of Murcia, Almería and Granada considering all the seismic sources involved. The black lines show the total seismic hazard curve and the colour lines the seismic hazard curve associated with different sources (zone and faults).**

[Figure]

**Table 1.** COV coefficient associated seismic moment rate obtained by synthetic catalogs.

| records | M(4.0 - 5.0) | | | | M(4.0 - 6.0) | | | | M(4.0 - 7.0) | | | |
|---|---|---|---|---|---|---|---|---|---|---|---|---|
| | b | | | | b | | | | b | | | |
| | 0.5 | 1.0 | 1.5 | 2.0 | 0.5 | 1.0 | 1.5 | 2.0 | 0.5 | 1.0 | 1.5 | 2.0 |
| 200 | 0.00 | 0.09 | 0.10 | 0.09 | 0.19 | 0.30 | 0.39 | 0.31 | 0.36 | 1.00 | 1.17 | 0.43 |
| 180 | 0.09 | 0.10 | 0.10 | 0.09 | 0.20 | 0.29 | 0.33 | 0.28 | 0.35 | 0.87 | 1.21 | 0.42 |
| 140 | 0.10 | 0.11 | 0.11 | 0.10 | 0.21 | 0.31 | 0.33 | 0.25 | 0.39 | 0.92 | 1.17 | 0.31 |
| 120 | 0.11 | 0.13 | 0.13 | 0.13 | 0.25 | 0.40 | 0.47 | 0.34 | 0.49 | 1.12 | 0.87 | 0.36 |
| 100 | 0.11 | 0.13 | 0.13 | 0.12 | 0.25 | 0.39 | 0.50 | 0.36 | 0.48 | 1.16 | 1.15 | 0.43 |
| 60 | 0.16 | 0.15 | 0.17 | 0.15 | 0.32 | 0.47 | 0.57 | 0.40 | 0.58 | 1.37 | 1.12 | 0.44 |
| 40 | 0.18 | 0.20 | 0.21 | 0.21 | 0.39 | 0.64 | 0.79 | 0.59 | 0.71 | 1.73 | 1.40 | 0.62 |
| 20 | 0.27 | 0.31 | 0.33 | 0.32 | 0.61 | 0.91 | 1.18 | 0.79 | 1.08 | 2.60 | 2.21 | 0.91 |

[Figure]

[Figure]

**Table 2. Seismic rate and seismic moment rate recorded in the different regions for two magnitude intervals ($M_{Min}$ - $M_{Max}$) and ($M_{Min}$ - $M_{MaxC}$). It is included the ratio of seismic moment rate of two intervals.**

| Region | $M_{Min}$ | $M_{Max}$ | $M_{MaxC}$ | $M_{Min}$ - $M_{Max}$ | | $M_{Min}$ - $M_{MaxC}$ | | % $\dot{M}o$ recorded in complete periods |
|---|---|---|---|---|---|---|---|---|
| | | | | $\dot{N}(4.0)$ | $\dot{M}o$ (Nm/year) | $\dot{N}(4.0)$ | $\dot{M}o$ (Nm/year) | |
| 28 | 4.0 | 5.4 | --- | 0.211 | 1.90E+22 | --- | --- | --- |
| 29 | 4.0 | 6.2 | --- | 0.176 | 7.90E+22 | --- | --- | --- |
| 30 | 4.0 | 4.6 | 4.6 | 0.053 | 1.86E+21 | 0.053 | 1.86E+21 | 100% |
| 31 | 4.0 | 6.5 | 5.7 | 0.241 | 7.88E+22 | 0.239 | 4.70E+22 | 60% |
| 33 | 4.0 | 5.4 | --- | 0.082 | 1.41E+22 | --- | --- | --- |
| 34 | 4.0 | 6.3 | 5.5 | 0.219 | 3.28E+22 | 0.218 | 2.56E+22 | 78% |
| 35 | 4.0 | 6.5 | 5.5 | 0.574 | 8.73E+22 | 0.570 | 7.09E+22 | 81% |
| 36 | 4.0 | 6.2 | 5.4 | 0.142 | 2.21E+22 | 0.141 | 1.41E+22 | 64% |
| 37 | 4.0 | 6.0 | 5.7 | 0.442 | 1.01E+23 | 0.440 | 9.07E+22 | 90% |
| 38 | 4.0 | 6.5 | 5.4 | 0.527 | 6.75E+22 | 0.525 | 5.70E+22 | 84% |
| 39 | 4.0 | 6.6 | 5.4 | 0.313 | 6.00E+22 | 0.308 | 4.30E+22 | 72% |
| 40 | 4.0 | 6.0 | --- | 0.135 | 4.06E+22 | --- | --- | --- |

[Figure]

[Figure]

**Table 3.** **b-value** estimate in the different regions and assigned to the zones. The coefficient of variation is included.

| Region | $M_{Min}$ | $M_{MaxC}$ | β- value region | Number record | COV coefficient |
|--------|-----------|------------|-----------------|---------------|-----------------|
| 30 | 4.0 | 4.6 | 1.800 | 7 | --- |
| 31 | 4.0 | 5.7 | 1.980 | 66 | 0.4 |
| 34 | 4.0 | 5.5 | 2.345 | 35 | 0.6 |
| 35 | 4.0 | 5.5 | 2.242 | 117 | 0.3 |
| 36 | 4.0 | 5.4 | 2.400 | 25 | 0.7 |
| 37 | 4.0 | 5.7 | 1.917 | 83 | 0.3 |
| 38 | 4.0 | 5.4 | 2.240 | 85 | 0.3 |
| 39 | 4.0 | 5.4 | 1.750 | 61 | 0.3 |

[Figure]

**Table 4. Seismic potential distribution of faults and zones.**

| Region | Source | β-valor | Ṅmin | Ṁo (Nm/yr) | |
|--------|--------|---------|------|-----------|-----|
| 30 | Σ fault | 1.700 | 0.0078 | 2.76E+20 | 15% |
| | zone | 1.800 | 0.0451 | 1.58E+21 | 85% |
| | Total | | 0.0529 | 1.86E+21 | |
| 31 | Σ fault | 1.950 | 0.0372 | 7.37E+21 | 16% |
| | zone | 1.980 | 0.2017 | 3.97E+22 | 84% |
| | Total | | 0.2389 | 4.70E+22 | |
| 34 | Σ fault | 2.250 | 0.0244 | 2.92E+21 | 11% |
| | zone | 2.345 | 0.1932 | 2.27E+22 | 89% |
| | Total | | 0.2176 | 2.56E+22 | |
| 35 | Σ fault | 2.186 | 0.3474 | 4.33E+22 | 61% |
| | zone | 2.242 | 0.2227 | 2.77E+22 | 39% |
| | Total | | 0.5701 | 7.09E+22 | |
| 36 | Σ fault | 2.330 | 0.0804 | 8.02E+21 | 57% |
| | zone | 2.400 | 0.0603 | 6.08E+21 | 43% |
| | Total | | 0.1407 | 1.41E+22 | |
| 37 | Σ fault | 1.900 | 0.1247 | 2.57E+22 | 28% |
| | zone | 1.917 | 0.3152 | 6.50E+22 | 72% |
| | Total | | 0.4399 | 9.07E+22 | |
| 38 | Σ fault | 2.180 | 0.3269 | 3.55E+22 | 62% |
| | zone | 2.240 | 0.2131 | 2.15E+22 | 38% |
| | Total | | 0.5400 | 5.70E+22 | |
| 39 | Σ fault | 1.820 | 0.1005 | 1.35E+22 | 31% |
| | zone | 1.750 | 0.2077 | 2.96E+22 | 69% |
| | Total | | 0.3082 | 4.30E+22 | |

---

## Referee Comment (RC4) · Anonymous Referee #4 · 13 Apr 2018

This manuscript presents an attempt at modeling the contribution of fault sources and zone sources for probabilistic seismic hazard assessment (PSHA). The authors start with explaining the methodological approach used to develop a hybrid model made up of faults and zones and then illustrate an application of this hybrid model in southern Spain. In this application, they also include a classical zone model to compare their hybrid model with.

The basic idea of this work is not new. It is at least as old as the seminal paper by Cornell (1968) on seismic risk analysis. Since then, there have been countless PSHA works that deal with the combination of zone sources, fault sources, and point

sources. None of these previous works is cited in the introduction, and with much disappointment of the reader, none of them is cited in the discussion. Indeed, not any other paper is cited in these two sections of the manuscript. On the one hand, this has to be regarded as an unethical issue related to the lack of recognition of others' work. On the other hand, this represents a major flaw that prevents the readers to understand what is actually different, innovative, and possibly better in this work with respect to other previous approaches. More specifically, it is very clear that the hybrid model presented by Rivas-Medina and coworkers is surprisingly similar to the Fault Source and Background (FSBG) model presented by Woessner et al. (2015) which, having being applied to all of Europe, obviously also covers the area of the application to southern Spain in this work. The only difference being the fact that Rivas-Medina and coworkers present their approach as if they have just (re)invented the wheel.

As regards the merit of this work, I found it is affected by a number of methodological flaws and possibly some miscalculations that challenge the overall validity of the results. Too many details are also missing to correctly understand how the model is developed and how the application to southern Spain was carried out. Mentioning the use of the software CRISIS is not enough for an explanation of the method and justification of strategic choices.

To be more specific, I'll touch in the following some of the main issues (I use P for the page number and L for the text line to identify the position of the text I'm referring to).

P3L18. The approach by Stepp (1972) for estimating the completeness periods needs various data manipulations that should be explained in more details to let the reader understand and replicate what was done here. This lack of details prevent the reader to appreciate the validity of the results.

P4L16. Here it is stated that the seismic moment rate of faults is calculated by summing up the seismic moment of earthquakes with magnitudes "close to m=0" up to a certain given maximum. To get an accurate seismic moment rate estimate for the

faults, all earthquakes with moment M0>0 must be considered. Notice that magnitude m=0 corresponds to seismic moment M0=1.27E+09 Nm using the relation by Kanamori and Brodsky (2001). Basically, this mistake leaves out a lot of seismic moment from the moment rate estimate when the Eq. (5) and Eq. (6) are used. This significantly impacts into the correct estimation of the number of earthquakes for which the hazard is then calculated.

Figure 5. Although this figure is not very clear (a map view would have been much better), it shows that several faults are cut by zone boundaries. How was the fault moment rate assigned to the zones in these cases?

Table 3 and Table 4. The b-values reported in these tables are utterly absurd. Are they really the b-values of the GR relation? If not, please explain what they actually are, otherwise I suspect that they result from gross calculation errors. In general b-values are ca. 1 everywhere in tectonic regions all around the world. I've seen b-value estimates in various works ranging from 0.7 to 1.3, but here they are in the range 1.7-2.4 that has never been seen anywhere.

P7L15. The use of the Stirling et al. (2002) fault scaling laws is questionable and potentially the source of additional miscalculations. First of all, which one of the Stirling et al.'s (2002) relationship was used here? Stirling et al. (2002) provide relationships between Mw and either length (L) or area (A). In the second case, A is obtained from L multiplied by an assumed fixed width. In both cases, L is the surface rupture length. Since the hazard model is concerned with seismic shaking that is generated at depth, the surface rupture is not the most suitable observation to relate with. Rather, a relationship between Mw and rupture dimension at depth should be used. The relationships by Leonard (2014) do provide such parameters and are based on a much larger and more updated dataset than Stirling et al. (2002). In addition, Leonard's (2014) would allow for differentiating between strike-slip and dip-slip faults. In all cases, the statement that the maximum magnitude is estimated from the fault geometry is too general. More detailed explanation is needed to let the reader understand and possibly replicate was

has been done here.

P8L5. Here it is stated that the GMPEs by Campbell (2013) are used. I suspect it is the Campbell and Bozorgnia (2014) in the references. These are GMPE developed for shallow crustal earthquakes in the western US in the moment magnitude range of 5.0 to 8.5. How well do they apply to earthquakes in the range starting at magnitude 4.0 in southern Spain? What criteria have been used to select this GMPE? The paper by Delavaud et al. (2012) delineates a robust procedure to select the appropriate GMPE to be used in Europe, why was this paper ignored? In addition, the GMPEs are different depending on the sense of movement. The QAFI database provides indication of the sense of movement of faults, but how was the sense of movement determined for the zones?

P9L27-28. What is said here does not prevent double counting at all. To prevent double counting every earthquake that is assigned to its causative fault should be removed from the rate estimate of the zone and vice versa to ensure it is counted only once.

The results of the various calculations are very poorly presented. The fault parameter estimates are not provided at all, and other results are provided only in aggregated form. It is thus very hard to judge the validity of the hazard results if the results of the intermediate calculations are missing.

In general, I found the English grammar and the organization of the manuscript to be rather poor. For example, there are sentences in the results that belong to the discussion. I've already commented on the lack of referencing. The symbols used in the equations are never explained. There is often confusion between symbols used for seismic moment and earthquake magnitude ($M$, $M_0$, $m$, which is which?). The same also for the b-value ($b$ or $beta$?). The units in the vertical axis of diagrams in figures 3 and 4 are unclear. Overall, the figure captions do not help much to understand what the figures show.

My conclusion is that this manuscript is not fit for publication. I also cannot see how

this manuscript can evolve quickly to an acceptable standard for the readers of NHESS and thus recommend it be rejected.

I don't give more technical suggestions here on how to improve the manuscript because they won't be useful until the major flaws are fixed.

References

Cornell CA (1968) Engineering seismic risk analysis. Bull Seismol Soc Am 58:1583–1606.

Delavaud E., Cotton F., Akkar S., Scherbaum F., Danciu L., Beauval C., Drouet S., Douglas J., Basili R., Sandikkaya M., Segou M., Faccioli E., Theodoulidis N. (2012). Toward a ground-motion logic tree for probabilistic seismic hazard assessment in Europe. Journal of Seismology, 16(3), 451-473, doi: 10.1007/s10950-012-9281-z.

Kanamori, H. & Brodsky, E.E., 2001. The physics of earthquakes, Phys. Today, 54(6), 34–40.

Leonard, M., 2014. Self-Consistent Earthquake Fault-Scaling Relations: Update and Extension to Stable Continental Strike-Slip Faults. Bulletin of the Seismological Society of America, doi: 10.1785/0120140087.

Woessner J., Danciu L., Giardini D., Crowley H., Cotton F., Grünthal G., Valensise G., Arvidsson R., Basili R., Demircioglu M., Hiemer S., Meletti C., Musson R.W., Rovida A., Sesetyan K., Stucchi M., and the SHARE consortium. (2015). The 2013 European Seismic Hazard Model - Key Components and Results. Bulletin of Earthquake Engineering, 13, 3553-3596, doi: 10.1007/s10518-015-9795-1.
* * *

---

## Short Comment (SC1) · 13 Apr 2018

GENERAL COMMENTS: The paper "Approach for combining faults and area sources in seismic hazard assessment: Application in southeastern Spain, by Alicia Rivas-Medina et al." can add interesting discussions points on the seismic hazard issues because of the hybrid source model. However, major changes are required to improve the paper. The background knowledge of the application area (Southeastern Spain) is poorly described and the choice of Mmaxc should be discussed more deeply.

SPECIFIC COMMENTS: Abstract - Lines 9-11 page 1: Instead of " . . .model composed by faults as independent entities and zones (containing the residual seismicity). The

seismic potential of both types of sources is derived from different data: for the zones, the recurrence model is estimated from the seismic catalog. For fault sources, it is inferred from kinematic parameters derived from paleoseismicty and GNSS measurements" a suggested re-writing could be (in the list put first fault and then zones): "... model composed by faults as independent entities and zones (containing the residual seismicity). The seismic potential of both types of sources is derived from different data: for the fault sources it is inferred from kinematic parameters derived from paleoseismicty and GNSS measurements, and for the zones the recurrence model is estimated from the seismic catalog".

Lines 15-17 pag 1: The concept of the Max Magnitude in the abstract (stated by the following words "This is derived from a completeness analysis and can be lower than the Mmax generated by the faults, taking into account that their the recurrence period can be higher than the observation period of the catalog") starts a discussion but it is not fully developed, it is just mentioned here and needs to be better explained in the "Discussion and Conclusions" section or when the results are presented. This part in the abstract should be removed and/or re-written.

Line 19 page 1: It is required an explanation of "...a seismic hazard model using the traditional zone". Is there any reference of this model? What do the authors mean by the word "traditional"? This part should be re-written.

1 Introduction - There are NO REFERENCES in the Introduction. This is an anomalous way of presenting a scientific paper in a peer review journal because the paper gives a narrow view of the subject. Please add appropriated references in the text. The Introduction looks more a discussion to motivate the paper than a wide presentation of the seismic hazard problem in the application region. Also, there is no mention of the previous seismic studies carried out in Southeastern Spain.

Line 27 page 1: References are required near the words "... in the last years, as more studies".

Line 28 page 1: Please write the acronimo as "GNSS (Global Navigation Satellite System)".

Line 2 page 2: Please add reference after " In most practical cases".

Line 3 page 2: Please add reference after "Other approaches".

Line 22 page 2: Please add reference after " Some authors ".

Lines 28-32 page 1: The sentences "The approach presented in this paper, as all probabilistic seismic hazard models, face the challenging question of estimating the expected ground motions with the basis of a short period of observations of earthquake occurrences and limited geological data (with significant uncertainty) to construct recurrence models. The purpose of this work is not to solve this challenge, but rather, to propose a model that contains different types of seismic sources (faults and zones) and distributes the seismic potential appropriately, avoiding double-counting and considering periods of completeness." present the purpose of this study in a superficial way. Those lines should be re-written. If you want to start a probabilistic hazard assessment, firstly you consider the potential Max magnitudes generated by the faults, and then you associate a low probability of earthquake occurrence with them on the basis of your study and considering the relative uncertainties. Certainly you don't exclude those max magnitudes just because of the completeness of the catalogue. A probabilistic hazard assessment should overcome those limitations. Again the choice of the Mmaxc should be properly explained, doubts on that should be solved and motivated by the authors in the paper. This part should be re-considered for the discussion in the last section.

2 Source hybrid approach (zones & faults) for hazard estimation - This section is presented in a schematic way, but should be completed with the description of the "Classical Method" which is also used in comparison with the Hybrid Model.

Lines 10-16 page 3 : The definitions should be clearer, in particular CP(m) and PC(m)

Interactive
comment

need a longer explanation.

Line 20 page 4: Please write GR after Gutenberg-Richter and may be write a reference for that (Gutenberg and Richter 1944 or 1954).

Line 22 page 4: In Eq 4 it should be written what "d" and "beta" are.

Line 18 page 5: The authors state " Considering that the fault may generate events with magnitudes larger than MMaxC , the corresponding distribution of seismic potential in the interval (MMaxC , MMax ] is calculated by extrapolation of the recurrence model with the last b-value adjusted". This concept should be further discussed to overcome the limitations raised by the choice of the Mmaxc.

3 Application of the approach in southeast Spain - It is not clear how the results in the application region were computed with the Classical Method.

Line 14 pag 7: Please explain what QAFI database is and/or write references for that.

Line 9 page 8: The "Classical Method" should be better explained, it is just mentioned for the Fig 6.

4 Discussion and conclusions - The discussion of the results and the conclusions should be include the point raised previously.

Line 6 page10: Please add references for 2009 L'Aquila and 2011 Lorca events.

References - They are a poorly and unsatisfactory list of the other paper on this subject. The References are simply the ones used to carried out the computation.

FIGURES - Figure 2: more explanations are needed in the caption, in particular about "AR".

TECHNICAL CORRECTIONS: Line 17 page 1: delete "the" before "recurrence".

Line 10 page: difficult to read. Instead of the sentence "using the records which origin time and magnitude are contained in the period for which the catalog can be consid-

ered complete" a suggested change could be: "using the records with origin time and magnitude contained in the period for which the catalog can be considered complete".

Line 27 page 8: This part on region 30 should be immediately after "return periods", to complete the part on hazard map.

Line 31 page 8: the sentence with "The hazard curves" should start a new paragraph, on the hazard curves.

---

## Referee Comment (RC5) · Anonymous Referee #5 · 22 Apr 2018

The Manuscript by Rivas-Medina et al. "Approach for combining faults and area sources in seismic hazard assessment: Application in southeastern Spain" addresses an important methodological question of how to incorporate individual faults and fault systems into the PSHA. This problem is even more actual for the PTHA (tsunami) studies due to the even larger impact of different faulting styles stimulating researchers to treat as much individual faults as possible. In particular, Authors suggest not using some constant accepted magnitude for distributing seismic potential between faults and background seismicity but employing more flexible criterion for the threshold magnitude based on completeness analysis.

As present Manuscript pretends to become an important methodological paper, I think it has to be significantly improved to meet the quality expected for such kind of paper, and propose major revision.

General comments:

References: The manuscript suffers from clear lack of proper referencing. It is, actually, unique in this sense. Especially, introduction lacks at least minimal comprehensive historical review with proper citations. One single statement at 2.22 (without citations) is definitely not enough.

Figures: Please provide proper explanatory captions. Figures should be self-explaining. In current state too much concise. Also give explanations to each abbreviation.

Language: please check with native speaker. Some sentences are hard to understand (e.g., the two last sentences of the abstract).

I suggest to decouple Discussion and Conclusions for more clarity.

Few concrete suggestions and comments below:

2.12 – "source" vs "source zone" possible misunderstanding 2.6 – unbiased estimates 3.11 - CP(m) or PC(m)? 3.12 – higher? 3.16 – rephrase

From the three equations 8-9-10 it is not clear how to update the b-values for individual faults, please explain in a better way.

In Table 3 caption and also through the manuscript: b-values mixed with beta-values.

---

## Author Comment (AC1) · 20 Jun 2018

Thank you for your comments and remarks. MmaxC refers to the maximum magnitude value that can be recorded in the catalog completely. This means that the catalog may contain events with higher magnitude value, but due to the long recurrence of these events, it can not be assured that the period of records covered by the catalog includes several recurrence periods of these events to make it possible to derive an (statistically) meaningful recurrence period value. The MmaxC value is used to constraint the distribution of seismic potential between faults and zones, but not to estimate seismic hazard. The maximum expected magnitudes for each source are

included in seismic hazard assessment. For fault sources, the maximum magnitude is obtained from the length of the fault plane. For zones (that contain the seismicity related to blind faults) it is more difficult to establish the maximum magnitude value, as it depends on the study area. It the application shown in this work, it is considered Mmaxzone=MmaxC+0.5. For our study area this implies a Mmaxzone value of up to 6.5. This is considered sufficient due to the short record of events with that magnitude in the catalog. The term Mmax is changed, distinguishing the different Mmax used for zones and fautls: Mmax(zone) and Mmax(fault) We only show the seismic hazard maps expressed in terms of PGA. The hazard maps for different spectral ordinates are not shown to reduce the length of the paper and because they do not display significant differences. Nevertheless, we include the response spectra for selected locations in figure 10. The case of Almeria and Granada, high acceleration values obtained with the HM are associated with documented faults, nearby these cities. The Mmax value for these faults is derived from the fault length , In the CM, the maximum magnitudes for each zone are modelled using a magnitude distribution that considers the maximum recorded magnitudes in the catalog and the maximum magnitudes expected in the faults (see details in Gaspar-Escribano et al., 2015). We make an effort to clarify these points in the text. We also include a new figure with the UHS spectra obtained for different cities (Granada, Almeria and Murcia) with both methods to show the impact of the source model in different spectral accelerations. Several references are included (including one to QAFI).

Please also note the supplement to this comment:
https://www.nat-hazards-earth-syst-sci-discuss.net/nhess-2018-28/nhess-2018-28-AC1-supplement.pdf

[Figure]

**Supplement:**

**Referee #1**

General: The paper is generally well written and presents a new alternative for combining faults and zones in a PSHA evaluation. The language is generally acceptable, but here and there the Spanish "accent" is coming through, and sometimes wrong wording is used. A thorough "language washing" by a native English-speaker is needed before publication.

On Mmax: This review is concerned with using the catalogue only up to Mmax recorded in the catalogue. What about blind faults that are not mapped and may trigger larger magnitudes away from the mapped faults. The authors should be more clear about this situation. This issue is exemplified in Fig. 6 where the Granada and Almeria hazard is so widely different from the zonation based hazard map. Fig. 6 should also be expanded with 1 Hz hazard difference in which the difference between methiods could be even higher.

On Mmax: After reading I am still somewhat uncertain how MmaxC and Mmax relate.

The authors should make some additional effort in clarifying the different Mmax used for faults and areas. May be use more clear annotations Mmax(fault)= MmaxF;

Mmax(zone)=MmaxZ and Mmin correspondingly.

On Mmax: May be I have overlooked, but how is Mmax established quantitatively from the catalogue?

QAFI database not defined/referenced.

Fig. 6 and 8: What is the Mmax used in the reference model? Due to the big differences it is important to be very clear about the reference computation. The implications in an application is significant.

Thank you for your comments and remarks.

MmaxC refers to the maximum magnitude value that can be recorded in the catalog completely. This means that the catalog may contain events with higher magnitude value, but due to the long recurrence of these events, it can not be assured that the period of records covered by the catalog includes several recurrence periods of these events to make it possible to derive an (statistically) meaningful recurrence period value. The MmaxC value is used to constraint the distribution of seismic potential between faults and zones, but not to estimate seismic hazard.

The maximum expected magnitudes for each source are included in seismic hazard assessment. For fault sources, the maximum magnitude is obtained from the length of the fault plane. For zones (that contain the seismicity related to blind faults) it is more difficult to establish the maximum magnitude value, as it depends on the study area. It the application shown in this work, it is considered Mmaxzone=MmaxC+0.5. For our study area this implies a Mmaxzone value of up to 6.5. This is considered sufficient due to the short record of events with that magnitude in the catalog.

The term Mmax is changed, distinguishing the different Mmax used for zones and fautls: Mmax(zone) and Mmax(fault)

We only show the seismic hazard maps expressed in terms of PGA. The hazard maps for different spectral ordinates are not shown to reduce the length of the paper and because they do not display significant differences. Nevertheless, we include the response spectra for selected locations in figure 10.

The case of Almeria and Granada, high acceleration values obtained with the HM are associated with documented faults, nearby these cities. The Mmax value for these faults is derived from the fault length ,

In the CM, the maximum magnitudes for each zone are modelled using a magnitude distribution that considers the maximum recorded magnitudes in the catalog and the maximum magnitudes expected in the faults (see details in Gaspar-Escribano et al., 2015).

We make an effort to clarify these points in the text. We also include a new figure with the UHS spectra obtained for different cities (Granada, Almeria and Murcia) with both methods to show the impact of the source model in different spectral accelerations.

Several references are included (including one to QAFI).

---

## Author Comment (AC2) · 20 Jun 2018

Thank you for your comments and remarks, which imply a considerable improvement to the manuscript. We give response to all the points raised by referee 2 below.

**General comments We include 33 new references in the document, specially in the introduction. This paragraph is changed, including the appropriate references and explaining the method. The graph of figure 2 illustrates this point. Equations 8 and 9 (new 12 and 13) add cumulative seismic moment rates and cumulative seismicity rates. Both are point values and not functions (as GR curves). The b-value give the ratio between events of different magnitudes and thus it changes the rate of earthquakes produced of**

each magnitude, but not the total earthquake rate nor the total released moment rate. We adjust the b-value that keeps those values (cumulative seismic moment rates and cumulative seismicity rates) constant for each source. We agree with you when you state that the GR curves cannot be just added, as they have different b-value.

**Specific Comments References have been included for all cases - P 3, Line 15: We wanted to refer to the fact that when recurrence periods are long, there are a few records of those events in the catalog, and consequently, the catalog presents a small sample, with limited statistical significance to establish recurrence periods. We changed the phrase for: "Magnitude values above MMaxC present recurrence periods higher than the catalogue OP. These values usually constitute a sample that does not include a high enough number of records to clearly establish the recurrence period, as this makes it increasingly difficult to constrain rates for rarer events." However, we want to indicate that we refer to recurrence periods (as the inverse of the earthquake occurrence rate) and we do not consider return periods at this point of the work (it is considered in hazard calculations). - P 3, lines 17-18: This paragraph is changed, references are included and the method is explained to understand figure 2 better. The reference to Stepp was a mistake that we have changed. - P3, Lines 19-20: We include two equations to explain how both parameters are estimated. However, we want to indicate that we extract these data from the catalog, and we do not assign a GR distribution in this part of the calculation. - P 3, Line 24: The text has been modified: "In this way, we avoid using magnitudes with long recurrence periods that have not been recorded in the catalogue within the completeness periods". Again, we make reference to recurrence periods and not to return periods. - P 4, Section 2.2: We consider as many faults per region as available in the fault database. All faults are assigned the same b-value and different occurrence rate (as each fault presents different moment rate). We corrected the ambiguity in the manuscript. - P 4, line 12: The slip rate (v), can be obtained from paleoseismicty studies and GNSS measurements. We include a reference to database of active faults providing this information. The shear modulus value is $\mu$ = 3.2 × 1010 Pa (Walters et al., 2009; Martínez-Díaz et al., 2012) - P 4, line**

16: The text is modified - P 4, line 26: We do not include events with magnitude below Mmin to estimate seismic hazard, but we take into account the moment rate related to events with magnitudes below Mmin to estimate the seismic moment rate in the interval Mmin-MmaxC. The moment rate assigned to each interval (0- MMin), (MMin, MMaxC), (MMaxC, MMax) depends on the b-value estimated for the fault. Hence, we must take into account this interval until a b-value is fixed. This part is changed for a better understanding. - P 4, line 22: The notation is changed and all parameters are defined - P 5, line 9: A comment is included - P 5, line 14: This topic is answered in "General comments". We remark that we do not equal the GR distributions, but only the total seismicity rate and moment rate, both values are concrete and constant for each source independently the ratio of different magnitudes that are calculated later on. - P5, Section 2.4. The uncertainty analysis focuses on how the input data (seismic catalog and fault database) affect the uncertainty in the end result. Concretely, table 1 shows how the magnitude interval, the number of records and the b-value of the seismic catalog affects the uncertainty in the seismic moment rate of the region, a parameter that will have a strong influence on the final result for all sources. Thus, table 1 only considers the uncertainties of the seismic catalog, which is used to model the seismic potential of the region. The uncertainty related to the input parameters of faults is tackled in the last paragraph of the section. - P7, line 4, line 14: The text is changed and the references included - P 7, line 16: For each fault we obtain a different Mmax, as a function of the length of the fault plane - P 7, line 29: Zone 30 only has 7 seismic records. Any statistical analysis with this sample size is not representative. The text is changed to clarify this point. - P 8, line 13: The text is changed - P8, line 20: A reference is included

**Technical corrections - P1, Line 6, P2, Line 18: The text is changed P2, Line 18: We understand that is that fault is active and generates earthquakes with not very long recurrence periods, the events associated to the fault are contained in the seismic catalog. We have rewritten the text to clarify this point. - P 2, line 26: Yes, it is complicated to assign a Mc value non-arbitrarily'. We have changed the text - P2: L28,**

P3: L5,10,11, P4: L1,20, P5: L9,19, P6: L11,12, P7: L26. line 9: 'An new' should be 'A new' We have changed the text

Please also note the supplement to this comment:
https://www.nat-hazards-earth-syst-sci-discuss.net/nhess-2018-28/nhess-2018-28-AC2-supplement.pdf
* * *
[Figure]

**Supplement:**

**Referee #2**

Thank you for your comments and remarks, which imply a considerable improvement to the manuscript. We give response to all the points raised by referee 2 below.

**General comments**

This paper is interesting and covers an important topic of interest to NHESS readers.

However, there are several significant weaknesses in the paper, which I estimate will require *major revisions* to address.

Although the major revisions may involve considerable work from the authors, I do think it should be possible to revise the paper to meet the standards of NHESS publication.

Before going into detail below, I note some 'overall' problematic aspects of the current draft:

- Poor referencing. The paper contains no references in the Introduction, despite discussing other work. Other areas also need more references (e.g. often non-obvious 'facts' are stated, without information on where they come from). I note a number of examples below, but not all – in general, the authors need to be more rigorous about justifying statements with references.

We include 33 new references in the document, specially in the introduction.

- There is insufficient explanation of the earthquake catalogue analysis methods (i.e. the discussion on page 3 in the paragraph around line 20 needs to be expanded).

This paragraph is changed, including the appropriate references and explaining the method. The graph of figure 2 illustrates this point.

- A number of statements in the paper are unclear or lacking in rigor (details below).

- I think there is a (fixable) logical flaw in the method. The authors are assuming (equations 8 and 9)

1: region_seismicity = zone_seismicity + fault_seismicity; and furthermore, at various points in the analysis they assume (equations 10 and 4):

2: all of these seismicity sources can individually be represented with GR curves;

The combination of (1) and (2) would be fine *if* all the seismicity sources have the same 'b-value', but the authors suggest different 'b-values' on fault vs region/zone.

Mathematically, I don't think (1) holds under this assumption (i.e. you can't sum 2 GR curves with different b values and expect to get another GR curve). I have suggested a possible fix below, which looks like it may be reasonable based on the data in the paper – but I can't be sure. Anyway, the authors should address this somehow.

Equations 8 and 9 (new 12 and 13) add cumulative seismic moment rates and cumulative seismicity rates. Both are point values and not functions (as GR curves).

The b-value give the ratio between events of different magnitudes and thus it changes the rate of earthquakes produced of each magnitude, but not the total earthquake rate nor the total released moment rate. We adjust the b-value that keeps those values (cumulative seismic moment rates and cumulative seismicity rates) constant for each source.

We agree with you when you state that the GR curves cannot be just added, as they have different b-value.

On the other hand, I expect these issues can be addressed. Furthermore I note there are good points to the paper: it treats an important topic, provides a good application of the methods (which I found much easier to read than the methods description itself), and includes a reasonable attempt to quantify the uncertainties (Section 2.4).

**Specific Comments**

- The Introduction does not contain any citations of other work. However, there is repeated mention of 'other studies' {e.g. p1 Line 25; p1 Line 27; p2 Line 3; p2, Line 22; p2, Line28 ... and many other instances, I will not list them all}. Throughout this section, the authors need to provide references to justify their descriptions of other work (even when they refer to general issues that are not specific to a particular study).

References have been included for all cases

- P 3, Line 15: ** Magnitudes values above MMaxC present recurrence periods higher than the catalog OP or constitute a sample an insufficient number of records to apply statistics. ** – This statement needs to be rephrased to correct the grammar. More importantly I think it is factually incorrect.

For illustration, suppose we have a catalogue of 100 years duration, and this catalogue contains zero events with Mw>=X. Then I claim that we **can apply statistics** to place bounds on the rate of events with Mw>=X, and furthermore, that **the return period of such events MIGHT be less than the observational period**. Both of these points contradict the author's statement. For example, with zero events in 100 years, an exact Poisson test of the true rate of events with Mw>=x (e.g. poisson.test(x=0,T=100) in the R software) gives a 95% confidence interval on the true rate of [0, 0.037] events/year.

So the return period is probably greater than around 1/0.037=27 years. This makes sense – if the return period were very small then it is likely an event would occur in the data. Furthermore, it is obviously possible that the return period is < 100 years (we don't get a '100 year event' in every hundred year observational period).

I think the authors probably want to say that it is increasingly difficult to constrain rates of rarer events – that's correct - but the authors need to weaken the statement accordingly.

We wanted to refer to the fact that when recurrence periods are long, there are a few records of those events in the catalog, and consequently, the catalog presents a small sample, with limited statistical significance to establish recurrence periods. We changed the phrase for:

"Magnitude values above MMaxC present recurrence periods higher than the catalogue OP. These values usually constitute a sample that does not include a high enough number of records to clearly establish the recurrence period, as this makes it increasingly difficult to constrain rates for rarer events."

However, we want to indicate that we refer to recurrence periods (as the inverse of the earthquake occurrence rate) and we do not consider return periods at this point of the work (it is considered in hazard calculations).

- P 3, lines 17-18: The authors should explain the 'Stepp (1972)' approach to estimating the reference years for different return periods. Although Figure 2 is related to this, I cannot understand the method from the figure.

This paragraph is changed, references are included and the method is explained to understand figure 2 better. The reference to Stepp was a mistake that we have changed.

- P3, Lines 19-20: **Then, it is possible to estimate ...** – the authors should explain how they do this estimation. There could be a range of methods (depending on the extent to which one makes parametric assumptions, like that the data-generating process has a GR distribution).

We include two equations to explain how both parameters are estimated. However, we want to indicate that we extract these data from the catalog, and we do not assign a GR distribution in this part of the calculation.

- P 3, Line 24: ** This way we avoid miscalculation problems ...**. I think this statement is too simplistic, and should be rewritten. In reality, errors in the estimated rates may be quite significant even for events with true return period significantly less than the catalogue duration. There are statistical methods for quantifying such uncertainties in different contexts.

For example, for individual rates one might assume a Poisson process and estimate confidence intervals as I suggested in an earlier comment. Or for a full magnitude frequency distribution, one might estimate uncertainty in the rates for any magnitude by assuming a GR distribution, and using maximum likelihood with profile likelihood confidence intervals - or using Bayesian techniques, etc. If the authors google-search 'Gutenberg Richter maximum likelihood estimation' they will find many related references.

The text has been modified: "In this way, we avoid using magnitudes with long recurrence periods that have not been recorded in the catalogue within the completeness periods". Again, we make reference to recurrence periods and not to return periods.

- P 4, Section 2.2: It's not clear to me whether the authors assume there is only 1 fault per region (this is how the math appears, and consistent with the use of the word 'fault' rather than 'faults' in most places), or if there are multiple faults and GR curves which are summed over (that would seem sensible, but no summation is

mentioned), or if there are multiple faults which are all treated with a single GR model (but there is no summation mentioned in equation 3, and the word 'fault' is mostly used, rather than 'faults'). This needs to be made much clearer to the reader, with 'summations over faults' included in the equations as appropriate. Table 4 makes me think it is 'multiple faults with single GR model', but it is far from clear.

We consider as many faults per region as available in the fault database. All faults are assigned the same b-value and different occurrence rate (as each fault presents different moment rate). We corrected the ambiguity in the manuscript.

- P 4, line 12: It is unclear how 'v' is estimated (I suppose from other geophysical studies? Or paleo work?). Please provide references. Also, how do you choose the shear modulus? References.

The slip rate (v), can be obtained from paleoseismicty studies and GNSS measurements. We include a reference to database of active faults providing this information.

The shear modulus value is $\mu = 3.2 \times 10^{10}$ Pa (Walters et al., 2009; Martínez-Díaz et al., 2012)

- P 4, line 16: Note that 'm=0' is not the lowest value of m. Maybe just say 'from very low up to the maximum ...'.

The text is modified

- P 4, line 26: Related to the above comment, you might make the lower summation index in the first right-hand-side term be "negative infinity" rather than zero. Also, I think you don't model earthquakes with m < M_min later in the paper? If so, I suggest mentioning that here. Indeed you might re-write this part to avoid mention of events below M_min, perhaps it is just confusing?

We do not include events with magnitude below Mmin to estimate seismic hazard, but we take into account the moment rate related to events with magnitudes below Mmin to estimate the seismic moment rate in the interval Mmin-MmaxC.

$$\sum_{M_{Min}}^{M_{MaxC}} \dot{n}(m) \cdot Mo(m) = \dot{Mo}_{fault} - \sum_{\sim 0}^{M_{Min}} \dot{n}(m) \cdot Mo(m) + \sum_{M_{MaxC}}^{M_{Max}} \dot{n}(m) \cdot Mo(m) \text{ of Eq. (6)}$$

The moment rate assigned to each interval (0- $M_{Min}$), ($M_{Min}$, $M_{MaxC}$), ($M_{MaxC}$, $M_{Max}$) depends on the b-value estimated for the fault. Hence, we must take into account this interval until a b-value is fixed.

This part is changed for a better understanding.

- P 4, line 22: I'm not sure if you have defined the $noverline{d}$ variable. Also I don't understand the notation "Mo|" in the numerator (what is the bar? what is Mo? Should it be for faults only?)

The notation is changed and all parameters are defined

- P 5, line 9: Here, I think you should remind the reader that the 'region' parameters are assumed known, based on Equations 1 and 2.

A comment is included

- P 5, line 14: **Note that the b-value of the zone appears in this equation can be equaled to the b-value of the region as both sources present similar seismic nature.** Mathematically I think this is problematic. Question: Is a distribution defined by summing 2 GR distributions with different 'b-values' still a GR distribution? I don't think so. However, that seems to be an implicit assumption of your method (i.e. because the regional seismicity is the sum of 2 different GR distributions, fault and zone, each having different b values). This suggests an internal contradiction in the methods, which should be fixed. Suggestion for a fix: Consider modifying your method so that b-fault is equal to b-region and b-zone. Then I think the issue would be avoided? Furthermore, from Table 4, that would look to be an OK approximation, given the statistical uncertainties in Table 1.

This topic is answered in "General comments". We remark that we do not equal the GR distributions, but only the total seismicity rate and moment rate, both values are concrete and constant for each source independently the ratio of different magnitudes that are calculated later on.

- P5, Section 2.4. I like this analysis overall. However, from Table 1 it seems like you are using a single b value in the simulations (?), in contrast to the above-mentioned 'distinct fault/region b values'. The testing here needs to be consistent with the methodology.

So if you change the analysis as I suggested above then this section is probably fine –but otherwise, you should treat these 'distinct fault/region b values'

The uncertainty analysis focuses on how the input data (seismic catalog and fault database) affect the uncertainty in the end result.

Concretely, table 1 shows how the magnitude interval, the number of records and the b-value of the seismic catalog affects the uncertainty in the seismic moment rate of the region, a parameter that will have a strong influence on the final result for all sources. Thus, table 1 only considers the uncertainties of the seismic catalog, which is used to model the seismic potential of the region.

The uncertainty related to the input parameters of faults is tackled in the last paragraph of the section.

- P7, line 4: Need a reference for the shortening rate.

- P 7, line 14: Reference for the QAFI database?

The text is changed and the references included

- P 7, line 16: Not clear to me how you estimate M-max if there are multiple faults per region. Do you assume they all rupture at once?

For each fault we obtain a different Mmax, as a function of the length of the fault plane

- P 7, line 29: '.. lacks a COV estimate because the sample of records is not significant ..' – I think you need to re-word this. What do you mean by 'not significant'? It's unclear – the word 'significant' is suggestive of 'statistical significance', but that does not appear to be what you mean. I think you just mean 'there is very little data'. However, I don't see why you can't calculate a COV coefficient (albeit a very uncertain one).

Zone 30 only has 7 seismic records. Any statistical analysis with this sample size is not representative. The text is changed to clarify this point.

- P 8, line 13: 'This result is characteristic of this method ...' – sounds like there must be other studies that show this, please add references.

The text is changed

- P8, line 20: "These results agree with real observations" – can you cite the study?

A reference is included

**Technical corrections**

- P1, Line 6: 'estimated' should be 'estimates'

- P2, Line 18: suggest changing 'a part of' to 'some of'

The text is changed

P2, Line 18: 'must be linked to faults' – is this always true? What about if there are not catalogue events on the faults? Suggest rewording.

We understand that is that fault is active and generates earthquakes with not very long recurrence periods, the events associated to the fault are contained in the seismic catalog. We have rewritten the text to clarify this point.

- P 2, line 26: 'fixing a Mc value results certainly complicated' - This doesn't make sense, suggest rewording. I'm not 100% sure what you want to say – do you mean 'it is difficult to choose Mc non-arbitrarily'?

Yes, it is complicated to assign a Mc value non-arbitrarily'. We have changed the text

- P2, Line 28: ** The approach presented in this paper, as all probabilistic seismic hazard models, face the challenging question of estimating the expected ground motions with the basis of a short period of observations of earthquake occurrences and limited geological data (with significant uncertainty) to construct recurrence models.** This needs to be re-worded to correct the grammar. One suggestion: ** The approach presented in this paper faces the challenging question of how to estimate the expected ground motion exceedance rate, using a short period of earthquake observations and limited geological data (with significant uncertainties). This challenge is faced by all probabilistic seismic hazard models. **

- P 3, L5: **The zone is defined as the source which seismic potential is residual, excluding the seismic potential of faults **. There are grammatical issues here, as stated it does not make sense. I think you mean 'The zone is the same as the region with fault seismicity removed'. However, you may choose to make different edits (considering also repetition in the subsequent sentence).

- P3, Line 10: This sentence needs rewording (grammar). What about: "The seismicity rate of the region is derived from the seismic catalog using events that are contained in the period for which ...."

- P 3, Line 11: You use 'CP' here, but on line 13 you use 'PC'. Please double check for consistency throughout the paper.

- P 3, Line 13: **The complete periods PC(m) (for different magnitude up to a maximum magnitude of completeness value, MMaxC.) are included in the observation period (OP) of the catalog. ** Is it really correct to say they are 'included'. I suspect you mean to say they are 'inferred using' or are 'less than'?

- P 4, line 1: I suggest you replace the word 'cumulative' with 'total'.

- P 4, line 20: 'establish' instead of 'stablish'

- P 4, line 20 and below. You refer to $\dot{n}_{Nmin}$ – should it have a subscript 'fault', considering later notation (Equation 8)?

- P5, line 9: 'An new' should be 'A new'

- P5, line 19: I think this should mention 'faults', e.g. 'by extrapolation of the faults recurrence model'

- P6, line 11: should say 'moment rates for different magnitude values, ...'

- P6, line 12: similar problem as above

- P 7, line 26: 'It is appreciated that ..' – this sounds strange – suggest you change to 'Note that ..'. Also, it's not clear

We have changed the text

---

## Author Comment (AC3) · 20 Jun 2018

Thank you for the numerous remarks. The changes that you propose are accurate and improve the paper considerably. All your suggestions are included in the manuscript. Here we answer to your questions English has been corrected by a native person in this version. Previous work is cited and referenced (more than 33 new references in the paper) - Seismic Moment Equation.: A reference and the equation of Hanks and Kanamori are included to facilitate understanding the issue - Equation 4: Two equations are included to explain how eq 4 (eq 8 in the new version ) is obtained,. All terms are explained - Equation 10: Eq 10 is explained with more detail - Figure

captions: Figures are explained with more detail - Figure 5: We include this 3D view instead of a map to get an idea of the depth dimension of faults. A location map is included - Fault sources: An annex listing the data of faults and faults segments used is included - Ground Motion Prediction Equation: The comparison between the results obtained with both methods must be performed in terms of relative acceleration values because the absolute acceleration values are conditioned by the specific ground motion prediction equation used. To avoid the influence of the GMPE in both results, the same GMPE is used to apply both methods. However, it remains open the possibility of integrating several GMPE in a logic tree framework to capture the epistemic uncertainty related to path effects. At the same time, it could be included the site effect in the analysis to see if the different source models affect different soil types similarly or not, This paper focuses on the impact of source models in hazard results. The impact of other factors (site effects, GMPE) can be the subject of future studies. This is indicated in the Discussion section. - Discussion and Conclusion: The Discussion and Conclusion sections have been separated. The issue regarding the use of GMPE is tackled therein. - Results. : The cited sentences are moved to the Discussion section - References list.: The reference list is reviewed - Table 2. : Table 2 includes the values of the region, not of faults. Regions without values refer to regions (28, 29, 33 and 40) with no faults identified within their limits. Thus, the distribution of potential between faults and zone is not done in these regions. This is clarified in the table. - Table 3. : Table 3 shows the values of the catalog that will be used n the distribution of seismic potential. Zones with very low MmaxC value, do not present events with higher magnitude in the catalog (as in region 30). For the rest of the regions, it is included the maximum recorded magnitude equal or lower than MmaxC. Recall that this interval is only to make the distribution of seismic potential, but the higher magnitudes recorded historically are considered in the seismic hazard calculations up t the maximum expected magnitudes. Please also note the supplement to this comment: All these comments have been taken into account

Please also note the supplement to this comment:
https://www.nat-hazards-earth-syst-sci-discuss.net/nhess-2018-28/nhess-2018-28-AC3-supplement.pdf

**Supplement:**

**Referee #3**

Thank you for the numerous remarks. The changes that you propose are accurate and improve the paper considerably. All your suggestions are included in the manuscript. Here we answer to your questions

The paper it is a valuable and original contribution in the subject of fault source modelling in PSHA. The proposed approach has been used in a SH calculation in Spain with satisfactory results. It would be interesting to check the viability of the model in other parts of the world with faster moving faults.

Nevertheless, there are few issues that, if tackled appropriately, will greatly improve the paper, and I think they should be addressed obligatory before publication in NHESS (please see list below).

Additionally, I attached the manuscript with many comments, suggestions, and corrections, all of them highlighted in yellow and commented using Adobe reader tools.

- Improve the English. Please, for a new submission make sure that the entire article is revised by an English native speaker. The style could also be greatly improved. Avoid the excessive use of "extra comments" in brackets (e.g., page 1, line 9; p.2 lines 17, 30, 31; p.3 line 4, 12;: : : and so many more: : :). I attached the manuscript pdf highlighting that, typos, few mistakes and so.

English has been corrected by a native person in this version.

- Citations. In the introduction there is not a single cite about other previous approaches about incorporating faults as sources in PSHA neither globally nor locally. The authors must cite other previous work in the subject (e.g., Youngs and Coppersmith, 1985; Wesnousky, 1986; Anderson and Luco, 1983; Bungum, 2007; : : :... and much more recently the different UCERF versions (e.g., Field et al., 2009, 2014), the SHARE project (Woessner et al., 2015),: : : among others. If there are, cite also other previous work done in your study area.

Previous work is cited and referenced (more than 33 new references in the paper)

- Seismic Moment Equation. It is necessary to cite and write the equation used for calculating the seismic moment from magnitude (Mo as function of (Mw), Hanks y Kanamori, 1979, IASPEI, 2005) for a completely comprehension of equation (2) and, later, equation (4). Additionally, you should state before that with the variable "m" you always refer to magnitude in the moment magnitude (Mw) scale.

A reference and the equation of Hanks and Kanamori are included to facilitate understanding the issue

- Equation 4. I assume this equation is an original contribution from this work? Please, you need to show how do you get to equation 4 from Anderson (1979) integral. Need to explain what is parameter d (in relation to Mo f (Mw).

Two equations are included to explain how eq 4 (eq 8 in the new version ) is obtained,. All terms are explained

- Equation 10. Further explanations should be given on how you get to equation 10.0

Eq 10 is explained with more detail

- Figure captions. They are very short. More explanations in the captions are needed to fully understand the figures. Particularly figure 2 (see attached pdf).

Figures are explained with more detail

- Figure 5. This figure can be greatly improved from what it is now (a dumped screen). A inset locating geographically the studied area is necessary.

We include this 3D view instead of a map to get an idea of the depth dimension of faults. A location map is included

- Fault sources. The paper will improve greatly if you further explain how did you select the faults and what are there characteristics in terms of slip rate, Mmax, kinematics, etc.

An annex listing the data of faults and faults segments used is included

- Ground Motion Prediction Equation. The GMPE you chose for your calculations was produced to address near fault-source effects. I understand that the point of the paper is to use your methodology for sharing the seismic potential between zones and faults, but because you have chosen this particular GMPE for your calculations you should explore the impact of using it in your hazard results compare to the use of a general GMPE one. I mean, this is important because in your results it is clearly shown that the hazard increases a lot near the faults, as you state p.10 l.3: "The results show an increment of expected accelerations near fault traces (in a factor of 2)." And later: "This increment is achieved at the expense of decreasing expected

accelerations in areas located farther away from faults." This statement should be properly discussed in the Discussion section.

The comparison between the results obtained with both methods must be performed in terms of relative acceleration values because the absolute acceleration values are conditioned by the specific ground motion prediction equation used. To avoid the influence of the GMPE in both results, the same GMPE is used to apply both methods.

However, it remains open the possibility of integrating several GMPE in a logic tree framework to capture the epistemic uncertainty related to path effects. At the same time, it could be included the site effect in the analysis to see if the different source models affect different soil types similarly or not, This paper focuses on the impact of source models in hazard results. The impact of other factors (site effects, GMPE) can be the subject of future studies. This is indicated in the Discussion section.

- Discussion and Conclusion should be separated sections. The paper will improve greatly if you discuss your results in terms of earthquake rates contribution from faults vs zones, instead of accelerations. This way it would be showed the real (pure) impact of your approach in the hazard, without consideration of the GMPE. Subsequently, you can explore the impact of using a different GMPE in the calculations.

The Discussion and Conclusion sections have been separated. The issue regarding the use of GMPE is tackled therein.

- Results. This section should be rewritten. It contains many statements that are better placed in a "Discussion" section. See attached pdf.

The cited sentences are moved to the Discussion section

- References list. There are a couple of references listed but missing in the manuscript.

The reference list is reviewed

- Table 2. Mmax relates to the Max event from the fault. When there is more than one fault in the region, which value do you state on this table? In the range Mmin-Mmax you show the accumulated moment rate from all the faults in the region? How can be regions with MaxC blank? : : : More information is needed to understand this table properly.

Table 2 includes the values of the region, not of faults. Regions without values refer to regions (28, 29, 33 and 40) with no faults identified within their limits. Thus, the distribution of potential between faults and zone is not done in these regions. This is clarified in the table.

- Table 3. More information is needed in the caption to understand it properly. Information on some regions is missed. The MmaxC values are strikingly low: : : What happened to the larger values (big historical events)?

Table 3 shows the values of the catalog that will be used n the distribution of seismic potential. Zones with very low MmaxC value, do not present events with higher magnitude in the catalog (as in region 30). For the rest of the regions, it is included the maximum recorded magnitude equal or lower than MmaxC. Recall that this interval is only to make the distribution of seismic potential, but the higher magnitudes recorded historically are considered in the seismic hazard calculations up t the maximum expected magnitudes.

Please also note the supplement to this comment:

https://www.nat-hazards-earth-syst-sci-discuss.net/nhess-2018-28/nhess-2018-28-RC3-supplement.pdf

All these comments have been taken into account

---

## Author Comment (AC4) · 20 Jun 2018

The Cornell method (1968) is a zoned probabilistic method, based on the consideration of seismogenic zones with homogenous seismic potential, which was raised precisely by its author in view of the difficulty of modeling the faults as independent seismic sources. It has been a method of widespread use in the last decades. Although in recent years, with the increasing increase of fault information, combined methods of zones and faults have begun to be proposed, such as those referenced in our current version of the manuscript. Obviously there may be many other works in this methodological line not mentioned in our work, but we are not presenting a paper on

the state of the art in the subject. We present a methodological approach that aims to be a contribution in this line of hybrid methods, and we say so in the manuscript. Some representative references have been cited by way of example. The qualification of "unethical issue related to the lack of recognition of other work" b is therefore unacceptable. We raise the following question: does each time a paper on a specific topic is published consider a lack of ethics not mentioning all the existing works on that topic? Where is the limit considered? The hybrid FSBG model presented by Woessner et al (2015) and applied in Europe resolves the distribution of the seismic potential between zones and faults adopting a Mc cutoff magnitude of Mw 6.5, above which earthquakes associated with faults are considered, taking as a background seismicity the one corresponding to magnitudes in the MW range (4.5-6.4), which is associated with the zone, that is, the method of Woessner et at (2015) considers a fixed cut magnitude, and precisely our approach is aimed at avoiding the adoption of a fixed Mc value, based on an essential question that is formulated on page 2 of the manuscript, where it is literally indicated: By not fixing this magnitude, the approach to distribute the seismic potential is obviously complicated and what we propose is a procedure that we detail in the paper, including its formulation. Therefore, our methodology differs substantially and essentially from that of Woesner et al (2015), both in the initial hypothesis and in the procedure to be followed. It must be added that a value of M = 6.5 is practically the Mmax that can generate the active faults in Spain, therefore it would not make sense to establish this value as Mc, but it is not easy to establish another alternative value either. It is surprising that the reviewer describes the methodology proposed here as "surprisingly similar to the Fault Source and Background (FSBG)" and denotes the lack of grasp of the essential aspects of both methodologies. To this we must add the notable difference of results in the application to the south of Spain. The comment: "The only difference being the fact that Rivas-Medina and coworkers present their approach as if they have just (re) invented the whee" is also offensive. Obviously, we are not trying to invent the wheel, but rather to propose an approach in an open line of research that is not based on the consideration of a Mc to distribute

the seismic potential between zones and faults. This is, in fact, a recognized problem among all the experts that work on seismic hazard towards which considerable efforts are being devoted, and our work is intended to be a contribution in this regard. So we have raised it humbly and repeatedly in the manuscript. In the work of Woessner et al. (2015) three source models implemented in a logical tree are presented: Area source (AS); Sismicity + fauts (SEIFA) and Fault souce (FS) & BackGroup (BG). Of the three previous models, only the last one deals with a hybrid model of faults and zones. The authors consider a cutoff magnitude (Mc = 6.5) between the faults and the zone (background seismicity). This idea, which is not novel in that work either, was proposed by Frankel et al. (1996), is the most important difference between the presented here and the work of Woessner et al. (2015). In fact, this issue is addressed in the introduction to this paper, since our approach is precisely not to use a previous cut magnitude. The approach presented in this paper is part of the PhD thesis of Alicia Rivas Medina, the first author of the paper. The public defense of the thesis was on March 2014. The pdf of the thesis was uploaded to the institutional, open repository of UPM on April 2014. It is accessible in http://oa.upm.es/23328/. P3L18: The completeness period was mistakenly referenced as Stepp (1972). The correct reference is provided in the text. P4L16: The seismic moment associated with an earthquake of M = 0, (M0 = 1.27E + 09 Nm), is a completely insignificant value when compared to the seismic momentum rate accumulated in a failure annually. For example, if we assume a slow failure with slip rate = 0.1 mm / year and a failure plane size of 45x10 km, the cumulative annual seismic moment rate is 1.35E + 22 Nm. This means that the moment released in an earthquake of magnitude M = 0 supposes 0.00000000001%, a completely insignificant value in a year, even more so in the periods of recurrence associated with slow faults. There should be many earthquakes of magnitude M = 0 to modify the result very slightly. Figure 5: There is no case in which zones cut faults, in fact the author of that zoning is also the author of the fault database (Garcia Mayordomo et al, 2010), and this zoning was designed to avoid that case. Maybe it is a misperception by the projection of the image Table 3 and Table 4:

[Figure]

The values shown in tables 3 and 4 are not values of b, but values of Beta, as clearly indicated in these tables. Do not confuse these two parameters, although there is an equivalence between them (Beta = b * ln (10)). Values of b (0.7 - 1.3) are equivalent to beta values (1.6 - 3.0). These equivalences are well known among those who work on issues of seismic hazard P7L15: It is a subjective opinion of the reviewer that it is preferable to use the relationships of Leonard (2014) to Stirling et al's (2002). The latter has been, together with that of Well and Coppersmith (1996), one of the most used for the purpose in question. P8L5: The Campbell and Bozorgnia model (2014) uses 15,521 records from 322 earthquakes of $3.0 \le M \le 7.9$. The total selected database comprises 11,125 records from 245 earthquakes of $3.0 \le M < 5.5$. The work of Delavaud et al. (2012) is prior to the model of Campbell and Bozorgnia (2014), so this model can not be assessed in that article. In addition, the GMPEs proposed (in the first places) Delavaud et al. (2012) do not consider the source effect with as much detail as the model of Campbell and Bozorgnia (2014), in this application it does not make sense to define the sources precisely if simpler models are later employed in the GMPEs. Nevertheless, the focus of the paper is the definition of the source model and the distribution of potential between zone and faults. The choice of the GMPE is a secondary issue in this regard. P9L27-28. The problem, precisely in a hybrid model of zones and faults, is to identify which earthquakes are associated to the zone and which to the fault. The events of the seismic catalog are not classified between zones and faults, but that a recurrence is established for the faults from the slip rate and another for the residual zone from the Catalogue. But this in turn will contain earthquakes that will have occurred in the fault, and if they are not easily identified they will be counted twice: one explicit in the area and another implicit in the fault. Most of the hybrid models, including that of adopting a solution to identify the events in the two types of sources: establish a Cut Magnitude Mc and consider Mw <Mc for the zone and Mw> Mc for the fault. But as we have already indicated, our approach tries to avoid this simplification and proposes a procedure for sharing, avoiding duplication. This question is key. The results: We include an annex containing a table with the

fault parameters included in the study. These data are taken from the QAFI database. Including in this table all the intermediate results for each fault would be too lengthy. English: English has been corrected by a native person in this version.

Please also note the supplement to this comment:
https://www.nat-hazards-earth-syst-sci-discuss.net/nhess-2018-28/nhess-2018-28-AC4-supplement.pdf

**Supplement:**

**Referee #4**

This manuscript presents an attempt at modeling the contribution of fault sources and zone sources for probabilistic seismic hazard assessment (PSHA). The authors start with explaining the methodological approach used to develop a hybrid model made up of faults and zones and then illustrate an application of this hybrid model in southern Spain. In this application, they also include a classical zone model to compare their hybrid model with.

The basic idea of this work is not new. It is at least as old as the seminal paper by Cornell (1968) on seismic risk analysis.

Since then, there have been countless PSHA works that deal with the combination of zone sources, fault sources, and point sources. None of these previous works is cited in the introduction, and with much disappointment of the reader, none of them is cited in the discussion. Indeed, not any other paper is cited in these two sections of the manuscript. On the one hand, this has to be regarded as an unethical issue related to the lack of recognition of others' work.

The Cornell method (1968) is a zoned probabilistic method, based on the consideration of seismogenic zones with homogenous seismic potential, which was raised precisely by its author in view of the difficulty of modeling the faults as independent seismic sources. It has been a method of widespread use in the last decades. Although in recent years, with the increasing increase of fault information, combined methods of zones and faults have begun to be proposed, such as those referenced in our current version of the manuscript. Obviously there may be many other works in this methodological line not mentioned in our work, but we are not presenting a paper on the state of the art in the subject. We present a methodological approach that aims to be a contribution in this line of hybrid methods, and we say so in the manuscript. Some representative references have been cited by way of example. The qualification of "unethical issue related to the lack of recognition of other work" b is therefore unacceptable. We raise the following question: does each time a paper on a specific topic is published consider a lack of ethics not mentioning all the existing works on that topic? Where is the limit considered?

On the other hand, this represents a major flaw that prevents the readers to understand what is actually different, innovative, and possibly better in this work with respect to other previous approaches. More specifically, it is very clear that the hybrid model presented by Rivas-Medina and coworkers is surprisingly similar to the Fault Source and Background (FSBG) model presented by Woessner et al. (2015) which, having being applied to all of Europe, obviously also covers the area of the application to southern Spain in this work. The only difference being the fact that Rivas-Medina and coworkers present their approach as if they have just (re)invented the wheel.

The hybrid FSBG model presented by Woessner et al (2015) and applied in Europe resolves the distribution of the seismic potential between zones and faults adopting a Mc cutoff magnitude of Mw 6.5, above which earthquakes associated with faults are considered, taking as a background seismicity the one corresponding to magnitudes in the MW range (4.5-6.4), which is associated with the zone, that is, the method of Woessner et at (2015) considers a fixed cut magnitude, and precisely our approach is aimed at avoiding the adoption of a fixed Mc value, based on an essential question that is formulated on page 2 of the manuscript, where it is literally indicated:

By not fixing this magnitude, the approach to distribute the seismic potential is obviously complicated and what we propose is a procedure that we detail in the paper, including its formulation. Therefore, our methodology differs substantially and essentially from that of Woesner et al (2015), both in the initial hypothesis and in the procedure to be followed.

It must be added that a value of M = 6.5 is practically the Mmax that can generate the active faults in Spain, therefore it would not make sense to establish this value as Mc, but it is not easy to establish another alternative value either.

It is surprising that the reviewer describes the methodology proposed here as "surprisingly similar to the Fault Source and Background (FSBG)" and denotes the lack of grasp of the essential aspects of both methodologies. To this we must add the notable difference of results in the application to the south of Spain.

The comment: "The only difference being the fact that Rivas-Medina and coworkers present their approach as if they have just (re) invented the whee" is also offensive. Obviously, we are not trying to invent the wheel, but rather to propose an approach in an open line of research that is not based on the consideration of a Mc to distribute the seismic potential between zones and faults. This is, in fact, a recognized problem among all the experts that work on seismic hazard towards which considerable efforts are being devoted, and our work is intended to be a contribution in this regard. So we have raised it humbly and repeatedly in the manuscript.

In the work of Woessner et al. (2015) three source models implemented in a logical tree are presented: Area source (AS); Sismicity + fauts (SEIFA) and Fault souce (FS) & BackGroup (BG). Of the three previous models, only the last one deals with a hybrid model of faults and zones. The authors consider a cutoff magnitude (Mc = 6.5) between the faults and the zone (background seismicity). This idea, which is not novel in that work either, was proposed by Frankel et al. (1996), is the most important difference between the presented here and the work of Woessner et al. (2015). In fact, this issue is addressed in the introduction to this paper, since our approach is precisely not to use a previous cut magnitude.

The approach presented in this paper is part of the PhD thesis of Alicia Rivas Medina, the first author of the paper. The public defense of the thesis was on March 2014. The pdf of the thesis was uploaded to the institutional, open repository of UPM on April 2014. It is accessible in http://oa.upm.es/23328/.

As regards the merit of this work, I found it is affected by a number of methodological flaws and possibly some miscalculations that challenge the overall validity of the results. Too many details are also missing to correctly understand how the model is developed and how the application to southern Spain was carried out. Mentioning the use of the software CRISIS is not enough for an explanation of the method and justification of strategic choices.

To be more specific, I'll touch in the following some of the main issues (I use P for the page number and L for the text line to identify the position of the text I'm referring to).

P3L18. The approach by Stepp (1972) for estimating the completeness periods needs various data manipulations that should be explained in more details to let the reader understand and replicate what was done here. This lack of details prevent the reader to appreciate the validity of the results.

The completeness period was mistakenly referenced as Stepp (1972). The correct reference is provided in the text.

P4L16. Here it is stated that the seismic moment rate of faults is calculated by summing up the seismic moment of earthquakes with magnitudes "close to m=0" up to a certain given maximum. To get an accurate seismic moment rate estimate for the faults, all earthquakes with moment M0>0 must be considered. Notice that magnitude m=0 corresponds to seismic moment M0=1.27E+09 Nm using the relation by Kanamori and Brodsky (2001). Basically, this mistake leaves out a lot of seismic moment from the moment rate estimate when the Eq. (5) and Eq. (6) are used. This significantly impacts into the correct estimation of the number of earthquakes for which the hazard is then calculated.

The seismic moment associated with an earthquake of M = 0, (M0 = 1.27E + 09 Nm), is a completely insignificant value when compared to the seismic momentum rate accumulated in a failure annually.

For example, if we assume a slow failure with slip rate = 0.1 mm / year and a failure plane size of 45x10 km, the cumulative annual seismic moment rate is 1.35E + 22 Nm.

This means that the moment released in an earthquake of magnitude M = 0 supposes 0.00000000001%, a completely insignificant value in a year, even more so in the periods of recurrence associated with slow faults. There should be many earthquakes of magnitude M = 0 to modify the result very slightly.

Figure 5. Although this figure is not very clear (a map view would have been much better), it shows that several faults are cut by zone boundaries. How was the fault moment rate assigned to the zones in these cases?

There is no case in which zones cut faults, in fact the author of that zoning is also the author of the fault database (Garcia Mayordomo et al, 2010), and this zoning was designed to avoid that case. Maybe it is a misperception by the projection of the image

Table 3 and Table 4. The b-values reported in these tables are utterly absurd. Are they really the b-values of the GR relation? If not, please explain what they actually are, otherwise I suspect that they result from gross calculation errors. In general bvalues are ca. 1 everywhere in tectonic regions all around the world. I've seen b-value estimates in various works ranging from 0.7 to 1.3, but here they are in the range 1.7-2.4 that has never been seen anywhere.

The values shown in tables 3 and 4 are not values of b, but values of Beta, as clearly indicated in these tables. Do not confuse these two parameters, although there is an equivalence between them (Beta = b * ln (10)). Values of b (0.7 - 1.3) are equivalent to beta values (1.6 - 3.0). These equivalences are well known among those who work on issues of seismic hazard

P7L15. The use of the Stirling et al. (2002) fault scaling laws is questionable and potentially the source of additional miscalculations. First of all, which one of the Stirling et al.'s (2002) relationship was used here? Stirling et al. (2002) provide relationships between Mw and either length (L) or area (A). In the second case, A is obtained from L multiplied by an assumed fixed width. In both cases, L is the surface rupture length. Since

the hazard model is concerned with seismic shaking that is generated at depth, the surface rupture is not the most suitable observation to relate with. Rather, a relationship between Mw and rupture dimension at depth should be used. The relationships by Leonard (2014) do provide such parameters and are based on a much larger and more updated dataset than Stirling et al. (2002). In addition, Leonard's (2014) would allow for differentiating between strike-slip and dip-slip faults. In all cases, the statement that the maximum magnitude is estimated from the fault geometry is too general. More detailed explanation is needed to let the reader understand and possibly replicate was has been done here.

It is a subjective opinion of the reviewer that it is preferable to use the relationships of Leonard (2014) to Stirling et al's (2002). The latter has been, together with that of Well and Coppersmith (1996), one of the most used for the purpose in question.

P8L5. Here it is stated that the GMPEs by Campbell (2013) are used. I suspect it is the Campbell and Bozorgnia (2014) in the references. These are GMPE developed for shallow crustal earthquakes in the western US in the moment magnitude range of 5.0 to 8.5. How well do they apply to earthquakes in the range starting at magnitude 4.0 in southern Spain? What criteria have been used to select this GMPE? The paper by Delavaud et al. (2012) delineates a robust procedure to select the appropriate GMPE to be used in Europe, why was this paper ignored? In addition, the GMPEs are different depending on the sense of movement. The QAFI database provides indication of the sense of movement of faults, but how was the sense of movement determined for the zones?

The Campbell and Bozorgnia model (2014) uses 15,521 records from 322 earthquakes of $3.0 \leq M \leq 7.9$. The total selected database comprises 11,125 records from 245 earthquakes of $3.0 \leq M < 5.5$.

The work of Delavaud et al. (2012) is prior to the model of Campbell and Bozorgnia (2014), so this model can not be assessed in that article. In addition, the GMPEs proposed (in the first places) Delavaud et al. (2012) do not consider the source effect with as much detail as the model of Campbell and Bozorgnia (2014), in this application it does not make sense to define the sources precisely if simpler models are later employed in the GMPEs.

Nevertheless, the focus of the paper is the definition of the source model and the distribution of potential between zone and faults. The choice of the GMPE is a secondary issue in this regard.

P9L27-28. What is said here does not prevent double counting at all. To prevent double counting every earthquake that is assigned to its causative fault should be removed from the rate estimate of the zone and vice versa to ensure it is counted only once.

The problem, precisely in a hybrid model of zones and faults, is to identify which earthquakes are associated to the zone and which to the fault. The events of the seismic catalog are not classified between zones and faults, but that a recurrence is established for the faults from the slip rate and another for the residual zone from the Catalogue. But this in turn will contain earthquakes that will have occurred in the fault, and if they are not easily identified they will be counted twice: one explicit in the area and another implicit in the fault. Most of the hybrid models, including that of adopting a solution to identify the events in the two types of sources: establish a Cut Magnitude Mc and consider Mw <Mc for the zone and Mw> Mc for the fault. But as we have already indicated, our approach tries to avoid this simplification and proposes a procedure for sharing, avoiding duplication. This question is key.

The results of the various calculations are very poorly presented. The fault parameter estimates are not provided at all, and other results are provided only in aggregated form. It is thus very hard to judge the validity of the hazard results if the results of the intermediate calculations are missing.

We include an annex containing a table with the fault parameters included in the study. These data are taken from the QAFI database. Including in this table all the intermediate results for each fault would be too lengthy.

In general, I found the English grammar and the organization of the manuscript to be rather poor. For example, there are sentences in the results that belong to the discussion. I've already commented on the lack of referencing. The symbols used in the equations are never explained. There is often confusion between symbols used for seismic moment and earthquake magnitude (M, M0, m, which is which?). The same also for the b-value (b or beta?). The units in the vertical axis of diagrams in figures 3 and 4 are unclear. Overall, the figure captions do not help much to understand what the figures show.

English has been corrected by a native person in this version.

My conclusion is that this manuscript is not fit for publication. I also cannot see how this manuscript can evolve quickly to an acceptable standard for the readers of NHESS and thus recommend it be rejected.

I don't give more technical suggestions here on how to improve the manuscript because they won't be useful until the major flaws are fixed. References Cornell CA (1968) Engineering seismic risk analysis.

Bull Seismol Soc Am 58:1583– 1606. Delavaud E., Cotton F., Akkar S., Scherbaum F., Danciu L., Beauval C., Drouet S., Douglas J., Basili R., Sandikkaya M., Segou M., Faccioli E., Theodoulidis N. (2012). Toward a ground-motion logic tree for probabilistic seismic hazard assessment in Europe. Journal of Seismology, 16(3), 451-473, doi: 10.1007/s10950-012-9281-z. Kanamori, H. & Brodsky, E.E., 2001. The physics of earthquakes, Phys. Today, 54(6), 34–40. Leonard, M., 2014. Self-Consistent Earthquake Fault-Scaling Relations: Update and Extension to Stable Continental Strike-Slip Faults. Bulletin of the Seismological Society of America, doi: 10.1785/0120140087. Woessner J., Danciu L., Giardini D., Crowley H., Cotton F., Grünthal G., Valensise G., Arvidsson R., Basili R., Demircioglu M., Hiemer S., Meletti C., Musson R.W., Rovida A., Sesetyan K., Stucchi M., and the SHARE consortium. (2015). The 2013 European Seismic Hazard Model - Key Components and Results. Bulletin of Earthquake Engineering, 13, 3553-3596, doi: 10.1007/s10518-015-9795-1.

---

## Author Comment (AC5) · 20 Jun 2018

Thank you for your time and the remarks on the paper. We acknowledge that your comments significantly improve the original manuscript. Below we provide response to your points with more detail. SPECIFIC COMMENTS: Abstract - Lines 9-11 page 1: The text has been modified Lines 15-17 pag 1: This is removed from the abstract and the text changed Line 19 page 1 and 1 Introduction: References have been included Line 28 page 1: It has been added to the text Line 2 page 2: References have been included Line 3 page 2: References have been included Line 22 page 2: References have been included Lines 28-32 page 1: We agree on that hose limitations

are inherent to the probabilistic method. With this paragraph, we just wanted to clarify that we do not try to solve them, but only to take them into account in the proposed approach. The maximum magnitudes associated to each fault are calculated, and their occurrence rates are estimated using a GR recurrence model. This model, considering Mmax of faults, is included in seismic hazard calculations. The MmaxC value is only considered for the distribution of seismic potential, but not for the input recurrence model incorporated in the hazard model. 2 Source hybrid approach (zones & faults) for hazard estimation. We consider that it is better to focus the paper in the specific features of the proposed approach. The manuscript is structured accordingly focusing on the description of the source hybrid approach. However, we include the reference to the classical method included in this study: IGN-UPM Working Group (2013). Lines 10-16 page 3: The text has been modified Line 20 page 4: The text is modified Line 22 page 4: The text is modified Line 18 page 5: This issue is included in the Discussion Line 14 pag 7: A reference to QAFI is included, as well as an annex with fault information Line 9 page 8: Details on how seismic sources are used in the CM can be found in IGN-UPM Working Group (2013) (reference included in the paper) We have not provided further details on this paper because it is not the objective of this paper. Line 6 page10: References are included References - New references are included FIGURES - Figure 2: The figure is explained with more detail TECHNICAL CORRECTIONS: Line 17 page 1: The text is modified

Please also note the supplement to this comment:
https://www.nat-hazards-earth-syst-sci-discuss.net/nhess-2018-28/nhess-2018-28-AC5-supplement.pdf

[Figure]

**Supplement:**

A. Grezio

Thank you for your time and the remarks on the paper. We acknowledge that your comments significantly improve the original manuscript. Below we provide response to your points with more detail.

GENERAL COMMENTS:

The paper "Approach for combining faults and area sources in seismic hazard assessment: Application in southeastern Spain, by Alicia RivasMedina et al." can add interesting discussions points on the seismic hazard issues because of the hybrid source model. However, major changes are required to improve the paper. The background knowledge of the application area (Southeastern Spain) is poorly described and the choice of Mmaxc should be discussed more deeply.

SPECIFIC COMMENTS:

Abstract - Lines 9-11 page 1: Instead of " . . .model composed by faults as independent entities and zones (containing the residual seismicity). The seismic potential of both types of sources is derived from different data: for the zones, the recurrence model is estimated from the seismic catalog. For fault sources, it is inferred from kinematic parameters derived from paleoseismicty and GNSS measurements" a suggested re-writing could be (in the list put first fault and then zones): ". . . model composed by faults as independent entities and zones (containing the residual seismicity). The seismic potential of both types of sources is derived from different data: for the fault sources it is inferred from kinematic parameters derived from paleoseismicty and GNSS measurements, and for the zones the recurrence model is estimated from the seismic catalog".

The text has been modified

Lines 15-17 pag 1: The concept of the Max Magnitude in the abstract (stated by the following words "This is derived from a completeness analysis and can be lower than the Mmax generated by the faults, taking into account that their the recurrence period can be higher than the observation period of the catalog") starts a discussion but it is not fully developed, it is just mentioned here and needs to be better explained in the "Discussion and Conclusions" section or when the results are presented. This part in the abstract should be removed and/or re-written.

This is removed from the abstract and the text changed

Line 19 page 1: It is required an explanation of ". . .a seismic hazard model using the traditional zone". Is there any reference of this model? What do the authors mean by the word "traditional"? This part should be re-written.

1 Introduction - There are NO REFERENCES in the Introduction. This is an anomalous way of presenting a scientific paper in a peer review journal because the paper gives a narrow view of the subject. Please add appropriated references in the text. The Introduction looks more a discussion to motivate the paper than a wide presentation of the seismic hazard problem in the application region. Also, there is no mention of the previous seismic studies carried out in Southeastern Spain. Line 27 page 1: References are required near the words ". . . in the last years, as more studies".

References have been included

Line 28 page 1: Please write the acronimo as "GNSS (Global Navigation Satellite System)".

It has been added to the text

Line 2 page 2: Please add reference after " In most practical cases".

References have been included

Line 3 page 2: Please add reference after "Other approaches".

References have been included

Line 22 page 2: Please add reference after " Some authors ".

References have been included

Lines 28-32 page 1: The sentences "The approach presented in this paper, as all probabilistic seismic hazard models, face the challenging question of estimating the expected ground motions with the basis of a short period of observations of earthquake occurrences and limited geological data (with significant uncertainty) to construct recurrence models. The purpose of this work is not to solve this challenge, but rather, to propose a model that contains different types of seismic sources (faults and zones) and distributes the seismic potential appropriately, avoiding double-counting and considering periods of completeness." present the purpose of this study in a superficial way. Those lines should be re-written. If you want to start a probabilistic hazard assessment, firstly you consider the potential Max magnitudes generated by the faults, and then you associate a low probability of earthquake occurrence with them on the basis of your study and considering the relative uncertainties. Certainly you don't exclude those max magnitudes just because of the completeness of the catalogue. A probabilistic hazard assessment should overcome those limitations. Again the choice of the Mmaxc should be properly explained, doubts on that should be solved and motivated by the authors in the paper. This part should be re-considered for the discussion in the last section.

We agree on that hose limitations are inherent to the probabilistic method. With this paragraph, we just wanted to clarify that we do not try to solve them, but only to take them into account in the proposed approach.

The maximum magnitudes associated to each fault are calculated, and their occurrence rates are estimated using a GR recurrence model. This model, considering Mmax of faults, is included in seismic hazard calculations. The MmaxC value is only considered for the distribution of seismic potential, but not for the input recurrence model incorporated in the hazard model.

2 Source hybrid approach (zones & faults) for hazard estimation - This section is presented in a schematic way, but should be completed with the description of the "Classical Method" which is also used in comparison with the Hybrid Model.

We consider that it is better to focus the paper in the specific features of the proposed approach. The manuscript is structured accordingly focusing on the description of the source hybrid approach. However, we include the reference to the classical method included in this study: IGN-UPM Working Group (2013).

Lines 10-16 page 3 : The definitions should be clearer, in particular CP(m) and PC(m) need a longer explanation.

The text has been modified

Line 20 page 4: Please write GR after Gutenberg-Richter and may be write a reference for that (Gutenberg and Richter 1944 or 1954).

The text is modified

Line 22 page 4: In Eq 4 it should be written what "d" and "beta" are.

The text is modified

Line 18 page 5: The authors state " Considering that the fault may generate events with magnitudes larger than MMaxC , the corresponding distribution of seismic potential in the interval (MMaxC , MMax ] is calculated by extrapolation of the recurrence model with the last b-value adjusted". This concept should be further discussed to overcome the limitations raised by the choice of the Mmaxc.

This issue is included in the Discussion

3 Application of the approach in southeast Spain - It is not clear how the results in the application region were computed with the Classical Method.

Line 14 pag 7: Please explain what QAFI database is and/or write references for that.

A reference to QAFI is included, as well as an annex with fault information

Line 9 page 8: The "Classical Method" should be better explained, it is just mentioned for the Fig 6.

Details on how seismic sources are used in the CM can be found in IGN-UPM Working Group (2013) (reference included in the paper) We have not provided further details on this paper because it is not the objective of this paper.

4 Discussion and conclusions - The discussion of the results and the conclusions should be include the point raised previously.

Line 6 page10: Please add references for 2009 L'Aquila and 2011 Lorca events.

References are included

References - They are a poorly and unsatisfactory list of the other paper on this subject. The References are simply the ones used to carried out the computation.

New references are included

FIGURES - Figure 2: more explanations are needed in the caption, in particular about "AR".

The figure is explained with more detail

TECHNICAL CORRECTIONS:

Line 17 page 1: delete "the" before "recurrence".

The text is modified

Line 10 page: difficult to read. Instead of the sentence "using the records which origin time and magnitude are contained in the period for which the catalog can be consid ered complete" a suggested change could be: "using the records with origin time and magnitude contained in the period for which the catalog can be considered complete".

The text is modified

Line 27 page 8: This part on region 30 should be immediately after "return periods", to complete the part on hazard map.

The text is modified

Line 31 page 8: the sentence with "The hazard curves" should start a new paragraph, on the hazard curves.

The text is changed

---

## Author Comment (AC6) · 20 Jun 2018

Thank you for your time and remarks, with help improving the manuscript significantly. More than 33 references have been included in the manuscript and figure captions are completed The English is reviewed, The discussion section is separated from the conclusions.

---

## Author Comment (AC8) · 20 Jun 2018

**Annex**

*Faults and fault segments included in seismic hazard calculations. L: lenght, Zmin, minimum depth, Zmax: maximum depth*

| ID | Fault and fault segment name | Strike (°) | Dip (°) | Rate (°) | L (km) | Zmin (km) | Zmax (km) | Slip rate (mm/año) |
|---|---|---|---|---|---|---|---|---|
| ES601 | Crevillente (Sector Murcia) (1/3) | 252 | 90 | 0 | 30 | 0 | 15 | 0.100 |
| ES602 | Crevillente (Sector Murcia) (2/3) | 243 | 90 | 0 | 44 | 0 | 15 | 0.100 |
| ES603 | Crevillente (Sector Murcia) (3/3) | 253 | 90 | 0 | 18 | 0 | 15 | 0.100 |
| ES604 | Crevillente (Sector Alicante) (1/2) | 250 | 90 | 0 | 30 | 0 | 15 | 0.070 |
| ES605 | Jumilla (Sector Murcia) (1/3) | 53 | 90 | 0 | 35 | 0 | 11 | 0.010 |
| ES606 | Jumilla (Sector Murcia) (2/3) | 57 | 90 | 0 | 30 | 0 | 11 | 0.010 |
| ES607 | Jumilla (Sector Murcia) (3/3) | 51 | 90 | 0 | 15 | 0 | 11 | 0.050 |
| ES608 | Jumilla (Sector Valencia) (2/2) | 65 | 90 | 0 | 35 | 0 | 15 | 0.050 |
| ES609 | Palomares (1/2) | 14 | 90 | 0 | 45 | 0 | 8 | 0.040 |
| ES610 | Palomares (2/2) | 37 | 90 | 0 | 30 | 0 | 8 | 0.050 |
| ES611 | Corredor de Las Alpujarras | 84 | 90 | 0 | 50 | 0 | 12 | 0.050 |
| ES612 | Las Moreras  - Esc. M(1/3) | 104 | 90 | 0 | 15 | 0 | 8 | 0.050 |
| ES613 | Las Moreras  - Esc. M(2/3) | 102 | 90 | 0 | 30 | 0 | 8 | 0.050 |
| ES614 | Las Moreras  - Esc. M(3/3) | 79 | 90 | 0 | 50 | 0 | 8 | 0.050 |
| ES615 | Carrascoy | 56 | 90 | 20 | 32 | 0 | 12 | 0.540 |
| ES616 | Torrevieja | 308 | 90 | 0 | 15 | 0 | 12 | 0.075 |
| ES617 | San Miguel de Salinas | 304 | 90 | 0 | 30 | 0 | 12 | 0.300 |
| ES618 | Bajo Segura (1/3) | 83 | 60 | 0 | 9 | 1 | 12 | 0.350 |
| ES619 | Bajo Segura (2/3) | 84 | 60 | 0 | 9 | 1 | 12 | 0.230 |
| ES620 | Bajo Segura (3/3) | 77 | 60 | 0 | 8 | 1 | 12 | 0.120 |
| ES622 | Muro de Alcoy (o de Mariola) | 335 | 60 | -90 | 6 | 0 | 15 | 0.200 |
| ES624 | Benasau | 300 | 60 | -90 | 5 | 0 | 15 | 0.200 |
| ES625 | Alhamilla Sur | 67 | 80 | 0 | 25 | 0 | 8 | 0.050 |
| ES626 | Alhama de Murcia (1/4) | 215 | 70 | 20 | 28 | 0 | 12 | 0.500 |
| ES627 | Alhama de Murcia (2/4) | 238 | 90 | 18 | 20 | 0 | 12 | 0.300 |
| ES629 | Alhama de Murcia (4/4) | 225 | 90 | 0 | 25 | 0 | 12 | 0.070 |
| ES630 | Carboneras (1/2) | 48 | 90 | 10 | 111 | 0 | 11 | 1.101 |
| ES631 | Amarguillo | 16 | 90 | 90 | 8 | 0 | 8 | 0.100 |
| ES632 | Las Viñas | 285 | 80 | 70 | 6 | 0 | 5 | 0.400 |
| ES634 | Jumilla (Sector Valencia) (1/2) | 60 | 90 | 0 | 40 | 0 | 15 | 0.050 |
| ES635 | Falla de Polopos (1/2) | 85 | 75 | 0 | 13 | 5 | 10 | 0.050 |
| ES636 | Falla de Polopos (2/2) | 95 | 80 | 0 | 12 | 5 | 10 | 0.071 |
| ES637 | Borde NO de Sierra Arana | 202 | 60 | -90 | 7 | 0 | 10 | 0.120 |
| ES638 | Norte de Cubillas | 72 | 50 | -90 | 9 | 0 | 5 | 0.060 |
| ES639 | E de Cubillas | 153 | 50 | -90 | 5 | 0 | 3 | 0.080 |
| ES640 | Oeste de Cubillas | 332 | 60 | -90 | 5 | 0 | 4 | 0.150 |
| ES641 | Obéilar - Pinos Puente | 80 | 75 | 0 | 8 | 0 | 10 | 0.500 |
| ES643 | Tocón-Obéilar | 79 | 45 | -90 | 9 | 0 | 3 | 0.100 |
| ES644 | Alitaje | 144 | 50 | -90 | 6 | 0 | 5 | 0.100 |
| ES646 | Pinos Puente | 140 | 60 | -90 | 9 | 0 | 10 | 0.400 |
| ES648 | Pedro Ruiz | 151 | 50 | -90 | 6 | 0 | 5 | 0.100 |
| ES650 | Daimuz Bajo | 149 | 60 | -90 | 7 | 0 | 3 | 0.080 |
| ES651 | Atarfe | 134 | 60 | -90 | 10 | 0 | 10 | 0.150 |
| ES653 | El Fargue-Jun | 138 | 60 | -90 | 12 | 0 | 10 | 0.350 |
| ES655 | Escóznar | 330 | 60 | -90 | 10 | 0 | 3 | 0.100 |
| ES656 | Santa Fe | 326 | 60 | -90 | 13 | 0 | 10 | 0.200 |
| ES658 | Granada | 150 | 60 | -90 | 17 | 0 | 10 | 0.380 |
| ES660 | Belicena-Alhendín | 326 | 60 | -90 | 10 | 0 | 5 | 0.200 |
| ES662 | Huenes | 212 | 65 | 0 | 5 | 0 | 10 | 0.250 |
| ES663 | Canales | 233 | 80 | 0 | 8 | 0 | 10 | 0.080 |
| ES664 | Dílar | 140 | 60 | -90 | 8 | 0 | 10 | 0.330 |
| ES666 | Padul | 134 | 55 | -90 | 15 | 0 | 5 | 0.350 |
| ES668 | Norte del Silleta | 268 | 60 | -90 | 8 | 0 | 5 | 0.040 |

| ID | Fault and fault segment name | Strike (°) | Dip (°) | Rate (°) | L (km) | Zmin (km) | Zmax (km) | Slip rate (mm/año) |
|---|---|---|---|---|---|---|---|---|
| ES669 | Padul-Dúrcal | 130 | 50 | -90 | 13 | 0 | 5 | 0.350 |
| ES671 | Escúzar | 268 | 60 | -90 | 16 | 0 | 5 | 0.030 |
| ES672 | Noroeste de Játar | 322 | 60 | -90 | 9 | 0 | 5 | 0.080 |
| ES673 | Norte de Sierra Tejeda | 282 | 75 | 0 | 23 | 0 | 10 | 0.125 |
| ES674 | Tablate | 201 | 60 | 0 | 5 | 0 | 5 | 0.080 |
| ES675 | Albuñuelas | 278 | 50 | -90 | 10 | 0 | 5 | 0.140 |
| ES677 | Béznar-Ízbor | 328 | 70 | -90 | 6 | 0 | 5 | 0.080 |
| ES678 | Lanjarón | 88 | 90 | 0 | 7 | 0 | 10 | 0.300 |
| ES679 | Vélez de Benaudalla | 218 | 30 | 0 | 6 | 0 | 3 | 0.150 |
| ES680 | Norte de Sierra de Lújar | 90 | 90 | 0 | 9 | 0 | 10 | 0.150 |
| ES681 | Ventas de Zafarraya | 282 | 60 | -102 | 23 | 0 | 15 | 0.350 |
| ES682 | Galera | 228 | 50 | 0 | 25 | 0 | 5 | 0.160 |
| ES684 | Botardo-Alfahuara | 114 | 90 | 180 | 22 | 0 | 5 | 0.030 |
| ES686 | Baza | 341 | 65 | -90 | 37 | 0 | 10 | 0.330 |
| ES695 | Zamborino | 148 | 65 | -90 | 24 | 0 | 5 | 0.160 |
| ES699 | Graena | 321 | 70 | -90 | 5 | 0 | 5 | 0.070 |
| ES716 | Albox | 255 | 50 | 110 | 10 | 0 | 10 | 0.020 |
| ES717 | El Acebuchal | 37 | 90 | 0 | 9 | 0 | 8 | 0.050 |
| ES718 | Los Alamillos | 230 | 90 | 0 | 7 | 0 | 8 | 0.050 |
| ES719 | El Carrascal | 4 | 90 | 0 | 5 | 0 | 8 | 0.010 |
| ES720 | Sierra de las Nieves | 18 | 90 | 0 | 11 | 0 | 11 | 0.032 |
| ES721 | La Robla | 28 | 90 | 0 | 12 | 0 | 11 | 0.021 |
| ES722 | Campanillas | 216 | 90 | 0 | 8 | 0 | 11 | 0.019 |
| ES723 | Cartama | 70 | 10 | 0 | 6 | 0.5 | 2.5 | 0.050 |
| ES724 | Mijas | 80 | 60 | 0 | 19 | 2.5 | 13 | 0.081 |
| ES725 | Villafranco de Guadalhorce | 70 | 10 | 0 | 7 | 0.5 | 2.5 | 0.050 |
| ES728 | Socovos (1/2) | 295 | 85 | 180 | 40 | 0 | 15 | 0.040 |
| ES729 | Socovos (2/2) | 272 | 75 | 135 | 36 | 0 | 15 | 0.040 |
| ES730 | Pozohondo | 135 | 75 | -120 | 11 | 0 | 11 | 0.103 |
| ES732 | Lietor (1/2) | 126 | 85 | -140 | 21 | 1 | 15 | 0.100 |
| ES733 | Lietor (2/2) | 92 | 85 | -140 | 19 | 1 | 15 | 0.100 |
| ES736 | E de Guadix | 142 | 90 | -90 | 11 | 0 | 5 | 0.020 |
| ES738 | Suroeste del Negratín | 226 | 60 | -90 | 3 | 0 | 5 | 0.150 |
| ES739 | Oeste del Negratín | 180 | 70 | -90 | 8 | 0 | 5 | 0.020 |
| ES740 | Hijate | 134 | 60 | -90 | 6 | 0 | 5 | 0.040 |
| ME005 | Cabo de Cullera. Falla Occidental | 327 | 60 | -90 | 28 | 1.5 | 15 | 0.030 |
| ME006 | Cabo de Cullera. Falla Central-Occidental | 331 | 60 | -90 | 25 | 1.5 | 15 | 0.030 |
| ME007 | Cabo de Cullera. Falla Central-Oriental | 197 | 60 | -90 | 48 | 1.5 | 15 | 0.020 |
| ME008 | Cabo de Cullera. Falla Oriental | 188 | 60 | -90 | 16 | 1.5 | 15 | 0.020 |
| ME009 | Suroeste de Cuenca de Columbretas | 335 | 60 | -90 | 10 | 1.5 | 15 | 0.020 |
| ME015 | Carboneras (2/2) | 58 | 90 | 0 | 36 | 0 | 11 | 0.040 |
| ME021 | Bajo Segura segmento marino | 81 | 60 | 0 | 30 | 1 | 12 | 0.120 |

---

## Editor Decision (ED1)

Editor comment on NHESS-2018-28

All referees agree that this paper presents an original and valuable contribution in Probabilistic Seismic Hazard Assessment. The short comments are also positive about the publication of the paper. Referee 1 and SC1 highlights the fact that this can be a new way of combining faults and zones – hybrid approach as referred by SC1. Also, referees agree that the paper need revision but it should be considered for publication. The quantification of uncertainties seems to be an important point of the paper.

As handling editor of this paper, I invite the authors to submit a revised paper of the manuscript. The authors should pay special attention on the comments raised all referees on Maximum Magnitude and correct (see referee's #1 and #2)

Special effort must be put in the revision of the English. Also, some of the referees point out the fact the lack of references in the introduction. Special care must be put in figure captions. The answer uploaded as AC7 includes significant corrections. The authors should proceed from this version.